# SLM: A Smoothed First-Order Lagrangian Method for Structured Constrained Nonconvex Optimization

**Songtao Lu**
IBM Research, Thomas J. Watson Research Center
Yorktown Heights, NY 10598
songtao@ibm.com

## Abstract

Functional constrained optimization (FCO) has emerged as a powerful tool for solving various machine learning problems. However, with the rapid increase in applications of neural networks in recent years, it has become apparent that both the objective and constraints often involve nonconvex functions, which poses significant challenges in obtaining high-quality solutions. In this work, we focus on a class of nonconvex FCO problems with nonconvex constraints, where the two optimization variables are nonlinearly coupled in the inequality constraint. Leveraging the primal-dual optimization framework, we propose a smoothed first-order Lagrangian method (SLM) for solving this class of problems. We establish the theoretical convergence guarantees of SLM to the Karush-Kuhn-Tucker (KKT) solutions through quantifying dual error bounds. By establishing connections between this structured FCO and equilibrium-constrained nonconvex problems (also known as bilevel optimization), we apply the proposed SLM to tackle bilevel optimization oriented problems where the lower-level problem is nonconvex. Numerical results obtained from both toy examples and hyper-data cleaning problems demonstrate the superiority of SLM compared to benchmark methods.

## 1 Introduction

In this paper, we consider a structured constrained optimization framework that has broad applications in diverse machine learning problems. The goal of this problem is to minimize the objective function using two blocks of optimization variables which are coupled in the functional constraint. More formally, we express this class of nonlinear constrained programming problems as follows:

$$\min_{x \in \mathcal{X}, y} f(x, y), \quad \text{s.t.} \quad g(x, y) - g^*(x) \le \delta \tag{1}$$

where feasible set $\mathcal{X}$ is convex and compact, the objective function $f(x, y)$ is smooth and nonconvex, constraint $g(x, y)$ is smooth and nonconvex and it also satisfies the Polyak-Łojasiewicz (PŁ) condition with respect to (w.r.t.) $y$, $g^*(x) \triangleq \min_y g(x, y)$ (which is also called value function [1]) is obtained by minimizing $g(x, y)$ over $y$, and constant $\delta > 0$.

When $\delta = 0$, problem (1) reduces to the classic bilevel optimization (BO) problem (aka mathematical programs with equilibrium constraints [2]). More specifically, it takes the form of

$$\min_{x \in \mathcal{X}, y} f(x, y), \quad \text{s.t.} \quad y \in \mathcal{S}(x) \triangleq \arg\min_{y'} g(x, y'), \tag{2}$$

where $f(x, y)$ and $g(x, y')$ denote the upper-level (UL) and lower-level (LL) objective functions, $\mathcal{S}(x)$ represents the set that contains the global optimal solutions of the LL problem w.r.t. the block-$y'$. This BO formulation has been shown to be a useful framework for modeling various multi-task

37th Conference on Neural Information Processing Systems (NeurIPS 2023).

Table 1: Comparison of existing related representative works on nonconvex functional constrained and/or bilevel optimization, where non-singleton: the existence of multiple global optimal solutions at the LL problem, oracle: requirements of accessing first-order and/or second-order derivative of either UL or LL loss functions, LL/C: the property that the LL objective funciton/constraint satisfies in problem (2) or (1), $\widetilde{\mathcal{O}}$: denotes that the iteration complexity hides the dependence on the hyperparameters, e.g., $\delta$, cvx: convex, ncvx: nonconvex, scvx: strongly convex.

| Algorithms | Problem | Opt. Framework | Oracle | Non-Singleton | LL/C | Rate |
|---|---|---|---|---|---|---|
| ConEx [12] | | primal-dual | first-order | ✓ | ncvx | $\mathcal{O}\left(\frac{1}{\epsilon^{3/2}}\right)$ |
| ITD/AID [13] | (2) | ITD/AID | second-order | | scvx | $\mathcal{O}\left(\frac{1}{\epsilon}\right)$ |
| F$^2$SA [14] | (2) | penalty | first-order | | scvx | $\mathcal{O}\left(\frac{1}{\epsilon^{3/2}}\right)$ |
| BOME [15] | (2) | penalty | first-order | ✓ | PŁ | $\mathcal{O}\left(\frac{1}{\epsilon^3}\right)$ $\mathcal{O}\left(\frac{1}{\epsilon^4}\right)$ |
| V-PBGD [16] | (2) | penalty | first-order | ✓ | PŁ | $\widetilde{\mathcal{O}}\left(\frac{1}{\epsilon}\right)$ |
| **SLM** | (1) | primal-dual | first-order | ✓ | PŁ | $\widetilde{\mathcal{O}}\left(\frac{1}{\epsilon}\right)$ |

machine learning problems, e.g., hyper-parameter optimization [3, 4], actor-critic schemes in reinforcement learning [5], meta-learning [6, 7], AUC maximization [8], distributed bilevel optimization [9, 10, 11], etc. In these scenarios, the decision/optimization variables are coupled in both levels of the minimization problems and the evaluation of the UL gradient depends on the optimal solution at the LL. Consequently, it is necessary to optimize the UL and LL loss functions jointly w.r.t. the block variables $x$ and $y$.

## 1.1 Related Work

**Bilevel Optimization**. Motivated by the numerous applications of BO, developing the efficient solvers for handling these problems has attracted great interest in the optimization community. There are two major directions: 1) focus on solving the BO problem in a general manner; 2) improve the computation or iteration complexity of the BO algorithms. It has been shown in [17] that a first-order method can find the global optimal solution when the UL objective function is strongly convex and the LL problem is convex. However, the strong convexity assumption restricts the applicability of the algorithm to many real-world machine learning problems. In contrast, some recent works have considered the case where the UL objective function is nonconvex and have shown that, given an oracle that computes the Jacobian and the inverse of Hessian matrices of the LL objective function, the complexity of finding an $\epsilon$-stationary point is $\mathcal{O}(\epsilon^{-1})$ [18, 13], which is the same as that of gradient descent. But this setting is still restrictive as they still assume that the LL problem is strongly convex.

**Non-Singleton at the LL**. When the LL problem is not strongly convex, the computation of the UL gradient $\nabla f(x, \mathcal{S}(x))$ can become ill-posed due to the presence of multiple optimal solutions. To address this issue, a new metric called $(\delta, \epsilon)$-Goldstein stationary point was introduced in [19], which aims to capture the discontinuity of $\nabla f(x, \mathcal{S}(x))$. Under the assumption that $f(x, \cdot)$ and $g(x, \cdot)$ are convex, it was shown that an inexact gradient-free method can find the Goldstein stationary point at a rate of $\mathcal{O}(\delta^{-1}\epsilon^{-4})$. In addition to the new metric used for theoretical convergence analysis, the concept of Hölderian error bound was employed in [20], which imposes further conditions on the shape of the LL objective function. For a class of simple BO problems where only one set of block variables needs to be optimized, the conditional gradient-based bilevel optimization (CG-BiO) algorithm proposed in [20] was shown to converge to stationary points at a rate of $\mathcal{O}(\epsilon^{-1})$ or $\mathcal{O}(\epsilon^{-2})$, depending on whether the Hölderian error bound holds.

In [21], the author considered the case of a nonconvex LL objective function but with an assumption that eliminates the presence of zero eigenvalues of the LL Hessian matrix. Another property used to characterize the landscape of the LL function is the PŁ condition. Based on the formulation in (1), researchers [16, 15] developed penalty methods for solving BO problems and discussed the relationship between local/global optimal solutions of the penalized problem formulation and the constrained problem (2) (i.e., when $\delta = \epsilon$ in (1)) . It was shown that the value-gap-based penalty-based bilevel gradient descent (V-PBGD) algorithm [16] can converge to stationary points of the reformulated single-level problem. Additionally, under the assumption of the constant rank con-

straint quantification condition, the bilevel optimization made easy (BOME) algorithm developed in [15] can achieve Karush-Kuhn-Tucker (KKT) points of the BO problem in (2).

**Computational Issue**. Another line of research focuses on reducing the computational complexity of implementing BO optimization algorithms. As mentioned earlier, computing the UL gradient can be computationally expensive. Existing works have employed strategies such as the Neumann series for calculating the inverse of Hessian matrices [18], or utilized reverse-mode iterative differentiation (ITD) and approximate implicit differentiation (AID) techniques [22, 13] for hypergradient computation. However, these strategies introduce a double-loop structure in the algorithm design, requiring users to specify hyperparameters for the inner loop subroutine. Subsequently, single-loop BO algorithms have been developed [23, 24], which alleviate the burden of hyperparameter tuning in the inner loop. Nevertheless, they still require the computation of the Jacobian matrix and Hessian-vector product.

To circumvent these computations, some existing works reformulate the BO problem as a single-level functional constrained optimization (FCO) problem, as shown in (1), and apply well-established constrained optimization techniques, such as primal-dual methods, for solving the BO problem using only gradients. In [25], it was shown that when the UL problem is strongly convex and the LL problem is convex, the primal-dual algorithm can find the global optimal solution at a rate of $\mathcal{O}(\epsilon^{-1/2})$. For the case where the UL problem is nonconvex, the fully first-order stochastic approximation (F$^2$SA) method proposed in [14] can find an $\epsilon$-stationary solution at a rate of $\mathcal{O}(\epsilon^{-3/2})$ when the LL problem is strongly convex. Similarly, penalty methods [26, 16] also just require the computation of UL and LL gradients while achieving the desired optimization results.

**Functional Constrained Optimization**. The formulation based on function values establishes a connection between FCO and BO [1, 27, 28]. By leveraging existing well-developed constrained optimization methods [12, 29, 30], efficient solutions to the BO problem can still be obtained. When the block variables $x$ and $y$ are linearly coupled, the alternating direction method of multipliers (ADMM) [31] has emerged as a successful approach by decoupling the optimization process into subproblems. Recent advancements in smoothed augmented Lagrangian methods [30, 32, 33] have improved the convergence rates for solving nonconvex linearly constrained problems. For nonlinear functional constraints with one block of variables, the constraint extrapolation (ConEx) method [12] has shown promising results in finding KKT points with a convergence rate of $\mathcal{O}(\epsilon^{-3/2})$ for nonconvex smooth functions. Besides, the properties of PŁ functions studied in min-max optimization problems [34] are expected to be useful in demonstrating the continuity of $\nabla f(x, \mathcal{S}(x))$ for BO problems within the framework of primal-dual methods. A comparison between our work and closely related previous works on FCO and BO is shown in Table 1.

### 1.2 Main Contributions of This Work

In this work, we propose a novel approach called the smoothed first-order Lagrangian method (SLM) to tackle the FCO problem presented in (1). The key advantage of SLM is that it only requires the computation of gradients, making it computationally efficient. Remarkably, under mild assumptions and a local regularity condition, we establish a strong error bound for SLM to solve this class of problems, and prove that SLM can converge to the KKT points even when the nonconvex objective in the constraint has multiple optimal solutions. To summarize, the main contributions of this work are highlighted as follows:

▶ This class of structured FCO problems is generic and can be used to formulate multiple machine learning problems. The proposed SLM is computationally efficient as it relies only on the gradient of the objective functions. Moreover, the obtained convergence rate of the proposed SLM, which is $\mathcal{O}(\epsilon^{-1})$, is optimal for finding the $\epsilon$-KKT points of this class of problems.

▶ SLM can be applied to solve nonconvex bilevel related optimization problems in the presence of nonconvexity and non-singleton at the lower level, when the LL objective function satisfies the PŁ condition.

▶ The numerical results showcase the competitiveness of the proposed SLM compared to state-of-the-art BO algorithms in both a toy example and a data hyper-cleaning problem w.r.t. convergence speed and achievable test accuracy.

Due to space constraints, all technical proofs are provided in the supplementary material.

Notations: The distance between $y$ and the set $\mathcal{S}(x)$ is defined as $d_{\mathcal{S}(x)}(y) \triangleq \min_{y' \in \mathcal{S}(x)} \|y - y'\|$. A high-dimensional ball with a radius of $R$ is denoted as $\mathcal{B}(R)$.

# 2 Smoothed First-Order Lagrangian Method

In this section, we will introduce the three-layer structure of our first-order method designed to solve problem (1). Towards this end, we can write the Lagrangian of this constrained problem as follows:

$$\mathcal{L}(x, y; \lambda) \triangleq f(x, y) + \lambda(g(x, y) - g^*(x) - \delta) \tag{3}$$

where the nonnegative $\lambda$ denotes the Lagrange multiplier or dual variable.

Next, it is natural to have

$$\nabla_x \mathcal{L}(x, y; \lambda) = \nabla_x f(x, y) + \lambda(\nabla_x g(x, y) - \nabla_x g^*(x)), \tag{4a}$$

$$\nabla_y \mathcal{L}(x, y; \lambda) = \nabla_y f(x, y) + \lambda \nabla_y g(x, y), \tag{4b}$$

$$\nabla_\lambda \mathcal{L}(x, y; \lambda) = g(x, y) - g^*(x) - \delta. \tag{4c}$$

In contrast to the classical primal-dual algorithm design, here, $g^*(x)$ is typically unknown. Based on the definition of $g^*(x)$, it is not hard to know that obtaining a closed-form expression for $\nabla g^*(x)$ requires the computation of the Jacobian matrix and the inverse of Hessian matrice even for the case when $g(x, y)$ is strongly convex w.r.t. $y$. Although many existing works have provided the iterative methods of computing these quantities for each iteration [13, 23, 24], the calculational cost and/or memory budget are still high for large-scale problems. Hence, it is motivated to have an efficient way of approximating the value and gradient of $g^*(x)$ in the algorithm design.

## 2.1 Bottom-Layer

One of the most straightforward approaches is to generate a sequence $u_t$ indexed by $t$ as an inner loop to obtain $u^*(x) \triangleq \arg\min_u g(x, u)$. When the objective function $g(x, y)$ satisfies certain properties, such as the PŁ condition or the strongly convex condition, applying gradient descent can lead to finding the global optimal solution of this subproblem with a linear convergence rate [35].

Let $r$ denote the index of the outer iteration steps. Then, the approximate of $u^*(x)$ at each iteration, i.e., $u^r$, can be estimated through the following process:

$$u_{t+1}^r = u_t^r - \gamma \nabla_u g(x^r, u_t^r), \tag{5}$$

where $u_t^r$ represents the $t$th iterate of the inner loop, $\gamma$ denotes the step-size, and $u_0^r$ is initialized within a bounded ball (which will ensure the boundedness of the iterates). After running this algorithm for $T^r$ iterations at each step, we set $u^{r+1} = u_{T^r}^r$ for updating the subsequent optimization variables.

## 2.2 Medium-Layer

Given that the quality of $g^*(x)$ can be guaranteed, the most interesting part lies in solving the constrained problem w.r.t. block $y$. Even if we assume that $g(x, y)$ follows the PŁ condition, the nonconvexity of $f(x, y)$ still poses challenges for the first-order methods in finding the optimal solution. To address this, we propose adding the quadratic proximal terms that renders the Lagrangian w.r.t. $x$ and $y$ strongly convex, as follows:

$$K(x, y, w, z; \lambda) \triangleq \mathcal{L}(x, y; \lambda) + \frac{p}{2}\|x - w\|^2 + \frac{p}{2}\|y - z\|^2. \tag{6}$$

Here, $K(x, y, w, z; \lambda)$ can be regarded as a modified Lagrangian for this problem. Let $L_f$ and $L_g$ denote as the smoothness constants of $f$ and $g$, respectively. Choosing a sufficiently large value for $p$ (specifically, $p = \Omega(L_f + \lambda L_g)$) ensures that $K(x, y, w, z; \lambda)$ becomes strongly convex.

Subsequently, we can develop the following gradient-based primal-dual algorithm for solving this constrained subproblem:

$$\lambda^{r+1} = \mathcal{P}_+ \left[ \lambda^r + \tau \nabla_\lambda \widehat{K}(x^r, y^r, w^r, z^r, u^{r+1}; \lambda^r) \right], \tag{7a}$$

$$y^{r+1} = y^r - \alpha \nabla_y K(x^r, y^r, w^r, z^r; \lambda^{r+1}), \tag{7b}$$

$$z^{r+1} = z^r + \beta(y^{r+1} - z^r), \tag{7c}$$

where $\widehat{K}(x, y, w, z, u; \lambda) \triangleq \widehat{\mathcal{L}}(x, y, u; \lambda) + \frac{p}{2}\|x - w\|^2 + \frac{p}{2}\|y - z\|^2$ and $\widehat{\mathcal{L}}(x, y, u; \lambda) \triangleq f(x, y) + \lambda(g(x, y) - g(x, u) - \delta)$, $\tau, \alpha, \beta$ are the step-sizes, and $\mathcal{P}_+$ denotes the projection of the dual variable onto a box constraint. In particular, $\mathcal{P}_+[\lambda]$ is given by $\mathcal{P}_+[\lambda] = 0$ if $\lambda \leq 0$; $\mathcal{P}_+[\lambda] = \lambda$ if $0 < \lambda \leq \Lambda$; $\mathcal{P}_+[\lambda] = \Lambda$ if $\lambda > \Lambda$.

*Remark 1.* The auxiliary sequence $z^r$ is defined as an exponentially weighted average sequence of $y^{r+1}$, which has been previously used in smoothed gradient descent ascent for nonconvex min-max optimization [36, 37] problems and in the smoothed proximal augmented Lagrangian method for linearly constrained nonconvex objective problems [32]. However, to the best of our knowledge, this is the first time that the smoothed primal-dual method is applied to solve nonconvex objective problems with nonlinear functional constraints.

### 2.3 Top-Layer

After updating the variables $y$, $z$, $u$, and $\lambda$, the variables $x$ and $w$ are updated as follows:

$$x^{r+1} = \mathcal{P}_{\mathcal{X}} \left[ x^r - \eta \nabla_x \widehat{K}(x^r, y^{r+1}, w^r, z^{r+1}, u^{r+1}; \lambda^{r+1}) \right], \tag{8a}$$

$$w^{r+1} = w^r + \beta(x^{r+1} - w^r), \tag{8b}$$

where $\mathcal{P}_{\mathcal{X}}$ denotes the projection operator, and $\eta$ is the step-size.

The detailed algorithm description is provided in Algorithm 1.

---

**Algorithm 1 S**moothed first-order **L**agrangian **M**ethod (SLM)

---

**Initialization:** step-sizes: $\tau, \eta, \alpha, \beta, \gamma$, variables: $x^1, w^1, y^1, z^1, \lambda^1$

1: **for** $r = 1, 2, \cdots, T$ **do**
2:     **for** $t = 0, 1, 2, \cdots, T^r - 1$ **do**
3:         Initialize $u_0^r \in \mathcal{B}(R)$
4:         $u_{t+1}^r = u_t^r - \gamma \nabla_u g(x^r, u_t^r)$            ▷ bottom layer
5:     **end for**
6:     $u^{r+1} = u_{T^r}^r$
7:     $\lambda^{r+1} = P_+ \left[ \lambda^r + \tau \left( g(x^r, y^r) - g(x^r, u^{r+1}) - \delta \right) \right]$   ▷ dual variable update
8:     $y^{r+1} = y^r - \alpha \left( \nabla_y f(x^r, y^r) + \lambda^{r+1} \nabla_y g(x^r, y^r) + p(y^r - z^r) \right)$   ▷ medium layer
9:     $z^{r+1} = z^r + \beta(y^{r+1} - z^r)$
10:    $x^{r+1} = \mathcal{P}_{\mathcal{X}}[x^r - \eta \left( \nabla_x f(x^r, y^{r+1}) + \lambda^{r+1}(\nabla_x g(x^r, y^{r+1}) - \nabla_x g(x^r, u^{r+1})) + p(x^r - w^r) \right)]$
11:    $w^{r+1} = w^r + \beta(x^{r+1} - w^r)$                  ▷ top layer
12: **end for**

---

## 3 Theoretical Convergence Results

Before presenting the theoretical convergence guarantees of SLM, it is necessary to introduce the following main classes of assumptions. These assumptions are essential in establishing the descent properties of some quantifiable (potential) function, enabling SLM to converge to KKT points of this class of nonconvex FCO problems. More detailed definitions and properties regarding these assumptions are deferred to the supplement.

### 3.1 Assumptions

The assumptions are mainly related to the continuity and boundedness of the objective function, as well as the mathematical property of the constraint.

    A1. (Smoothness) Assume that functions $f(x, y), g(x, y)$ are differentiable and jointly smooth with constants $L_f, L_g$ w.r.t. both $x, y$.

    A2. (Compactness) The feasible set $\mathcal{X}$ (projection friendly) is convex and compact.

    A3. (Boundedness) Assume that the objective function $f(x, y)$ is lower bounded and denoted as $\underline{f}$.

    A4. (Coercivity) The set $\{y | f(x, y) \leq R, g(x, y) - g^*(x) \leq \delta\}$ is bounded for any $R > 0$.

    A5. (PŁ condition) Function $g(x, y)$ satisfies the PŁ condition, i.e., there exists a constant $\mu_g$ such that $\|\nabla_y g(x, y)\|^2 \geq 2\mu_g(g(x, y) - g^*(x)), \forall x, y$.

*Remark 2.* Assumptions A1-A3 are standard in the optimization literature, while A4, also referred to as coerciveness, is a commonly used assumption in theoretical convergence analyses to guarantee boundedness of the iterates, as observed in methods like smoothed-GDA [36], ADMM [38], BO algorithms [39, Assumption 3], asynchronous algorithms [40, Assumption 5], etc. In practice, regularization terms are often included in the objective function, which can ensure that the level set is

bounded [41]. For instance, as suggested in [39], introducing a small $\ell_2$-penalty to a non-negative loss (e.g. cross-entropy or mean-squared loss) can establish boundedness for both the level set and the iterates. An example involving neural networks can be found in [42, Theorem 2]. A5 is a reasonable and practical assumption for neural network applications. Previous studies, such as [43], have demonstrated theoretically that when neural networks are overparametrized, the loss function satisfies the PŁ condition. Other examples include the nonconvex discounted return objective in reinforcement learning [44], and the loss function of a specific class of linear quadratic regulator models [45, Lemma 3].

*Local Regularity Condition.* Let $\bar{x}^*(w, z), \bar{y}^*(w, z)$ be the KKT solution of problem $\min_{x,y} f(x, y) + \frac{p}{2}\|x - w\|^2 + \frac{p}{2}\|y - z\|^2$, s.t. $g(x, y) \leq g^*(x) + \delta$. Then, there exist positive constants $\underline{\delta}$ and $\varrho$ such that when $\|w - \bar{x}^*(w, z)\|^2 + \|z - \bar{y}^*(w, z)\|^2 \leq \varrho$, the following inequality holds $g(\bar{x}^*(w, z), \bar{y}^*(w, z)) \geq g^*(\bar{x}^*(w, z)) + \underline{\delta}(w, z)$, where $\underline{\delta}(w, z) \geq \underline{\delta}$.

*Remark 3.* Given the feasibility of achieving the global optimal solution of $g^*(\bar{x}^*(w, z))$ and the requirement for variable $y$ to minimize both objective functions, it is reasonable to expect that $\bar{y}^*(w, z)$ can be close to $\arg\min g(\bar{x}^*(w, z), y)$ up to some very small constant. It is worth noting that this assumption is fairly mild, as we only require this condition to hold in the neighborhoods of the KKT points.

Given these assumptions, we are now in a position to provide the following theoretical convergence guarantees for SLM.

## 3.2 Convergence Rates of SLM

Let $\mathcal{G}(x^r, y^r)$ be defined as $\mathcal{G}(x^r, y^r) \triangleq [\eta^{-1}(x^r - \mathcal{P}_{\mathcal{X}}(x^r - \eta \nabla_x \mathcal{L}(x^r, y^r; \lambda^r))); \nabla_y \mathcal{L}(x^r, y^r; \lambda^r)]$. We denote $\|\mathcal{G}(x^r, y^r)\|$ as the stationary gap.

**Theorem 1.** *(Convergence Rate of SLM to the KKT Points of problem* (1)*) Suppose that A1-A5 are satisfied, and the local regularity condition holds. Assume that the iterates $\{x^r, y^r, w^r, z^r, u^r, \lambda^r\}$ are generated by SLM. If the step-sizes are chosen as $\eta, \alpha \sim \mathcal{O}(1/\lambda^r)$, $\gamma = \mathcal{O}(1/L_g)$, $p = \Theta(\lambda^r)$, $\tau = \mathcal{O}(1)$, $\beta = \mathcal{O}(\delta^{1.5})$ when $0 < \delta < 1$, and $\beta = \mathcal{O}(1)$ when $\delta > 1$, and $T^r = \Omega(\log(r\lambda^r))$, $\Lambda \sim \Theta(1/\sqrt{\delta})$, then, $\varrho = \mathcal{O}(\sqrt{\delta})$ and the following results hold*

A1. *Every limit point of $\{x^r, y^r, w^r, z^r, u^r, \lambda^r\}$ generated by SLM is a KKT point of problem* (1).

A2.

$$\frac{1}{T}\sum_{r=1}^{T}\|\mathcal{G}(x^r, y^r)\|^2 = \mathcal{O}\left(\frac{1}{T}\right), \tag{9a}$$

$$\frac{1}{T}\sum_{r=1}^{T}|g(x^r, y^r) - g^*(x^r) - \delta|_+^2 = \mathcal{O}\left(\frac{1}{T}\right) \tag{9b}$$

$$\frac{1}{T}\sum_{r=1}^{T}|(g(x^r, y^r) - g^*(x^r) - \delta)\lambda^r|^2 = \mathcal{O}\left(\frac{1}{T}\right) \tag{9c}$$

*where $|\cdot|_+$ takes the positive part, and $T$ denotes the total number of iterations.*

*Remark 4.* From Theorem 1, we can conclude that the iteration complexity of SLM to reach an $\epsilon$-KKT stationary point is $\mathcal{O}(1/\epsilon)$. It is also worth noting that this convergence rate is optimal in terms of $\epsilon$, as it matches the lower bound established by the first-order method for finding an $\epsilon$-stationary point of nonconvex smooth optimization problems [46].

**Corollary 1.** *(Convergence Rate of SLM to $\epsilon$-KKT Solutions of the BO problem* (2)*) Suppose that A1-A5 hold and the local regularity condition holds. Assume that the iterates $\{x^r, y^r, w^r, z^r, u^r, \lambda^r\}$ are generated by SLM. Given the condition that $\delta = \epsilon$ and assuming $\underline{\delta} = \epsilon$, if the step-sizes $\eta, \alpha = \mathcal{O}(1/\lambda^r)$, $\beta = \mathcal{O}(\epsilon^{1.5})$, $\gamma = \mathcal{O}(1/L_g)$, $\tau = \mathcal{O}(1)$, $T^r = \Omega(\log(1/\epsilon))$ and $p, \Lambda = \Theta(1/\sqrt{\epsilon})$, then $\varrho = \mathcal{O}(\sqrt{\epsilon})$ and the iteration complexity of SLM for finding an $\epsilon$-KKT stationary solution is $\mathcal{O}(1/\epsilon^{3.5})$.*

*Remark 5.* Here, an $\epsilon$-KKT stationary solution of this FCO problem refers to a point $\{x^*, y^*\}$ that satisfies $\|\mathcal{G}(x^*, y^*)\|^2 \leq \epsilon$ and $|g(x^*, y^*) - g^*(x^*) - \delta|_+^2 \leq \epsilon$, and $|(g(x^*, y^*) - g^*(x^*) - \delta)\lambda^*|^2 \leq \epsilon$.

# 4 Proof Sketch

In this section, we will present the main theorem proving techniques employed to establish the results in Theorem 1. Let

$$D(w, z; \lambda) \triangleq \min_{x \in \mathcal{X}, y} K(x, y, w, z; \lambda), \tag{10a}$$

$$P(w, z) \triangleq \min_{x \in \mathcal{X}, y} \max_{0 \leq \lambda \leq \Lambda} K(x, y, w, z; \lambda), \tag{10b}$$

$$x^*(w, z; \lambda), y^*(w, z; \lambda) \triangleq \arg \min_{x \in \mathcal{X}, y} K(x, y, w, z; \lambda), \tag{10c}$$

$$\bar{x}^*(w, z), \bar{y}^*(w, z) \triangleq \arg \min_{x \in \mathcal{X}, y} \max_{0 \leq \lambda \leq \Lambda} K(x, y, w, z; \lambda). \tag{10d}$$

These four quantities are associated with the dual error bounds. The first two quantities, given by (10a) and (10b), define the optimal values of $x, y$ given $w, z$, and $\lambda$, while the last two quantities, defined by (10c) and (10d), represent the optimal solutions for these intermediate subproblems.

It is obvious that when $p$ is sufficiently large, $K(x, y, w, z; \lambda)$ is strongly convex jointly w.r.t. $x$ and $y$. First, under A1 and A5, we can easily show that

$$d^2_{\mathcal{S}(x^r)}(u^r_{T^r}) \leq \mu_g \left(1 - \frac{\gamma}{2\mu_g}\right)^{T^r} (g(x^r, u^r_0) - g^*(x^r)), \tag{11}$$

so under A2 and the assumption that $u^r$ is initialized within a bounded set, it follows that $d^2_{\mathcal{S}(x^r)}(u^r_{T^r})$ decreases to 0 at a linear rate. This inequality (11) is instrumental in quantifying the bias term $|g^*(x^r) - g(x^r, u^{r+1})|^2$ that arises due to the inaccurate estimation of $\arg\min_y g(x, y)$. Next, we will present the following three lemmas which play a crucial role in establishing the proof of Theorem 1.

## 4.1 Descent Lemmas

After one round update of SLM (i.e., from $(x^r, y^r, w^r, z^r, \lambda^r)$ to $(x^{r+1}, y^{r+1}, w^{r+1}, z^{r+1}, \lambda^{r+1})$), we can obtain the following result.

**Lemma 1.** *Under A1-A5, suppose that the sequence is generated by SLM. When $\alpha, \eta \leq \mathcal{O}(1/\lambda^{r+1})$, $0 < \beta < 1$, $T^r = \Omega(\log(r\lambda^{r+1}))$, and $p > L$, there exists a constant $\zeta$ such that*

$$\mathcal{Q}(x^{r+1}, y^{r+1}, w^{r+1}, z^{r+1}; \lambda^{r+1}) - \mathcal{Q}(x^r, y^r, w^r, z^r; \lambda^r)$$

$$\leq -\left(\frac{1}{4\eta} - \frac{17\tau(\ell_g^2 + \ell_g'^2)\sigma_3^2}{\eta^2}\right) \|x^{r+1} - x^r\|^2 - \left(\frac{1}{2\alpha} - \frac{9\tau\ell_g^2\sigma_3^2}{\alpha^2}\right) \|y^{r+1} - y^r\|^2$$

$$- p\left(\frac{1}{2\beta} - \left(\frac{1}{\zeta} + \frac{2p}{p - L} + 36\zeta\sigma_1^2\right)\right) \|z^{r+1} - z^r\|^2 - p\left(\frac{1}{2\beta} - \left(\frac{1}{\zeta} + \frac{2p}{p - L}\right)\right) \|w^{r+1} - w^r\|^2$$

$$+ 36p\zeta \left(\|y^*(w^r, z^r; \lambda_+^{r+1}(w^r, z^r)) - \bar{y}^*(w^r, z^r)\|^2 + \|x^*(w^r, z^r; \lambda_+^{r+1}(w^r, z^r)) - \bar{x}^*(w^r, z^r)\|^2\right)$$

$$- \tau\left(\frac{1}{56} - 48p\zeta\sigma_2^2\tau\right) |g(x^*(w^r, z^r; \lambda^{r+1}), y^*(w^r, z^r; \lambda^{r+1})) - g^*(x^*(w^r, z^r; \lambda^{r+1})) - \delta|^2$$

$$+ \left(\mu_g L_g^2 + 2\ell_g^2\right)\frac{D_{\mathcal{S}}}{r^2} - \frac{1}{4\tau}\|\lambda^{r+1} - \lambda^r\|^2 \tag{12}$$

*where $\sigma_1 \triangleq (p - L)/p$, $\sigma_2 = (p + L)/(p - L)$, $\sigma_3 = 1/(p - L)$, $L = L_f + \lambda(2L_g + L_g^2/(2\mu_g))$, $\lambda_+(w, z) = \mathcal{P}_+[\lambda + \tau\nabla_\lambda K(x^*(w, z; \lambda), y^*(w, z; \lambda), w, z; \lambda)]$, $\ell_g, \ell_g'$ denote Lipschitz constant of $g(x, y)$ and $g^*(x)$ respectively, the potential function is*

$$\mathcal{Q}(x^r, y^r, w^r, z^r; \lambda^r) \triangleq K(x^r, y^r, w^r, z^r; \lambda^r) - 2D(w^r, z^r; \lambda^r) + 2P(w^r, z^r), \tag{13}$$

*and $D_{\mathcal{S}} \triangleq 2\mu_g^{-1}(\max g(x, u_0) - \min g^*(x)) \,\forall x \in \mathcal{X}, u_0 \in \mathcal{B}(R)$.*

It can be observed that the coefficients in front of the terms $\|x^{r+1} - x^r\|^2$, $\|y^{r+1} - y^r\|^2$, $\|z^{r+1} - z^r\|^2$, $\|w^{r+1} - w^r\|^2$ and $|g(x^*(w^r, z^r; \lambda^{r+1}), y^*(w^r, z^r; \lambda^{r+1})) - g^*(x^*(w^r, z^r; \lambda^{r+1})) - \delta|^2$ can be negative when the step-sizes are properly chosen and the bias term can be summed up to a constant. Hence, to ensure a sufficient decrease of $\mathcal{Q}(x^r, y^r, w^r, z^r; \lambda^r)$, it is necessary to bound the positive terms w.r.t. $\|y^*(w^r, z^r; \lambda_+^{r+1}(w^r, z^r)) - \bar{y}^*(w^r, z^r)\|^2$ and $\|x^*(w^r, z^r; \lambda_+^{r+1}(w^r, z^r)) - \bar{x}^*(w^r, z^r)\|^2$. To accomplish this, two novel dual error bounds are provided as follows.

## 4.2 Dual Error Bounds

**Lemma 2.** *(*Weak Dual Error Bound*) Under A1-A3 and A5, suppose that the sequence $\{x^r, y^r, w^r, z^r, u^r, \lambda^r\}$ is generated by SLM. Then, it holds that*

$$\|y^*(w^r, z^r; \lambda_+^{r+1}(w^r, z^r)) - \bar{y}^*(w^r, z^r)\|^2 + \|x^*(w^r, z^r; \lambda_+^{r+1}(w^r, z^r)) - \bar{x}^*(w^r, z^r)\|^2$$
$$\leq \sigma_{weak}\|\lambda^{r+1} - \lambda_+^{r+1}(w^r, z^r)\|\|\lambda(w^r, z^r) - \lambda_+^{r+1}(w^r, z^r)\| \tag{14}$$

*where*

$$\sigma_{weak} \triangleq \frac{1 + \tau(2\ell_g + \ell'_g)\sigma_2}{2\tau(p - L)}, \quad \lambda(w, z) \in \arg \max_{0 \leq \lambda \leq \Lambda} K(\bar{x}^*(w, z), \bar{y}^*(w, z), w, z; \lambda). \tag{15}$$

Although (14) has quantified the variations between $y^*(w^r, z^r; \lambda_+^{r+1}(w^r, z^r))$ and $\bar{y}^*(w^r, z^r)$ and between $x^*(w^r, z^r; \lambda_+^{r+1}(w^r, z^r))$ and $\bar{x}^*(w^r, z^r)$ when the dual variable is perturbed[1], this bound is in a non-homogeneous form. Therefore, it is not sufficient to demonstrate the $\mathcal{O}(1/\epsilon)$ convergence rate of the sequence generated by SLM. Fortunately, under the local regularity condition, we can obtain the following stronger result, which provides further insight.

**Lemma 3.** *(*Strong Dual Error Bound*) Under A1-A3, A5 and the local regularity condition, suppose that the sequence $\{x^r, y^r, w^r, z^r, u^r, \lambda^r\}$ is generated by SLM. Then, there exists a constant $\sigma_{strong}$ such that*

$$\|y^*(w^r, z^r; \lambda_+^{r+1}(w^r, z^r)) - \bar{y}^*(w^r, z^r)\| + \|x^*(w^r, z^r; \lambda_+^{r+1}(w^r, z^r)) - \bar{x}^*(w^r, z^r)\|$$
$$\leq 2\sigma_{strong}\tau|g(x^*(w^r, z^r; \lambda^{r+1}), y^*(w^r, z^r; \lambda^{r+1})) - g^*(x^*(w^r, z^r; \lambda^{r+1})) - \delta| \tag{16}$$

*where*

$$\sigma_{strong} \triangleq \sigma_{weak}\frac{L_f + p + \Lambda L_g}{\sqrt{2\mu_g\underline{\delta}}}. \tag{17}$$

Given this inequality, we can quantify the sufficient descent of the potential function, which further leads to the convergence rate of SLM.

## 5 Numerical Results

In this section, we evaluate our proposed algorithm for solving the bilevel optimization related problems and compare the performance of our proposed SLM with the state-of-the-art methods.

### 5.1 Toy Example

First, we consider the following toy example

$$\min_{x \in [1,2]} f(x, y) \triangleq x^2 + y^2 + 3x\sin^2(y), \quad \text{s.t.} \quad g(x, y) - g^*(x) \leq \delta \tag{18}$$

where the LL objective function is $g(x, y) \triangleq xy^2 + 3x\sin^2(y)$ and $\delta = 1 \times 10^{-3}$. It can be easily checked that function $g(x, y)$ satisfies the PŁ condition w.r.t. variable $y$ and function $f(x, y)$ is nonconvex w.r.t. $x$ and $y$ jointly.

In the numerical results, we initialize variables as $x^1 = y^1 = u^1 = 1$ and choose the step-sizes for updating these variables $(u, y, x)$ as $1 \times 10^{-3}$ for all the compared methods. Additionally, we set $p = 1$ and $\beta = 0.5$ for SLM. It can be seen from Figure 1 that SLM provides the advantage

---

[1]$\lambda_+^{r+1}(w^r, z^r)$ is a perturbation of $\lambda^{r+1}$.

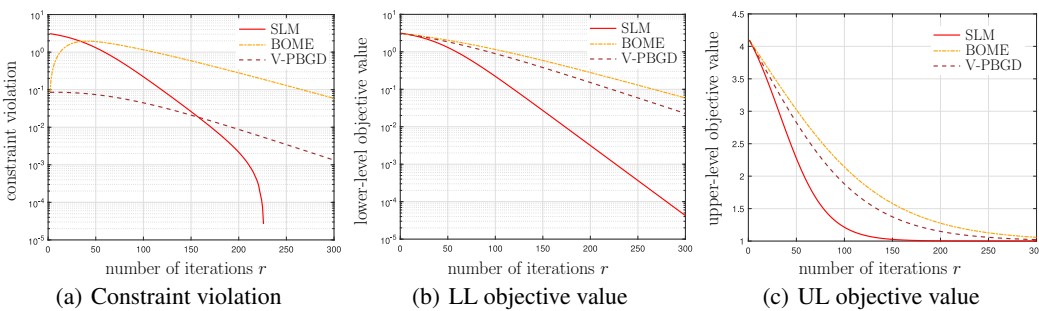

|                | (a) Constraint violation | (b) LL objective value | (c) UL objective value |

Figure 1: Convergence performance of SLM, BOME [15], and V-PBGD [16] on the toy example.

of enforcing the constraint to be satisfied even the LL objective is noncovnex. Moreover, all the algorithms converge to the global optimal solution of the LL problem. It is worth noting that SLM achieves the lowest UL objective value compared to the other methods. This can be attributed to the relaxation introduced by $\delta$ in achieving the optimal solution of the LL problem, which provides more flexibility in searching for lower UL objective values.

## 5.2 Data Hyper-Cleaning

Next, we further evaluate SLM for the data hyper-cleaning problem, which has gained widespread adoption in the machine learning community [3, 16]. The goal of this learning task is to identify and select the clean data samples from the polluted ones, enabling the pre-trained model to generalize effectively on the clean test dataset. The optimization problem is commonly formulated as follows:

$$\min_{x,y} \ell^{\text{val}}(y), \quad \text{s.t.} \quad y \in \arg\min_{y'} \ell^{\text{train}}(x,y') \quad (19)$$

where $\ell^{\text{val}}$ and $\ell^{\text{train}}$ represent the validation (at the UL) and training (at the LL) losses, respectively. The LL loss function is defined as $\ell^{\text{train}} \triangleq \sum_{i=1}^{m} \sigma(x_i)\ell(y)$, where $m$ denotes the total number

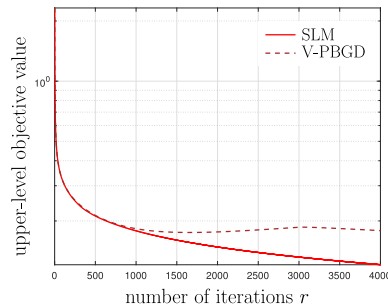

Figure 2: Comparison between SLM and V-PBGD on the data hyper-cleaning task over the MNIST dataset.

of training data samples, $\sigma(x)$ is the sigmoid function, and $y$ denotes the weights of the neural network. Adhering to the experimental settings used to test the V-PBGD as shown in [16], we are given $5,000$ training data samples from the MNIST dataset. Among these samples, $50\%$ are randomly polluted with incorrect labels. Additionally, we have a validation dataset consisting of $5,000$ samples and a separate test set comprising $10,000$ samples. The model parameter $y$ includes the neural network weights of a hidden layer with a size of $10 \times 784$ and bias terms with a size of $10$.

It has been shown in [16] that V-PBGD outperforms significantly the state-of-the-art methods, including BOME [15], IAPTT-GM [47], RHG [3], T-RHG [4], for this particular example. We adapt the code used to implement V-PBGD to evaluate the performance of SLM and set $\tau = p = 20$, $\beta = 0.5$, $\delta = 1.2$ for SLM. From Figure 2, it can be observed that SLM initially converges at a similar rate to V-PBGD in terms of the test loss values. However, after a certain number of iterations, SLM achieves smaller loss values. The reason behind this observation is that once the constraint is satisfied, the optimization variables primarily focus on minimizing the UL objective values, leading to lower test losses in this example. Although SLM achieves a test accuracy similar to V-PBGD, with the highest test accuracy of $90.44\%$, when the number of iterations becomes large (e.g., exceeding $3,000$ in the figure), the model begins to overfit the training data. However, SLM maintains a test accuracy of $88.8\%$, while V-PBGD can only attain $88.1\%$.

As for the constraint violation, we define $c_{\text{error}} \triangleq \ell^{\text{train}}(x,y) - v(x)$, where $v(x)$ is computed through the updating process of variable $u$. This measure represents the distance between the LL objective value and the optimal value. V-PBGD results in a $c_{\text{error}}$ value of approximately $0.68$, whereas SLM achieves a $(c_{\text{error}} - \delta)$ value of $0.001$ when the total number of iterations exceeds $2,000$ in the figure. This further reinforces the fact that due to the presence of the optimization variables appearing in both levels or the coupling between the LL and UL variables, it is highly challenging for $y$ in the BO formulation to find the global optimal solution of the LL problem. In contrast, SLM can find KKT solutions for this class of structured FCO problems as long as the hyper-parameter $\delta$ is appropriately defined.

## 5.3 Neural Network on MNIST Data Set

We also evaluate the performance of these algorithms on a 2-layer neural network. The size of the first layer is $784 \times 300$, and the size of the second layer is $300 \times 10$. The activation function used is the sigmoid function. For all the compared algorithms, we set the step-sizes of the block-$x$ and block-$y$ updates to $1$. The step-size $\gamma$ for the auxiliary block-$u$ update is set to $0.1$. Additionally, we choose $\tau = 0.01$, $\beta = 0.5$, and $p = 20$ for SLM, while for BOME, we use $\eta = 0.1$. The remaining settings remain the same as mentioned in the main text for the linear case.

The results, averaged over $5$ independent trials, are presented in Figure 3. It can be observed from Figure 3(a) that SLM with $\delta = 1$ achieves the lowest test loss and converges faster compared to

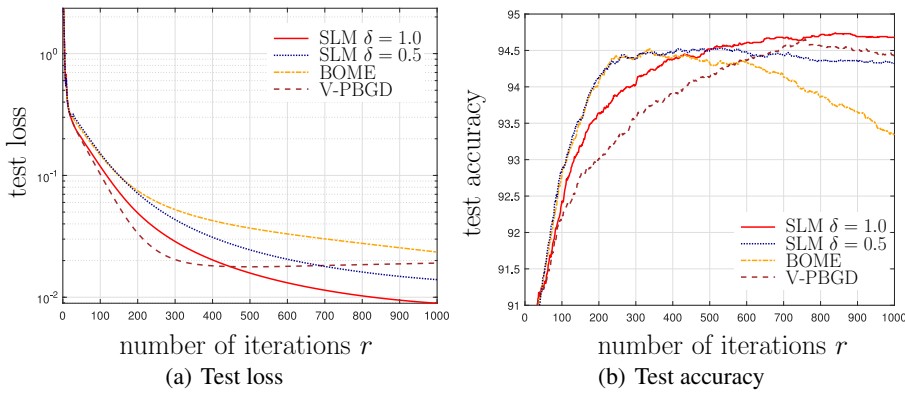

| | |
|:---:|:---:|
| (a) Test loss | (b) Test accuracy |

Figure 3: Convergence performance of SLM, BOME [15], and V-PBGD [16] on the neural network.

BOME. When $\delta = 0.5$, the convergence rate of SLM becomes slower. Both of these observations align with the theoretical convergence analysis. The test loss obtained by V-PBGD initially decreases and then increases, which is reasonable as this method optimizes both the UL (validation) and LL (training) loss values, with the penalty parameter increasing as the number of iterations rises. In contrast, SLM primarily minimizes the objective loss values, specifically the validation loss in this case, leading to better generalization performance as long as the constraint is satisfied. The constraint tolerance parameter $\delta$ and the dual variable implicitly play the tradeoff between the training and validation losses.

Regarding the test accuracy shown in Figure 3(b), SLM with $\delta = 1$ achieves a minimum of $94.74 \pm 0.04\%$, which is the lowest among all the algorithms compared. V-PBGD can reach $94.65 \pm 0.09\%$ but converges slower than SLM with $\delta = 1$. SLM with $\delta = 0.5$ and BOME converge relatively quickly initially and reach peak accuracies of $94.53 \pm 0.06\%$ and $94.52 \pm 0.11\%$, respectively. However, it is evident that the test accuracy achieved by BOME decreases rapidly as the algorithm proceeds, even though the test loss continues to decrease, forming a clear U-shape. This implies that BOME overfits the validation data when the number of iterations is large. In contrast, the test accuracy achieved by SLM remains relatively stable, or at least does not decline as rapidly as that attained by BOME. This further confirms that the use of structured constraints in this class aids in improving the generalization performance of bilevel models.

## 6  Concluding Remarks

In this work, we focused on solving a class of structured nonconvex FCO problems, where the functional constraint satisfies the PŁ condition w.r.t. one block of variables. In practical applications, this class of FCO problems can effectively model a wide range of nested or hierarchical learning problems, including data hyper-cleaning, meta-learning, corset selection, and more. To address these challenges, we developed a smoothed primal-dual algorithm based on the Lagrangian method. The proposed algorithm achieves a convergence rate of $\mathcal{O}(\epsilon^{-1})$ to the $\epsilon$-KKT solutions of this problem by utilizing only the first-order oracles for both the objective function and the constraint.

The major difficulty from a theoretical perspective lies in showing the dominance of the descent achieved by the algorithm over the ascent induced by enforcing the nonconvex functional constraint. To the best of our knowledge, this is the first result showcasing that the dual error bound also holds for solving this class of nonlinear constraints. We conducted extensive numerical experiments to evaluate the performance of the proposed SLM against benchmark nonconvex BO methods. The results demonstrate that SLM not only ensures the satisfaction of the functional constraint but also achieves small objective values.

It is important to note that this work focuses on the scenario with a single constraint. Future research endeavors will explore the extension of the algorithm design and theoretical results to more general settings. Addressing this limitation will contribute to a broader applicability of the proposed method.

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

# Supplementary Material

## A Preliminaries

In this section, we provide some technical preliminaries for the proofs of the lemmas and theorems claimed in the main body of this paper and the supplementary material, including parameter definitions and supporting results.

### A.1 Inequalities

Basic inequalities used in the proof include

1. Strong Convexity: if function $f(x)$ is strongly convex with modulus $\mu$, then

$$f(x) - f(x') \geq \frac{\mu}{2}\|x - x'\|^2. \tag{20}$$

2. Young's inequality with parameter $\theta > 0$:

$$\langle x, y \rangle \leq \frac{1}{2\theta}\|x\|^2 + \frac{\theta}{2}\|y\|^2, \quad \forall x, y. \tag{21}$$

3. Triangle inequality and reverse triangle inequality:

$$\|x + y\| \leq \|x\| + \|y\|, \quad \|x - y\| \geq \|x\| - \|y\|. \tag{22}$$

4. Cauchy-Schwarz inequality

$$\langle x, y \rangle \leq \|x\|\|y\|. \tag{23}$$

### A.2 Notations

The definitions of the parameters and assumptions are further listed in Table 2.

Note that under A2 (i.e., compactness of the feasible set $\mathcal{X}$) the Lipschitz continuity of $f(x, y), g(x, y)$ w.r.t. $x$ holds. In Section C.3, we will demonstrate that the iterates (such as $y^r, \bar{y}^*(w^r, z^r)$) for which we need to evaluate the gradients or function values of $g(x, y)$, $g^*(x)$, and $f(x, y)$ are bounded, which implies that the corresponding Lipschitz continuity holds. To be more precise, $\ell_f, L_g, \ell_g, \ell'_g$ and others are listed in the section of notations in Table 2.

### A.3 Primal Error Bounds

Recall that the definition of $K(x, y, w, z; \lambda)$ is

$$K(x, y, w, z; \lambda) \triangleq f(x, y) + \lambda(g(x, y) - g^*(x) - \delta) + \frac{p}{2}\|x - w\|^2 + \frac{p}{2}\|y - z\|^2. \tag{24}$$

Based on the assumptions listed in Table 2, we have that $f, g$ are gradient Lipschitz continuous. Further, it has been shown in [34, Lemma A5] that $\nabla g^*(x) = \nabla_x g(x, y^*)$ for any $y^* \in \mathcal{S}(x)$ and $g^*(x)$ is Lipschitz smooth with modulus $L_g + L_g^2/(2\mu_g)$. Therefore, the Lagrangian $\mathcal{L}(x, y; \lambda)$ is gradient Lipschitz continuous with parameter

$$L \triangleq L_f + \lambda\left(2L_g + \frac{L_g^2}{2\mu_g}\right). \tag{25}$$

Subsequently, function $K(x, y, w, z; \lambda)$ is strongly convex of $x$ and $y$ with parameter $p - L$ and gradient Lipschitz continuous with parameter

$$L_K \triangleq L + p. \tag{26}$$

Given these property and the assumptions listed in Table 2, we can obtain the following primal error bounds that have been studied in [33, Lemma 5], [32, Lemma 3.5 and Lemma 3.10] and [36, Lemma

Table 2: Summary of Definitions. ("Lips.": Lipschitz; "grad.": gradient; "const.": constant; "opt.": optimal; "$\cdot$" represents the gradient is taken w.r.t. either $x$ or $y$; "$[x; y]$" symbolizes the concatenation of $x$ and $y$; PEB: primal error bound; DEB: dual error bound.)

| **A1** | Definition | Annotation |
|---|---|---|
| $L_f$ | $\|\nabla.f(x,y) - \nabla.f(x',y)\| \leq L_f\|x - x'\|$ | grad. Lips. const. of $f(x,y)$ w.r.t $x$ |
| | $\|\nabla.f(x,y) - \nabla.f(x,y')\| \leq L_f\|y - y'\|$ | grad. Lips. const. of $f(x,y)$ w.r.t $y$ |
| $L_g$ | $\|\nabla_y g(x,y) - \nabla_y g(x,y')\| \leq L_g\|y - y'\|$ | grad. Lips. const. of $g$ w.r.t. $y$ |
| **A2** | Definition | Annotation |
| $\mathcal{X}$ | closed and bounded | imply (grad.) Lips. of $f, g$ w.r.t. $x$ |
| **A5** | Definition | Annotation |
| $\mu_g$ | $\|\nabla_y g(x,y)\|^2 \geq 2\mu_g(g(x,y) - g^*(x))$ | PŁ condition |
| Parameter | Definition | Annotation |
| $d_{\mathcal{S}(x)}(y)$ | $\min_{y' \in \mathcal{S}(x)}\|y - y'\|$ | min distance from $y$ to $\mathcal{S}(x)$ |
| $D_{\mathcal{S}}$ | $\max_u d^2_{\mathcal{S}(x)}(u)$ (cf. (33)) | upper bound of $d^2_{\mathcal{S}(x)}(u)$ |
| $D(w,z;\lambda)$ | $\min_{x,y} K(x,y,w,z;\lambda)$ | |
| $x^*(w,z;\lambda)$ | $\arg\min_{x,y} K(x,y,w,z;\lambda)$ | opt. solution of $D(w,z;\lambda)$ w.r.t. $x$ |
| $y^*(w,z;\lambda)$ | $\arg\min_{x,y} K(x,y,w,z;\lambda)$ | opt. solution of $D(w,z;\lambda)$ w.r.t. $y$ |
| $P(w,z)$ | $\min_{x,y}\max_{0\leq\lambda\leq\Lambda} K(x,y,w,z;\lambda)$ | |
| $\bar{x}^*(w,z)$ | $\arg\min_{x,y}\max_{0\leq\lambda\leq\Lambda} K(x,y,w,z;\lambda)$ | opt. solution of $P(w,z)$ w.r.t. $x$ |
| $\bar{y}^*(w,z)$ | $\arg\min_{x,y}\max_{0\leq\lambda\leq\Lambda} K(x,y,w,z;\lambda)$ | opt. solution of $P(w,z)$ w.r.t. $y$ |
| $v$ | $v \triangleq (w,z)$ | abbreviation of $w, z$ |
| Notation | Definition | Annotation |
| $\ell_f$ | $\|f(x,y) - f(x,y')\| \leq \ell_f\|y - y'\|$ | Lips. const. of $f(x,y)$ w.r.t. $y$ |
| $L_g$ | $\|\nabla.g(x,y) - \nabla.g(x',y)\| \leq L_g\|x - x'\|$ | grad. Lips. const. of $g(x,y)$ |
| $\ell_g$ | $\|g(x,y) - g(x,y')\| \leq \ell_g\|y - y'\|$ | Lips. const. of $g(x,y)$ w.r.t. $y$ |
| | $\|g(x,y) - g(x',y)\| \leq \ell_g\|x - x'\|$ | Lips. const. of $g(x,y)$ w.r.t. $x$ |
| $\ell'_g$ | $\|g^*(x) - g^*(x')\| \leq \ell'_g\|x - x'\|$ (cf. [34]) | Lips. const. of $g^*(x)$ w.r.t. $x$ |
| $L$ | $\|\nabla\mathcal{L}(x,y;\lambda) - \nabla\mathcal{L}(x',y';\lambda)\|$ $\leq L\|[x;y] - [x';y']\|$ (cf. (25)) | grad. Lips. const. of $\mathcal{L}()$ w.r.t. $x, y$ |
| $L_K$ | $L + p$ (cf. (26)) | grad. Lips. const. of $K()$ w.r.t. $x, y$ |
| $\sigma_1$ | $p(p - L)^{-1}$ (cf. (27a)) | const. of PEB w.r.t. $v$ |
| $\sigma_2$ | $(p + L)(p - L)^{-1}$ (cf. (28a)) | const. of PEB w.r.t. $\lambda$ |
| $\sigma_3$ | $(p - L)^{-1}$ (cf. (29a)) | const. of PEB w.r.t. $x^r$ or $y^r$ |
| $\sigma_{\text{weak}}$ | $(1 + \tau(2\ell_g + \ell'_g)\sigma_2)(\tau(p - L))^{-1}$ (cf. (80)) | const. of weak DEB |
| $\sigma_{\text{aux}}$ | $(L_f + p + \Lambda L_g)/\sqrt{2\mu_g\underline{\delta}}$ (cf. (94)) | const. of auxiliary DEB |
| $\sigma_{\text{strong}}$ | $\sigma_{\text{weak}}\sigma_{\text{aux}}$ (cf. (100)) | const. of strong DEB |

B.2]. To be more specific, from [36, Lemma B.2] we have

$$\|y^*(w,z;\lambda) - y^*(w,z';\lambda)\| \leq \underbrace{\frac{p}{p - L}}_{\triangleq \sigma_1}\|z - z'\|, \tag{27a}$$

$$\|x^*(w,z;\lambda) - x^*(w',z;\lambda)\| \leq \sigma_1\|w - w'\|, \tag{27b}$$

$$\|\bar{y}^*(w,z) - \bar{y}^*(w,z')\| \leq \sigma_1\|z - z'\|, \tag{27c}$$

$$\|\bar{x}^*(w,z) - \bar{x}^*(w',z)\| \leq \sigma_1\|w - w'\|. \tag{27d}$$

Similarly, following [36, Lemma B.2], we can also have

$$\|y^*(w, z; \lambda) - y^*(w, z; \lambda')\| \leq \underbrace{\frac{p + L}{p - L}}_{\triangleq \sigma_2} \|\lambda - \lambda'\|, \tag{28a}$$

$$\|x^*(w, z; \lambda) - x^*(w, z; \lambda')\| \leq \sigma_2 \|\lambda - \lambda'\|, \tag{28b}$$

and from [32, Lemma 3.10] or [33, Lemma 5] we can directly get

$$\|y^*(w^r, z^r; \lambda^{r+1}) - y^r\| \leq \frac{1}{\alpha} \underbrace{\frac{1}{p - L}}_{\triangleq \sigma_3} \|y^{r+1} - y^r\|, \tag{29a}$$

$$\|x^*(w^r, z^r; \lambda^{r+1}) - x^r\| \leq \frac{\sigma_3}{\eta} \|x^{r+1} - x^r\|, \tag{29b}$$

$$\|y^*(w^r, z^r; \lambda^{r+1}) - y^{r+1}\| \leq \frac{1 + \alpha(p - L)}{\alpha(p - L)} \|y^{r+1} - y^r\|, \tag{29c}$$

$$\|x^*(w^r, z^r; \lambda^{r+1}) - x^{r+1}\| \leq \frac{1 + \eta(p - L)}{\eta(p - L)} \|x^{r+1} - x^r\|, \tag{29d}$$

where the primal error bounds (29b) and (29d) hold as $\widehat{K}$ (which is used for updating $x$) is also $(p - L)$-strongly convex and $(p + L)$-Lipschitz smooth.

### A.4   Outline of the Proof

We will begin by presenting the contraction property of the block-$u$ update in Lemma 4. Next, we will show the primal descent in Lemma 5, dual ascent in Lemma 6, and proximal descent in Lemma 7 by following the update rule of SLM, which will be used to construct the potential function. However, these lemmas alone are insufficient to establish the decrease of the potential function to achieve the $\mathcal{O}(1/\epsilon)$ convergence rate, as mentioned in the main text. Hence, we prove the strong dual error bound Lemma 3 by combining the weak dual error bound presented in Lemma 2 and the key auxiliary lemma in Lemma 8, which ensures the homogeneity of the dual error bound.

With the strong dual error bound established, we can demonstrate the sufficient decrease of the potential function in Lemma 9. This, in turn, leads to the final convergence results shown in Section Section D. Throughout the process of proving the lemmas and theorems, we initially assume the boundedness of the dual variable, as well as $y^r, \bar{y}^*(w^r, z^r), y^*(w^r, z^r; \lambda^{r+1})$. We then verify these assumptions through mathematical induction, which are shown in Lemma 11 and Lemma 12.

## B   Convergence Analysis

We now present the proofs, related results, and technical details that establish the lemmas and theorems of our convergence analysis.

We first show that the contraction property of $u$-update for the value function.

### B.1   Contraction of Auxiliary Variable $u$

**Lemma 4.**

*Under A1, A2, and A5. Assume that the sequence $\{x^r, y^r, w^r, z^r, u^r, \lambda^r, \forall r\}$ is generated by SLM. When*

$$\gamma \leq \min \left\{ \frac{1}{L_g}, \frac{1}{2\mu_g} \right\}, \tag{30}$$

*we have*

$$d_{\mathcal{S}(x^r)}^2(u^{r+1}) \leq \frac{2}{\mu_g} \left(1 - 2\gamma\mu_g\right)^{T^r} \left(g(x^r, u_0^r) - g^*(x^r)\right). \tag{31}$$

*Further, we can obtain*

$$d_{\mathcal{S}(x^r)}^2(u^{r+1}) \leq \left(1 - 2\gamma\mu_g\right)^{T^r} D_{\mathcal{S}} \quad \forall r \tag{32}$$

*where*

$$D_{\mathcal{S}} \triangleq 2\mu_g^{-1} \left(\max g(x, u_0) - \min g^*(x)\right) \forall x \in \mathcal{X}, u_0 \in \mathcal{B}(R). \tag{33}$$

*Proof.* According to the gradient Lipschitz continuity of function $g(x, y)$ w.r.t. $y$ (cf. A1), we have

$$g(x^r, u_{t+1}^r) \leq g(x^r, u_t^r) - \gamma \langle \nabla_u g(x^r, u_t^r), u_{t+1}^r - u_t^r \rangle + \frac{L_g \gamma^2}{2} \|\nabla_u g(x^r, u_t^r)\|^2 \tag{34}$$

$$\overset{(a)}{\leq} g(x^r, u_t^r) - \gamma \|\nabla_u g(x, u_t^r)\|^2 + \frac{L_g \gamma^2}{2} \|\nabla_u g(x^r, u_t^r)\|^2 \tag{35}$$

$$\overset{(b}{\leq} g(x^r, u_t^r) - \frac{\gamma}{2} \|\nabla_u g(x, u_t^r)\|^2 \tag{36}$$

where in $(a)$ we apply (5), $(b)$ holds when $\gamma \leq 1/L_g$.

Under A5, we have

$$g(x^r, u_{t+1}^r) - g^*(x^r) \leq g(x^r, u_t^r) - g^*(x^r) - 2\gamma \mu_g \left( g(x^r, u_t^r) - g^*(x^r) \right) \tag{37}$$

$$\leq (1 - 2\gamma \mu_g) \left( g(x^r, u_t^r) - g^*(x^r) \right). \tag{38}$$

From [35, Theorem 2], we know that the PŁ condition implies the error bound with the same constant, i.e., $d_{\mathcal{S}(x^r)}^2(u_t^r) \leq 2/\mu_g(g(x^r, u_t^r) - g^*(x^r))$, which yields

$$d_{\mathcal{S}(x^r)}^2(u^{r+1}) \overset{(a)}{=} d_{\mathcal{S}(x^r)}^2(u_{T^r}^r)$$

$$\leq \frac{2}{\mu_g} \left( g(x^r, u_{t+1}^r) - g^*(x^r) \right) \tag{39}$$

$$\leq \frac{2}{\mu_g} (1 - 2\gamma \mu_g)^{T^r} \left( g(x^r, u_0^r) - g^*(x^r) \right) \tag{40}$$

where $(a)$ holds due to the line 7 of Algorithm 1. Combining the facts that $\mathcal{X}$ is compact and $u^r$ is initialized within a ball gives this lemma directly. $\qquad \square$

## B.2  Descent Lemmas and Dual Ascent

### B.2.1  Primal Descent Lemma

**Lemma 5.**

*(Primal Descent) Under A1-A5, suppose that the sequence $\{x^r, y^r, w^r, z^r, u^r, \lambda^r, \forall r\}$ is generated by SLM. When*

$$0 < \eta, \alpha \leq \frac{1}{L_K}, \quad 0 < \beta \leq 1, \quad \text{and } T^r \geq -2 \log_{(1-2\gamma\mu_g)} (\lambda^{r+1} \sqrt{\eta} r), \tag{41}$$

*then the primal descent inequality holds, namely,*

$$K(x^{r+1}, y^{r+1}, w^{r+1}, z^{r+1}; \lambda^{r+1}) - K(x^r, y^r, w^r, z^r; \lambda^r)$$

$$\leq -\frac{1}{4\eta} \|x^{r+1} - x^r\|^2 + \langle g(x^r, y^r) - g^*(x^r) - \delta, \lambda^{r+1} - \lambda^r \rangle$$

$$- \frac{1}{2\alpha} \|y^{r+1} - y^r\|^2 - \frac{p}{2\beta} \|z^{r+1} - z^r\|^2 - \frac{p}{2\beta} \|w^{r+1} - w^r\|^2 + \frac{\mu_g L_g^2 D_\mathcal{S}}{r^2}. \tag{42}$$

*Proof. $x$-update:* First, recall

$$\nabla_x \widehat{K}(x^r, y^{r+1}, w^r, z^{r+1}, u^{r+1}; \lambda^{r+1})$$

$$\triangleq \nabla_x f(x^r, y^{r+1}) + \lambda^{r+1} \left( \nabla_x g(x^r, y^{r+1}) - \nabla_x g(x^r, u^{r+1}) \right) + p(x^r - w^r). \tag{43}$$

From (8a), one step of the projected gradient descent step gives

$$K(x^{r+1}, y^{r+1}, w^r, z^{r+1}; \lambda^{r+1}) - K(x^r, y^{r+1}, w^r, z^{r+1}; \lambda^{r+1})$$

$$\overset{(a)}{\leq} \langle \nabla K(x^r, y^{r+1}, w^r, z^{r+1}; \lambda^{r+1}), x^{r+1} - x^r \rangle + \frac{L_K}{2} \|x^{r+1} - x^r\|^2 \tag{44}$$

$$\leq \langle \nabla \widehat{K}(x^r, y^{r+1}, w^r, z^{r+1}, u^{r+1}; \lambda^{r+1}), x^{r+1} - x^r \rangle + \frac{L_K}{2} \|x^{r+1} - x^r\|^2$$

$$+ \langle \nabla K(x^r, y^{r+1}, w^r, z^{r+1}; \lambda^{r+1}) - \nabla \widehat{K}(x^r, y^{r+1}, w^r, z^{r+1}, u^{r+1}; \lambda^{r+1}), x^{r+1} - x^r \rangle \tag{45}$$

where $(a)$ is because gradient Lipschitz continuity of $K(x, y, w, z; \lambda)$ is with parameter $L_K$.

From the optimality condition of (8a), we have

$$\langle \nabla_x \widehat{K}(x^r, y^{r+1}, w^r, z^{r+1}, u^{r+1}; \lambda^{r+1}), x^{r+1} - x^r \rangle \leq -\frac{1}{\eta} \|x^{r+1} - x^r\|^2. \tag{46}$$

Regarding the last term at the right hand side (RHS) of (45), we have

$$\langle \nabla K(x^r, y^{r+1}, w^r, z^{r+1}; \lambda^{r+1}) - \nabla \widehat{K}(x^r, y^{r+1}, w^r, z^{r+1}, u^{r+1}; \lambda^{r+1}), x^{r+1} - x^r \rangle$$

$$\leq \eta \|\nabla K(x^r, y^{r+1}, w^r, z^{r+1}; \lambda^{r+1}) - \nabla \widehat{K}(x^r, y^{r+1}, w^r, z^{r+1}, u^{r+1}; \lambda^{r+1})\|^2 + \frac{1}{4\eta} \|x^{r+1} - x^r\|^2$$

where we apply Young's inequality with parameter 2.

For the first term at the RHS of the above inequality, we can further have

$$\|\nabla K(x^r, y^{r+1}, w^r, z^{r+1}; \lambda^{r+1}) - \nabla \widehat{K}(x^r, y^{r+1}, w^r, z^{r+1}, u^{r+1}; \lambda^{r+1})\|^2$$

$$\leq (\lambda^{r+1})^2 \|\nabla g^*(x^r) - \nabla g(x^r, u^{r+1})\|^2 \tag{47}$$

$$\leq (\lambda^{r+1})^2 L_g^2 d_{\mathcal{S}(x^r)}^2 (u^{r+1}) \tag{48}$$

$$\overset{(32)}{\leq} (\lambda^{r+1})^2 L_g^2 \mu_g (1 - 2\gamma\mu_g)^{T^r} D_{\mathcal{S}} \tag{49}$$

$$\overset{(a)}{\leq} \frac{\mu_g L_g^2 D_{\mathcal{S}}}{\eta r^2} \tag{50}$$

where in $(a)$ we take $T^r \geq -2 \log_{(1-2\gamma\mu_g)}(\lambda^{r+1}\sqrt{\eta}r)$.

Substituting (46) and (50) back to (45) gives

$$K(x^{r+1}, y^{r+1}, z^{r+1}; \lambda^{r+1}) - K(x^r, y^{r+1}, z^{r+1}; \lambda^{r+1}) \leq -\frac{1}{4\eta} \|x^{r+1} - x^r\|^2 + \frac{\mu_g L_g^2 D_{\mathcal{S}}}{r^2} \tag{51}$$

where we select $\eta \leq 1/L_K$.

*w-update:* From (6), we have

$$K(x^{r+1}, y^{r+1}, w^{r+1}, z^{r+1}; \lambda^{r+1}) - K(x^{r+1}, y^{r+1}, w^r, z^{r+1}; \lambda^{r+1})$$

$$= \frac{p}{2} \left( \|x^{r+1} - w^{r+1}\|^2 - \|x^{r+1} - w^r\|^2 \right) \tag{52}$$

$$= \frac{p}{2} \langle w^r - w^{r+1}, x^{r+1} - w^{r+1} + x^{r+1} - w^r \rangle \tag{53}$$

$$\overset{(a)}{\leq} -\frac{p}{2\beta} \|w^{r+1} - w^r\|^2 \tag{54}$$

where $(a)$ holds due to

$$\langle w^r - w^{r+1}, x^{r+1} - w^{r+1} + x^{r+1} - w^r \rangle$$

$$= \langle w^r - w^{r+1}, x^{r+1} - w^r + w^r - w^{r+1} + x^{r+1} - w^r \rangle \tag{55}$$

$$= \left(1 - \frac{2}{\beta}\right) \|w^{r+1} - w^r\|^2 \tag{56}$$

and (8b) for $\beta \leq 1$.

*λ-update:* According to the dual variable update (7a), we have

$$K(x^r, y^r, w^r, z^r; \lambda^{r+1}) - K(x^r, y^r, w^r, z^r; \lambda^r) = \langle g(x^r, y^r) - g^*(x^r) - \delta, \lambda^{r+1} - \lambda^r \rangle. \tag{57}$$

*y-update:* The update of $y$ shown in (7b) is the standard gradient descent, which gives

$$K(x^r, y^{r+1}, w^r, z^r; \lambda^{r+1}) - K(x^r, y^r, w^r, z^r; \lambda^{r+1})$$

$$\leq \langle \nabla K(x^r, y^r, w^r, z^r; \lambda^{r+1}), y^{r+1} - y^r \rangle + \frac{L_K}{2} \|y^{r+1} - y^r\|^2 \tag{58}$$

$$\leq -\frac{1}{2\alpha} \|y^{r+1} - y^r\|^2 \tag{59}$$

where $\alpha \leq 1/L_K$.

*z-update:* Similar to the $w$-update. From (7c), we have

$$K(x^r, y^{r+1}, w^r, z^{r+1}; \lambda^{r+1}) - K(x^r, y^{r+1}, w^r, z^r; \lambda^{r+1}) \leq -\frac{p}{2\beta}\|z^{r+1} - z^r\|^2. \tag{60}$$

$\square$

### B.2.2 Dual Ascent Lemma

**Lemma 6.**

*(Dual Ascent) Under A1-A5, suppose that the sequence $\{x^r, y^r, w^r, z^r, u^r, \lambda^r, \forall r\}$ is generated by SLM. When $p > L$, the dual ascent inequality holds, namely,*

$$D(w^{r+1}, z^{r+1}; \lambda^{r+1}) - D(w^r, z^r; \lambda^r)$$
$$\geq \langle \lambda^{r+1} - \lambda^r, g(x^*(w^r, z^r; \lambda^{r+1}), y^*(w^r, z^r; \lambda^{r+1})) - g^*(x^*(w^r, z^r; \lambda^{r+1}) - \delta) \rangle$$
$$+ \frac{p}{2}\langle z^{r+1} - z^r, z^{r+1} + z^r - 2y^*(w^r, z^{r+1}; \lambda^{r+1}) \rangle$$
$$+ \frac{p}{2}\langle w^{r+1} - w^r, w^{r+1} + w^r - 2x^*(w^{r+1}, z^{r+1}; \lambda^{r+1}) \rangle. \tag{61}$$

*Proof.* Recall

$$K(x, y, w, z; \lambda) \triangleq \mathcal{L}(x, y; \lambda) + \frac{p}{2}\|x - w\|^2 + \frac{p}{2}\|y - z\|^2. \tag{62}$$

Then, we can have

$$D(w^r, z^r; \lambda^{r+1}) - D(w^r, z^r; \lambda^r)$$
$$= K(x^*(w^r, z^r; \lambda^{r+1}), y^*(w^r, z^r; \lambda^{r+1}), w^r, z^r; \lambda^{r+1})$$
$$\quad - K(x^*(w^r, z^r; \lambda^r), y^*(w^r, z^r; \lambda^r), w^r, z^r; \lambda^r) \tag{63}$$
$$\overset{(a)}{\geq} K(x^*(w^r, z^r; \lambda^{r+1}), y^*(w^r, z^r; \lambda^{r+1}), w^r, z^r; \lambda^{r+1})$$
$$\quad - K(x^*(w^r, z^r; \lambda^{r+1}), y^*(w^r, z^r; \lambda^{r+1}), w^r, z^r; \lambda^r) \tag{64}$$
$$= \langle \lambda^{r+1} - \lambda^r, g(x^*(w^r, z^r; \lambda^{r+1}), y^*(w^r, z^r; \lambda^{r+1})) - g^*(x^*(w^r, z^r; \lambda^{r+1}) - \delta) \rangle \tag{65}$$

where in $(a)$ we use the definition of $y^*(w^r, z^r; \lambda^{r+1})$ for $p > L$.

Similarly, we have

$$D(w^r, z^{r+1}; \lambda^{r+1}) - D(w^r, z^r; \lambda^{r+1})$$
$$= K(x^*(w^r, z^{r+1}; \lambda^{r+1}), y^*(w^r, z^{r+1}; \lambda^{r+1}), w^r, z^{r+1}; \lambda^{r+1})$$
$$\quad - K(x^*(w^r, z^r; \lambda^{r+1}), y^*(w^r, z^r; \lambda^{r+1}), w^r, z^r; \lambda^{r+1}) \tag{66}$$
$$\geq K(x^*(w^r, z^{r+1}; \lambda^{r+1}), y^*(w^r, z^{r+1}; \lambda^{r+1}), w^r, z^{r+1}; \lambda^{r+1})$$
$$\quad - K(x^*(w^r, z^{r+1}; \lambda^{r+1}), y^*(w^r, z^{r+1}; \lambda^{r+1}), w^r, z^r; \lambda^{r+1}) \tag{67}$$
$$= \frac{p}{2}\left(\|y^*(w^r, z^{r+1}; \lambda^{r+1}) - z^{r+1}\|^2 - \|y^*(w^r, z^{r+1}; \lambda^{r+1}) - z^r\|^2\right) \tag{68}$$
$$= \frac{p}{2}\langle z^{r+1} - z^r, z^{r+1} + z^r - 2y^*(w^r, z^{r+1}; \lambda^{r+1}) \rangle, \tag{69}$$

and

$$D(w^{r+1}, z^{r+1}; \lambda^{r+1}) - D(w^r, z^{r+1}; \lambda^{r+1})$$
$$= K(x^*(w^{r+1}, z^{r+1}; \lambda^{r+1}), y^*(w^{r+1}, z^{r+1}; \lambda^{r+1}), w^{r+1}, z^{r+1}; \lambda^{r+1})$$
$$\quad - K(x^*(w^r, z^{r+1}; \lambda^{r+1}), y^*(w^r, z^{r+1}; \lambda^{r+1}), w^r, z^{r+1}; \lambda^{r+1}) \tag{70}$$
$$\geq K(x^*(w^{r+1}, z^{r+1}; \lambda^{r+1}), y^*(w^{r+1}, z^{r+1}; \lambda^{r+1}), w^{r+1}, z^{r+1}; \lambda^{r+1})$$
$$\quad - K(x^*(w^{r+1}, z^{r+1}; \lambda^{r+1}), y^*(w^{r+1}, z^{r+1}; \lambda^{r+1}), w^r, z^{r+1}; \lambda^{r+1}) \tag{71}$$
$$= \frac{p}{2}\left(\|x^*(w^{r+1}, z^{r+1}; \lambda^{r+1}) - w^{r+1}\|^2 - \|x^*(w^{r+1}, z^{r+1}; \lambda^{r+1}) - w^r\|^2\right) \tag{72}$$
$$= \frac{p}{2}\langle w^{r+1} - w^r, w^{r+1} + w^r - 2x^*(w^{r+1}, z^{r+1}; \lambda^{r+1}) \rangle, \tag{73}$$

which give the desired result. □

### B.2.3 Proximal Descent Lemma

**Lemma 7.**

*(Proximal Descent) Under A1-A5, suppose that the sequence $\{x^r, y^r, w^r, z^r, u^r, \lambda^r, \forall r\}$ is generated by SLM. Assume that $y^r$ is bounded and $p > L$, then the proximal descent inequality holds, namely,*

$$
\begin{aligned}
&P(w^{r+1}, z^{r+1}) - P(w^r, z^r) \\
&\leq p(z^{r+1} - z^r)^T (z^r - \bar{y}^*(w^r, z^r)) + p(w^{r+1} - w^r)^T (w^r - \bar{x}^*(w^r, z^{r+1})) \\
&\quad + \frac{p}{2}\left(\frac{p}{p-L} + 1\right)\left(\|z^{r+1} - z^r\|^2 + \|w^{r+1} - w^r\|^2\right).
\end{aligned}
\tag{74}
$$

*Proof.* First, note that $K(x, y, w, z; \lambda)$ is strongly convex w.r.t. $x$ and $y$ jointly with parameter $p - L$. Under A2, A5 and the assumption that $y^r$ is bounded, we can obtain that $\nabla_z P(w^r, z^r) = p(z^r - \bar{y}^*(w^r, z^r))$ by applying the Danskin's theorem in the convex analysis [48]. Then, using the primal error bound (27c), we can show that $\nabla_z P(x^r)$ has a Lipschitz constant, i.e.,

$$
\|\nabla P(w^r, z^{r+1}) - \nabla P(w^r, z^r)\| \leq p\left(\frac{p}{p-L} + 1\right)\|z^{r+1} - z^r\|.
\tag{75}
$$

Therefore, it is straightforward that

$$
P(w^r, z^{r+1}) - P(w^r, z^r) \leq p(z^{r+1} - z^r)^T(z^r - \bar{y}^*(w^r, z^r)) + \frac{p}{2}\left(\frac{p}{p-L} + 1\right)\|z^{r+1} - z^r\|^2.
\tag{76}
$$

Similarly, we have

$$
\|\nabla_w P(w^{r+1}, z^{r+1}) - \nabla_w P(w^r, z^{r+1})\| \leq p\left(\frac{p}{p-L} + 1\right)\|w^{r+1} - w^r\|,
\tag{77}
$$

which gives

$$
\begin{aligned}
&P(w^{r+1}, z^{r+1}) - P(w^r, z^{r+1}) \\
&\leq p(w^{r+1} - w^r)^T(w^r - \bar{x}^*(w^r, z^{r+1})) + \frac{p}{2}\left(\frac{p}{p-L} + 1\right)\|w^{r+1} - w^r\|^2.
\end{aligned}
\tag{78}
$$

□

## B.3 Dual Error Bounds

### B.3.1 Proof of Weak Dual Error Bound (Lemma 2)

**Lemma 2** (Formal)

*Under A1-A5, suppose that $\lambda, \bar{y}^*(w, z)$ are bounded and $p > L$, then the weak error bound holds, namely,*

$$
\begin{aligned}
&\|y^*(w, z; \lambda_+(w, z)) - \bar{y}^*(w, z)\|^2 + \|x^*(w, z; \lambda_+(w, z)) - \bar{x}^*(w, z)\|^2 \\
&\leq \sigma_{\text{weak}}\|\lambda - \lambda_+(w, z)\|\|\lambda(w, z) - \lambda_+(w, z)\|
\end{aligned}
\tag{79}
$$

*where*

$$
\sigma_{\text{weak}} \triangleq \frac{1 + \tau(2\ell_g + \ell_g')\sigma_2}{2\tau(p-L)}, \quad \text{and} \quad \sigma_2 \triangleq \frac{p+L}{p-L}.
\tag{80}
$$

*Proof.* First, let

$$
\lambda_+(v) = P_+\left[\lambda + \tau\nabla_\lambda K(x^*(v; \lambda), y^*(v; \lambda), v; \lambda)\right],
\tag{81a}
$$

$$
\lambda(v) \in \arg\max_{0 \leq \lambda \leq \Lambda} K(\bar{x}^*(v), \bar{y}^*(v), v; \lambda).
\tag{81b}
$$

Based on the strong convexity of $K(x, \cdot, w, z; \lambda)$ and $K(\cdot, y, w, z; \lambda)$, we have

$$K(\bar{x}^*(v), \bar{y}^*(v), v; \lambda_+(v)) - K(x^*(v; \lambda_+(v)), \bar{y}^*(v), v; \lambda_+(v))$$
$$\geq \frac{p-L}{2} \|x^*(v; \lambda_+(v)) - \bar{x}^*(v)\|^2, \quad (82a)$$

$$K(x^*(v; \lambda_+(v)), \bar{y}^*(v), v; \lambda_+(v)) - K(x^*(v; \lambda_+(v)), y^*(v; \lambda_+(v)), v; \lambda_+(v))$$
$$\geq \frac{p-L}{2} \|y^*(v; \lambda_+(v)) - \bar{y}^*(v)\|^2, \quad (82b)$$

$$K(x^*(v; \lambda_+(v)), y^*(v; \lambda_+(v)), v; \lambda(v)) - K(x^*(v; \lambda_+(v)), \bar{y}^*(v), v; \lambda(v))$$
$$\geq \frac{p-L}{2} \|y^*(v; \lambda_+(v)) - \bar{y}^*(v)\|^2, \quad (82c)$$

$$K(x^*(v; \lambda_+(v)), \bar{y}^*(v), v; \lambda(v)) - K(\bar{x}^*(v), \bar{y}^*(v), v; \lambda(v))$$
$$\geq \frac{p-L}{2} \|x^*(v; \lambda_+(v)) - \bar{x}^*(v)\|^2. \quad (82d)$$

Note that $\lambda_+(v)$ is the maximizer of the following problem.

$$\max_{0 \leq \bar{\lambda} \leq \Lambda} \tau K(x^*(v; \lambda_+(v)), y^*(v; \lambda_+(v)), v; \bar{\lambda}) - \delta^T(v; \lambda, \lambda_+(v))\bar{\lambda} \quad (83)$$

where

$$\delta(v; \lambda, \lambda_+(v)) = (\lambda_+(v) + \tau \nabla_\lambda K(x^*(v; \lambda_+(v)), y^*(v; \lambda_+(v)), v; \lambda_+(v)))$$
$$- (\lambda + \tau \nabla_\lambda K(x^*(v; \lambda), y^*(v; \lambda), v; \lambda)). \quad (84)$$

According to the Lipschitz continuity of $\nabla_\lambda K$, we have

$$\|\delta(v; \lambda, \lambda_+(v))\|$$
$$\leq \|\lambda_+(v) - \lambda\| + \tau \|g(x^*(v; \lambda_+(v)), y^*(v; \lambda_+(v))) - g(x^*(v; \lambda), y^*(v; \lambda))\|$$
$$+ \tau \|g^*(x^*(v; \lambda_+(v))) - g^*(x^*(v; \lambda))\| \quad (85)$$
$$\overset{(a)}{\leq} \left(1 + \tau(2\ell_g + \ell_g')\sigma_2\right) \|\lambda - \lambda_+(v)\| \quad (86)$$

where in (a) we use the primal error bounds (28a), (28b), and apply the Lipschitz continuity of $g(,)$ and $g^*()$.

Based on the definition of $\lambda_+(v)$ (cf. (81a)), we have

$$\tau K(x^*(v; \lambda_+(v)), y^*(v; \lambda_+(v)), v; \lambda(v)) - \delta^T(v; \lambda, \lambda_+(v))\lambda(v)$$
$$\leq \tau K(x^*(v; \lambda_+(v)), y^*(v; \lambda_+(v)), v; \lambda_+(v)) - \delta^T(v; \lambda, \lambda_+(v))\lambda_+(v). \quad (87)$$

Subsequently, we can obtain

$$\tau K(x^*(v; \lambda_+(v)), y^*(v; \lambda_+(v)), v; \lambda(v)) - \tau K(x^*(v; \lambda_+(v)), y^*(v; \lambda_+(v)), v; \lambda_+(v))$$
$$\leq (\lambda(v) - \lambda_+(v))^T \delta(v; \lambda, \lambda_+(v)) \quad (88)$$
$$\overset{(86)}{\leq} \|\lambda(v) - \lambda_+(v)\| \left(1 + \tau(2\ell_g + \ell_g')\sigma_2\right) \|\lambda - \lambda_+(v)\|. \quad (89)$$

According to the definition of $\lambda(v)$ (cf. (81b)), we have

$$K(\bar{x}^*(v), \bar{y}^*(v), v; \lambda(v)) \geq K(\bar{x}^*(v), \bar{y}^*(v), v; \lambda_+(v)). \quad (90)$$

Combing (82a) to (82d), and (90) yields

$$2\tau(p - L) \left(\|y^*(v; \lambda_+(v)) - \bar{y}^*(v)\|^2 + \|x^*(v; \lambda_+(v)) - \bar{x}^*(v)\|^2\right)$$
$$\leq \|\lambda(v) - \lambda_+(v)\| \left(1 + \tau(2\ell_g + \ell_g')\sigma_2\right) \|\lambda - \lambda_+(v)\|. \quad (91)$$

Therefore, we have

$$\|y^*(w^r, z^r; \lambda_+(w^r, z^r)) - \bar{y}^*(w^r, z^r)\|^2 + \|x^*(w^r, z^{r+1}; \lambda_+(w^r, z^r)) - \bar{x}^*(w^r, z^r)\|^2$$
$$\leq \underbrace{\frac{1 + \tau(2\ell_g + \ell_g')\sigma_2}{2\tau(p - L)}}_{\triangleq \sigma_{\text{weak}}} \|\lambda^{r+1} - \lambda_+(w^r, z^r)\| \|\lambda(w^r, z^r) - \lambda_+(w^r, z^r)\|. \quad (92)$$

$\square$

### B.3.2 Proof of Auxiliary Dual Error Bound

**Lemma 8.**

*Assume that A1-A3, A5 are satisfied and that the local regularity condition holds. Suppose that $\lambda, y^r, y^*(v; \lambda)$ are bounded, then the following inequality holds.*

$$
\|\lambda(w, z) - \lambda_+(w, z)\|
$$
$$
\leq \sigma_{aux}\left(\|y^*(w, z; \lambda_+(w, z)) - \bar{y}^*(w, z)\| + \|x^*(w, z; \lambda_+(w, z)) - \bar{x}^*(w, z)\|\right) \quad (93)
$$

*where*

$$
\sigma_{aux} \triangleq \frac{L_f + p + \Lambda L_g}{\sqrt{2\mu_g\underline{\delta}}}. \quad (94)
$$

*Proof.* From the optimality condition of the block-$y$ update and by following the notations defined in the proof of Lemma 2, we have

$$
\nabla_y f(\bar{x}^*(v), \bar{y}^*(v)) + \lambda(v)\nabla_y g(\bar{x}^*(v), \bar{y}^*(v)) + p(\bar{y}^*(v) - z) = 0, \quad (95a)
$$
$$
\nabla_y f(x^*(v; \lambda_+(v)), y^*(v; \lambda_+(v)))
$$
$$
+ \lambda_+(v)\nabla_y g(x^*(v; \lambda_+(v)), y^*(v; \lambda_+(v))) + p(y^*(v; \lambda_+(v)) - z) = 0. \quad (95b)
$$

Taking difference of these two equations and using the Lipschitz continuity $\nabla_y g(,)$, we have

$$
\|(\lambda(v) - \lambda_+(v))\nabla_y g(\bar{x}^*(v), \bar{y}^*(v))\|
$$
$$
\leq \|\nabla_y f(\bar{x}^*(v), \bar{y}^*(v)) - \nabla_y f(x^*(v; \lambda_+(v)), y^*(v; \lambda_+(v)))\|
$$
$$
+ \lambda_+(v)\|\nabla_y g(\bar{x}^*(v), \bar{y}^*(v)) - \nabla_y g(x^*(v; \lambda_+(v)), y^*(v; \lambda_+(v)))\| + p\|\bar{y}^*(v) - y^*(v; \lambda_+(v))\|
$$
$$
\leq (L_f + p + \Lambda L_g)\left(\|\bar{y}^*(v) - y^*(v; \lambda_+(v))\| + \|\bar{x}^*(v) - x^*(v; \lambda_+(v))\|\right). \quad (96)
$$

Further, we know that the left hand side (LHS) of (96) has a lower bound, i.e.,

$$
\|(\lambda(v) - \lambda_+(v))\nabla_y g(\bar{x}^*(v), \bar{y}^*(v))\|
$$
$$
\overset{(a)}{\geq} |\lambda(v) - \lambda_+(v)|\sqrt{2\mu_g(g(\bar{x}^*(v), \bar{y}^*(v)) - g^*(x^*(v)))} \quad (97)
$$
$$
\overset{(b)}{\geq} |\lambda(v) - \lambda_+(v)|\sqrt{2\mu_g\underline{\delta}} \quad (98)
$$

where in $(a)$ we apply the PŁ condition, and $(b)$ results from the regularity condition. $\square$

### B.3.3 Proof of Strong Dual Error Bound (Lemma 3)
**Lemma 3** (Formal)

*Assume that A1-A5 are satisfied and that the local regularity condition holds. Suppose that $\lambda, \bar{y}^*(w, z), \forall r$ are bounded and $p > L$, then the strong error bound holds, namely, there exists a constant $\sigma_{strong}$ such that*

$$
\|y^*(w^r, z^r; \lambda_+^{r+1}(w^r, z^r)) - \bar{y}^*(w^r, z^r)\| + \|x^*(w^r, z^r; \lambda_+^{r+1}(w^r, z^r)) - \bar{x}^*(w^r, z^r)\|
$$
$$
\leq 2\sigma_{strong}\tau|g(x^*(w^r, z^r; \lambda^{r+1}), y^*(w^r, z^r; \lambda^{r+1})) - g^*(x^*(w^r, z^r; \lambda^{r+1})) - \delta| \quad (99)
$$

*where*

$$
\sigma_{strong} \triangleq \sigma_{weak}\sigma_{aux} = \sigma_{weak}\frac{L_f + p + \Lambda L_g}{\sqrt{2\mu_g\underline{\delta}}}. \quad (100)
$$

*Proof.* Under the assumptions that $p > L$, and $\lambda^r, y^r, \bar{y}^*(w^r, z^r)$ are bounded, combing (14) and (93) yields the following strong error bound.

$$
\frac{1}{2}\left(\|y^*(w^r, z^r; \lambda_+^{r+1}(w^r, z^r)) - \bar{y}^*(w^r, z^r)\| + \|x^*(w^r, z^r; \lambda_+^{r+1}(w^r, z^r)) - \bar{x}^*(w^r, z^r)\|\right)^2
$$
$$
\leq \underbrace{\sigma_{weak}\sigma_{aux}}_{\triangleq \sigma_{strong}}\|\lambda^{r+1} - \lambda_+(w^r, z^r)\|
$$
$$
\cdot \left(\|y^*(w^r, z^r; \lambda_+^{r+1}(w^r, z^r)) - \bar{y}^*(w^r, z^r)\| + \|x^*(w^r, z^r; \lambda_+^{r+1}(w^r, z^r)) - \bar{x}^*(w^r, z^r)\|\right),
$$

which gives

$$\|y^*(w^r, z^r; \lambda_+^{r+1}(w^r, z^r)) - \bar{y}^*(w^r, z^r)\| + \|x^*(w^r, z^r; \lambda_+^{r+1}(w^r, z^r)) - \bar{x}^*(w^r, z^r)\|$$

$$\leq 2\sigma_{\text{strong}}\|\lambda^{r+1} - \lambda_+^{r+1}(w^r, z^r)\| \tag{101}$$

$$\overset{(81a)}{\leq} 2\sigma_{\text{strong}} \tau |\nabla_\lambda K(x^*(w^r, z^r; \lambda^{r+1}), y^*(w^r, z^r; \lambda^{r+1}), z^r; \lambda^{r+1})| \tag{102}$$

$$\leq 2\sigma_{\text{strong}}\tau |g(x^*(w^r, z^r; \lambda^{r+1}), y^*(w^r, z^r; \lambda^{r+1})) - g^*(x^*(w^r, z^r; \lambda^{r+1})) - \delta|. \tag{103}$$

$\square$

## B.4 Proof of Potential function (Lemma 1)

**Lemma 1** (Formal)

*Assume that A1-A5 are satisfied. Suppose that the sequence $\{x^r, y^r, w^r, z^r, u^r, \lambda^r, \forall r\}$ is generated by SLM, $p > L$, and $\lambda^r, y^r, \bar{y}^*(w^r, z^r)$ are bounded. When $\gamma, \eta, \alpha, T^r$ respectively satisfy (30),(41) and $T^r \geq -2\log_{(1-2\gamma\mu_g)}(\sqrt{\tau}r)$, then, there exists a constant $\zeta$ such that*

$$\mathcal{Q}(x^{r+1}, y^{r+1}, w^{r+1}, z^{r+1}; \lambda^{r+1}) - \mathcal{Q}(x^r, y^r, w^r, z^r; \lambda^r)$$

$$\leq -\left(\frac{1}{4\eta} - \frac{17\tau(\ell_g^2 + \ell_g'^2)\sigma_3^2}{\eta^2}\right)\|x^{r+1} - x^r\|^2 - \left(\frac{1}{2\alpha} - \frac{9\tau\ell_g^2\sigma_3^2}{\alpha^2}\right)\|y^{r+1} - y^r\|^2$$

$$- p\left(\frac{1}{2\beta} - \left(\frac{1}{\zeta} + \frac{2p}{p-L} + 36\zeta\sigma_1^2\right)\right)\|z^{r+1} - z^r\|^2 - p\left(\frac{1}{2\beta} - \left(\frac{1}{\zeta} + \frac{2p}{p-L}\right)\right)\|w^{r+1} - w^r\|^2$$

$$+ 36p\zeta\|x^*(w^r, z^r; \lambda_+^{r+1}(w^r, z^r)) - \bar{x}^*(w^r, z^r)\|^2$$

$$+ 36p\zeta\|y^*(w^r, z^r; \lambda_+^{r+1}(w^r, z^r)) - \bar{y}^*(w^r, z^r)\|^2$$

$$- \tau\left(\frac{1}{56} - 48p\zeta\sigma_2^2\tau\right)|g(x^*(w^r, z^r; \lambda^{r+1}), y^*(w^r, z^r; \lambda^{r+1})) - g^*(x^*(w^r, z^r; \lambda^{r+1})) - \delta|^2$$

$$+ \left(\mu_g L_g^2 + 2\ell_g^2\right)\frac{D_\mathcal{S}}{r^2} - \frac{1}{4\tau}\|\lambda^{r+1} - \lambda^r\|^2 \tag{104}$$

*where*

$$\mathcal{Q}^r \triangleq \mathcal{Q}(x^r, y^r, w^r, z^r; \lambda^r) \triangleq K(x^r, y^r, w^r, z^r; \lambda^r) - 2D(w^r, z^r; \lambda^r) + 2P(w^r, z^r). \tag{105}$$

*Proof.* Recall that

$$\mathcal{Q}^r = K(x^r, y^r, w^r, z^r; \lambda^r) - 2D(w^r, z^r; \lambda^r) + 2P(w^r, z^r). \tag{106}$$

Merging (42), (61), and (74) gives

$$\mathcal{Q}^{r+1} - \mathcal{Q}^r$$

$$\leq -\left(\frac{1}{4\eta}\|x^{r+1} - x^r\|^2 + \frac{1}{2\alpha}\|y^{r+1} - y^r\|^2 + \frac{p}{2\beta}\|z^{r+1} - z^r\|^2 + \frac{p}{2\beta}\|w^{r+1} - w^r\|^2\right)$$

$$+ \langle g(x^r, y^r) - g^*(x^r) - \delta, \lambda^{r+1} - \lambda^r\rangle$$

$$- 2\langle \lambda^{r+1} - \lambda^r, g(x^*(w^r, z^r; \lambda^{r+1}), y^*(w^r, z^r; \lambda^{r+1})) - g^*(x^*(w^r, z^r; \lambda^{r+1})) - \delta)\rangle$$

$$\underbrace{-p\langle z^{r+1} - z^r, z^{r+1} + z^r - 2y^*(w^r, z^{r+1}; \lambda^{r+1})\rangle - 2p(z^{r+1} - z^r)^T(\bar{y}^*(w^r, z^r) - z^r)}_{\triangleq T_1}$$

$$\underbrace{-p\langle w^{r+1} - w^r, w^{r+1} + w^r - 2x^*(w^{r+1}, z^{r+1}; \lambda^{r+1})\rangle - 2p(w^{r+1} - w^r)^T(\bar{x}^*(w^r, z^{r+1}) - w^r)}_{\triangleq T_2}$$

$$+ p\left(\frac{p}{p-L} + 1\right)\left(\|z^{r+1} - z^r\|^2 + \|w^{r+1} - w^r\|^2\right) + \frac{\mu_g L_g^2 D_\mathcal{S}}{r^2}. \tag{107}$$

First, we can get the upper bound of term $T_1$ as follows:

$$T_1 = -p\left\langle z^{r+1} - z^r, z^{r+1} - z^r - 2\left(y^*(w^r, z^{r+1}; \lambda^{r+1}) - \bar{y}^*(w^r, z^r)\right)\right\rangle$$

$$= -p\left\langle z^{r+1} - z^r, z^{r+1} - z^r - 2\left(y^*(w^r, z^{r+1}; \lambda^{r+1}) - y^*(w^r, z^r; \lambda^{r+1})\right)\right\rangle$$

$$- p\left\langle z^{r+1} - z^r, y^*(w^r, z^r; \lambda^{r+1}) - \bar{y}^*(w^r, z^r)\right\rangle \tag{108}$$

$$= -p\|z^{r+1} - z^r\|^2 + 2p\left\langle z^{r+1} - z^r, y^*(w^r, z^{r+1}; \lambda^{r+1}) - y^*(w^r, z^r; \lambda^{r+1})\right\rangle$$

$$- p\left\langle z^{r+1} - z^r, y^*(w^r, z^r; \lambda^{r+1}) - \bar{y}^*(w^r, z^r)\right\rangle. \tag{109}$$

For the last term of the above inequality, we can further have

$$p\left\langle z^{r+1} - z^r, y^*(w^r, z^r; \lambda^{r+1}) - \bar{y}^*(w^r, z^r)\right\rangle$$

$$\leq \frac{p\|z^{r+1} - z^r\|^2}{2\zeta} + \frac{p\zeta}{2}\|y^*(w^r, z^r; \lambda^{r+1}) - \bar{y}^*(w^r, z^r)\|^2. \tag{110}$$

For the second term of (109), we can get

$$\left\langle z^{r+1} - z^r, y^*(w^r, z^{r+1}; \lambda^{r+1}) - y^*(w^r, z^r; \lambda^{r+1})\right\rangle$$

$$\overset{(a)}{\leq} \|z^{r+1} - z^r\|\|y^*(w^r, z^{r+1}; \lambda^{r+1}) - y^*(w^r, z^r; \lambda^{r+1})\| \tag{111}$$

$$\overset{(b)}{\leq} \frac{p}{p-L}\|z^{r+1} - z^r\|^2 \tag{112}$$

where $(a)$ is true by applying the Cauchy-Schwarz inequality, in $(b)$ we use the primal error bound (27a).

Similarly, we have

$$T_2 = -p\left\langle w^{r+1} - w^r, w^{r+1} - w^r - 2\left(x^*(w^{r+1}, z^{r+1}; \lambda^{r+1}) - \bar{x}^*(w^r, z^{r+1})\right)\right\rangle$$

$$= -p\left\langle w^{r+1} - w^r, w^{r+1} - w^r - 2\left(x^*(w^{r+1}, z^{r+1}; \lambda^{r+1}) - x^*(w^r, z^{r+1}; \lambda^{r+1})\right)\right\rangle$$

$$- p\left\langle w^{r+1} - w^r, x^*(w^r, z^{r+1}; \lambda^{r+1}) - \bar{x}^*(w^r, z^{r+1})\right\rangle \tag{113}$$

$$= -p\|w^{r+1} - w^r\|^2 + 2p\left\langle w^{r+1} - w^r, x^*(w^{r+1}, z^{r+1}; \lambda^{r+1}) - x^*(w^r, z^{r+1}; \lambda^{r+1})\right\rangle$$

$$- p\left\langle w^{r+1} - w^r, x^*(w^r, z^{r+1}; \lambda^{r+1}) - \bar{x}^*(w^r, z^{r+1})\right\rangle. \tag{114}$$

For the last term of the above inequality, we can further have

$$p\left\langle w^{r+1} - w^r, x^*(w^r, z^{r+1}; \lambda^{r+1}) - \bar{x}^*(w^r, z^{r+1})\right\rangle$$

$$\leq \frac{p\|w^{r+1} - w^r\|^2}{2\zeta} + \frac{p\zeta}{2}\|x^*(w^r, z^{r+1}; \lambda^{r+1}) - \bar{x}^*(w^r, z^{r+1})\|^2. \tag{115}$$

In addition, we have

$$\left\langle w^{r+1} - w^r, x^*(w^{r+1}, z^{r+1}; \lambda^{r+1}) - y^*(w^r, z^{r+1}; \lambda^{r+1})\right\rangle$$

$$\overset{(a)}{\leq} \|w^{r+1} - w^r\|\|x^*(w^{r+1}, z^{r+1}; \lambda^{r+1}) - x^*(w^r, z^{r+1}; \lambda^{r+1})\| \tag{116}$$

$$\overset{(b)}{\leq} \frac{p}{p-L}\|w^{r+1} - w^r\|^2 \tag{117}$$

where $(a)$ is true by applying the Cauchy-Schwarz inequality, in $(b)$ we use the primal error bound (27b).

Substituting (110), (112), (115), (117) into (107) yields

$$
\begin{aligned}
&\mathcal{Q}^{r+1} - \mathcal{Q}^r \\
&\leq -\left(\frac{1}{4\eta}\|x^{r+1} - x^r\|^2 + \frac{1}{2\alpha}\|y^{r+1} - y^r\|^2\right) \\
&\quad - p\left(\frac{1}{2\beta} - \left(\frac{1}{\zeta} + \frac{2p}{p-L}\right)\right)\|z^{r+1} - z^r\|^2 \\
&\quad - p\left(\frac{1}{2\beta} - \left(\frac{1}{\zeta} + \frac{2p}{p-L}\right)\right)\|w^{r+1} - w^r\|^2 \\
&\quad + p\zeta\left(\|x^*(w^r, z^{r+1}; \lambda^{r+1}) - \bar{x}^*(w^r, z^{r+1})\|^2 + \|y^*(w^r, z^r; \lambda^{r+1}) - \bar{y}^*(w^r, z^r)\|^2\right) + \frac{\mu_g L_g^2 D_{\mathcal{S}}}{r^2} \\
&\quad + \langle g(x^r, y^r) - g^*(x^r) - \delta, \lambda^{r+1} - \lambda^r\rangle \\
&\quad - 2\langle \lambda^{r+1} - \lambda^r, g(x^*(w^r, z^r; \lambda^{r+1}), y^*(w^r, z^r; \lambda^{r+1})) - g^*(x^*(w^r, z^r; \lambda^{r+1})) - \delta)\rangle. \quad (118)
\end{aligned}
$$

Second, we will give an upper bound of the last two terms of (118) as follows.

*Step 1.)* From the update of the dual variable, we can have

$$
\langle g(x^r, y^r) - g(x^r, u^{r+1}) - \delta, \lambda^{r+1} - \lambda^r\rangle \geq \frac{1}{\tau}\|\lambda^{r+1} - \lambda^r\|^2, \quad (119)
$$

which is equivalent to

$$
-\langle g(x^r, y^r) - g(x^r, u^{r+1}) - \delta, \lambda^{r+1} - \lambda^r\rangle \leq -\frac{1}{\tau}\|\lambda^{r+1} - \lambda^r\|^2. \quad (120)
$$

Therefore, we can obtain

$$
\begin{aligned}
&\langle g(x^r, y^r) - g^*(x^r) - \delta, \lambda^{r+1} - \lambda^r\rangle \\
&= \langle g(x^r, y^r) - g^*(x^r) - \delta, \lambda^{r+1} - \lambda^r\rangle + \langle g(x^r, y^r) - g(x^r, u^{r+1}) - \delta, \lambda^{r+1} - \lambda^r\rangle \\
&\quad - \langle g(x^r, y^r) - g(x^r, u^{r+1}) - \delta, \lambda^{r+1} - \lambda^r\rangle \\
&\leq 2\langle g(x^r, y^r) - g^*(x^r) - \delta, \lambda^{r+1} - \lambda^r\rangle + \langle g^*(x^r) - g(x^r, u^{r+1}), \lambda^{r+1} - \lambda^r\rangle \\
&\quad - \frac{1}{\tau}\|\lambda^{r+1} - \lambda^r\|^2. \quad (122)
\end{aligned}
$$

Subsequently, we can derive an upper bound for the sum of the last two terms in (118) as follows.

$$
\begin{aligned}
&2\langle g(x^r, y^r) - g^*(x^r), \lambda^{r+1} - \lambda^r\rangle \\
&\quad - 2\langle g(x^*(w^r, z^r; \lambda^{r+1}), y^*(w^r, z^r; \lambda^{r+1})) - g^*(x^*(w^r, z^r; \lambda^{r+1})), \lambda^{r+1} - \lambda^r\rangle \\
&\quad + \langle \lambda^{r+1} - \lambda^r, g^*(x^r) - g(x^r, u^{r+1})\rangle - \frac{1}{\tau}\|\lambda^{r+1} - \lambda^r\|^2 \quad (123) \\
&\overset{(a)}{\leq} 4\tau\left(\|g(x^r, y^r) - g^*(x^r) - (g(x^*(w^r, z^r; \lambda^{r+1}), y^*(w^r, z^r; \lambda^{r+1})) - g^*(x^*(w^r, z^r; \lambda^{r+1})))\|^2\right) \\
&\quad + \tau\|g^*(x^r) - g(x^r, u^{r+1})\|^2 - \frac{1}{2\tau}\|\lambda^{r+1} - \lambda^r\|^2 \quad (124) \\
&\leq 8\tau\left(2(\ell_g^2 + \ell_g'^2)\|x^*(w^r, z^r; \lambda^{r+1}) - x^r\|^2 + \ell_g^2\|y^*(w^r, z^r; \lambda^{r+1}) - y^r\|^2\right) \\
&\quad + \tau\|g^*(x^r) - g(x^r, u^{r+1})\|^2 - \frac{1}{2\tau}\|\lambda^{r+1} - \lambda^r\|^2 \quad (125) \\
&\overset{(b)}{\leq} 8\tau\sigma_3^2\left(\frac{2(\ell_g^2 + \ell_g'^2)}{\eta^2}\|x^{r+1} - x^r\|^2 + \frac{\ell_g^2}{\alpha^2}\|y^{r+1} - y^r\|^2\right) \\
&\quad + \tau\|g^*(x^r) - g(x^r, u^{r+1})\|^2 - \frac{1}{2\tau}\|\lambda^{r+1} - \lambda^r\|^2 \quad (126)
\end{aligned}
$$

where in $(a)$ we use the Cauchy-Schwarz inequality, in $(b)$ we apply the primal error bounds (29a) and (29b).

*Step 2.)* Applying the reverse triangle inequality, we can get

$$\|\lambda^{r+1} - \lambda^r\|^2$$
$$= \|\lambda^{r+1} - \lambda_+^{r+1}(w^r, z^r) + \lambda_+^{r+1}(w^r, z^r) - \lambda^r\|^2 \tag{127}$$
$$\geq \frac{\|\lambda_+^{r+1}(w^r, z^r) - \lambda^r\|^2}{2} - \|\lambda^{r+1} - \lambda_+^{r+1}(w^r, z^r)\|^2. \tag{128}$$

According to the definition of $\lambda_+(w^r, z^r)$ (cf. (81a)), we have

$$\|\lambda^{r+1} - \lambda_+^{r+1}(w^r, z^r)\|^2$$
$$\stackrel{(a)}{\leq} \|\lambda^r + \tau(g(x^r, y^r) - g(x^r, u^{r+1}) - \delta) - [\lambda^{r+1} + \tau\nabla_\lambda K(x^*(v^r; \lambda^{r+1}), y^*(v^r; \lambda^{r+1}), v^r; \lambda^{r+1})]\|^2$$
$$\leq 2\|\lambda^{r+1} - \lambda^r\|^2$$
$$\quad + 2\tau^2 \left( \|g(x^r, y^r) - g(x^r, u^{r+1}) - (g(x^*(v^r; \lambda^{r+1}), y^*(v^r; \lambda^{r+1})) - g^*(x^*(v^r; \lambda^{r+1})))\|^2 \right)$$
$$\leq 2\|\lambda^{r+1} - \lambda^r\|^2 + 6\tau^2 \left( 2(\ell_g^2 + \ell_g'^2)\|x^r - x^*(v^r; \lambda^{r+1})\|^2 + \ell_g^2\|y^r - y^*(v^r; \lambda^{r+1})\|^2 \right)$$
$$\quad + 6\tau^2\|g(x^r, u^{r+1}) - g^*(x^r)\|^2 \tag{129}$$

where $(a)$ is true due to the nonexpansiveness of the nonnegative projection operator. Also, we have

$$\|\lambda^r - \lambda_+^{r+1}(w^r, z^r)\|^2$$
$$= \|\lambda^r - [\lambda^{r+1} + \tau\nabla_\lambda K(x^*(v^r; \lambda^{r+1}), y^*(v^r; \lambda^{r+1}), v^r; \lambda^{r+1})]\| \tag{130}$$
$$\geq \frac{\|\tau\nabla_\lambda K(x^*(v^r; \lambda^{r+1}), y^*(v^r; \lambda^{r+1}), v^r; \lambda^{r+1})\|^2}{2} - \|\lambda^{r+1} - \lambda^r\|^2. \tag{131}$$

Combining (128), (129), and (131) gives

$$\frac{7}{2}\|\lambda^{r+1} - \lambda^r\|^2$$
$$\geq \frac{\|\tau\nabla_\lambda K(x^*(v^r; \lambda^{r+1}), y^*(v^r; \lambda^{r+1}), v^r; \lambda^{r+1})\|^2}{4}$$
$$\quad - 6\tau^2 \left( 2(\ell_g^2 + \ell_g'^2)\|x^r - x^*(v^r; \lambda^{r+1})\|^2 + \ell_g^2\|y^r - y^*(v^r; \lambda^{r+1})\|^2 \right)$$
$$\quad - 6\tau^2\|g(x^r, u^{r+1}) - g^*(x^r)\|^2. \tag{132}$$

Applying the primal error bounds (29a) and (29b), we can obtain

$$-\frac{1}{4\tau}\|\lambda^{r+1} - \lambda^r\|^2$$
$$\leq -\frac{\tau\|\nabla_\lambda K(x^*(v^r; \lambda^{r+1}), y^*(v^r; \lambda^{r+1}), v^r; \lambda^{r+1})\|^2}{56}$$
$$\quad + \frac{3\tau}{7} \left( 2(\ell_g^2 + \ell_g'^2)\|x^r - x^*(v^r; \lambda^{r+1})\|^2 + \ell_g^2\|y^r - y^*(v^r; \lambda^{r+1})\|^2 + \|g(x^r, u^{r+1}) - g^*(x^r)\|^2 \right).$$
$$\tag{133}$$
$$\leq -\frac{\tau\|g(x^*(w^r, z^r; \lambda^{r+1}), y^*(w^r, z^r; \lambda^{r+1})) - g^*(x^*(w^r, z^r; \lambda^{r+1})) - \delta\|^2}{56}$$
$$\quad + \frac{3\tau}{7} \left( \frac{2(\ell_g^2 + \ell_g'^2)\sigma_3^2}{\eta^2}\|x^r - x^{r+1}\|^2 + \frac{\ell_g^2\sigma_3^2}{\alpha^2}\|y^r - y^{r+1}\|^2 + \|g(x^r, u^{r+1}) - g^*(x^r)\|^2 \right).$$
$$\tag{134}$$

*Step 3.)* Substituting (134) and (126) back to (118) gives

$$\mathcal{Q}^{r+1} - \mathcal{Q}^r$$

$$\leq -\left(\frac{1}{4\eta} - \frac{17\tau(\ell_g^2 + \ell_g'^2)\sigma_3^2}{\eta^2}\right)\|x^{r+1} - x^r\|^2 - \left(\frac{1}{2\alpha} - \frac{9\tau\ell_g^2\sigma_3^2}{\alpha^2}\right)\|y^{r+1} - y^r\|^2$$

$$- p\left(\frac{1}{2\beta} - \left(\frac{1}{\zeta} + \frac{2p}{p-L}\right)\right)\|z^{r+1} - z^r\|^2 - p\left(\frac{1}{2\beta} - \left(\frac{1}{\zeta} + \frac{2p}{p-L}\right)\right)\|w^{r+1} - w^r\|^2$$

$$+ 6p\zeta\left(\|y^*(w^r, z^r; \lambda^{r+1}) - \bar{y}^*(w^r, z^r)\|^2 + \|x^*(w^r, z^{r+1}; \lambda^{r+1}) - \bar{x}^*(w^r, z^{r+1})\|^2\right)$$

$$- \frac{\tau}{56}|g(x^*(w^r, z^r; \lambda^{r+1}), y^*(w^r, z^r; \lambda^{r+1})) - g^*(x^*(w^r, z^r; \lambda^{r+1})) - \delta|^2$$

$$+ \frac{\mu_g L_g^2 D_{\mathcal{S}}}{r^2} + 2\tau\|g(x^r, u^{r+1}) - g^*(x^r)\|^2 - \frac{1}{4\tau}\|\lambda^{r+1} - \lambda^r\|^2. \tag{135}$$

Finally, by observing the weak dual error bound, we need to further quantify $\|y^*(w^r, z^r; \lambda^{r+1}) - \bar{y}^*(w^r, z^r)\|^2 + \|x^*(w^r, z^{r+1}; \lambda^{r+1}) - \bar{x}^*(w^r, z^{r+1})\|^2$ as follows. Note that

$$\|y^*(w^r, z^r; \lambda^{r+1}) - \bar{y}^*(w^r, z^r)\|^2 + \|x^*(w^r, z^{r+1}; \lambda^{r+1}) - \bar{x}^*(w^r, z^{r+1})\|^2$$

$$\leq \|y^*(w^r, z^r; \lambda^{r+1}) - \bar{y}^*(w^r, z^r)\|^2 + 3\|x^*(w^r, z^r; \lambda^{r+1}) - \bar{x}^*(w^r, z^r)\|^2$$

$$+ 3\|x^*(w^r, z^{r+1}; \lambda^{r+1}) - x^*(w^r, z^r; \lambda^{r+1})\|^2 + 3\|\bar{x}^*(w^r, z^{r+1}) - \bar{x}^*(w^r, z^r)\|^2 \tag{136}$$

$$\overset{(a)}{\leq} 2\|y^*(w^r, z^r; \lambda_+^{r+1}(w^r, z^r)) - \bar{y}^*(w^r, z^r)\|^2 + 6\|x^*(w^r, z^r; \lambda_+^{r+1}(w^r, z^r)) - \bar{x}^*(w^r, z^r)\|^2$$

$$+ 2\|y^*(w^r, z^r; \lambda^{r+1}) - y^*(w^r, z^r; \lambda_+^{r+1}(w^r, z^r))\|^2$$

$$+ 6\|x^*(w^r, z^r; \lambda^{r+1}) - x^*(w^r, z^r; \lambda_+^{r+1}(w^r, z^r))\|^2 + 6\sigma_1^2\|z^{r+1} - z^r\|^2 \tag{137}$$

$$\overset{(b)}{\leq} 2\|y^*(w^r, z^r; \lambda_+^{r+1}(w^r, z^r)) - \bar{y}^*(w^r, z^r)\|^2 + 6\|x^*(w^r, z^r; \lambda_+^{r+1}(w^r, z^r)) - \bar{x}^*(w^r, z^r)\|^2$$

$$+ 8\sigma_2^2\tau^2 \left\|g(x^*(w^r, z^r; \lambda^{r+1}), y^*(w^r, z^r; \lambda^{r+1})) - g(x^*(w^r, z^r; \lambda^{r+1})) - \delta\right\|^2$$

$$+ 6\sigma_1^2\|z^{r+1} - z^r\|^2 \tag{138}$$

where in $(a)$ we use the primal error bounds (27a) and (27c), $(b)$ holds as we first apply the primal error bounds (28a) and (28b) and then use the definition $\lambda_+(w, z)$ as follows

$$\|\lambda^{r+1} - \lambda_+^{r+1}(w^r, z^r)\|$$

$$= \|\lambda^{r+1} - P_{\geq 0}\left[\lambda^{r+1} + \tau\nabla_\lambda K(x^*(w^r, z^r; \lambda^{r+1}), y^*(w^r, z^r; \lambda^{r+1}), w^r, z^r; \lambda^{r+1})\right]\| \tag{139}$$

$$\leq \tau\left\|g(x^*(w^r, z^r; \lambda^{r+1}), y^*(w^r, z^r; \lambda^{r+1})) - g(x^*(w^r, z^r; \lambda^{r+1})) - \delta\right\|. \tag{140}$$

As a result, we can get

$$\mathcal{Q}^{r+1} - \mathcal{Q}^r$$

$$\leq -\left(\frac{1}{4\eta} - \frac{17\tau(\ell_g^2 + \ell_g'^2)\sigma_3^2}{\eta^2}\right)\|x^{r+1} - x^r\|^2 - \left(\frac{1}{2\alpha} - \frac{9\tau\ell_g^2\sigma_3^2}{\alpha^2}\right)\|y^{r+1} - y^r\|^2$$

$$- p\left(\frac{1}{2\beta} - \left(\frac{1}{\zeta} + \frac{2p}{p-L} + 36\zeta\sigma_1^2\right)\right)\|z^{r+1} - z^r\|^2 - p\left(\frac{1}{2\beta} - \left(\frac{1}{\zeta} + \frac{2p}{p-L}\right)\right)\|w^{r+1} - w^r\|^2$$

$$+ 36p\zeta\left(\|y^*(w^r, z^r; \lambda_+^{r+1}(w^r, z^r)) - \bar{y}^*(w^r, z^r)\|^2 + \|x^*(w^r, z^r; \lambda_+^{r+1}(w^r, z^r)) - \bar{x}^*(w^r, z^r)\|^2\right)$$

$$- \tau\left(\frac{1}{56} - 48p\zeta\sigma_2^2\tau\right)|g(x^*(w^r, z^r; \lambda^{r+1}), y^*(w^r, z^r; \lambda^{r+1})) - g^*(x^*(w^r, z^r; \lambda^{r+1})) - \delta|^2$$

$$+ \frac{\mu_g L_g^2 D_{\mathcal{S}}}{r^2} + 2\tau\|g(x^r, u^{r+1}) - g^*(x^r)\|^2 - \frac{1}{4\tau}\|\lambda^{r+1} - \lambda^r\|^2 \tag{141}$$

From Lemma 4, we know that $d_{\mathcal{S}(x^r)}^2(u^{r+1}) \leq (1 - 2\gamma\mu_g)^{T^r} D_{\mathcal{S}}$, so when $T^r \geq -2\log_{(1-2\gamma\mu_g)}(\sqrt{\tau}r)$,

$$2\tau\|g(x^r, u^{r+1}) - g^*(x^r)\|^2 \leq \frac{2\ell_g^2 D_{\mathcal{S}}}{r^2}. \tag{142}$$

Substituting (142) back to (141) gives the desired result. $\qquad\square$

## B.5 Descent of Potential Function

**Lemma 9.**

*Assume that A1-A5 are satisfied and that the local regularity condition holds. Suppose that the sequence $\{x^r, y^r, w^r, z^r, u^r, \lambda^r, \forall r\}$ is generated by SLM, $p > L$, and $\lambda^r, y^r, \bar{y}^*(w^r, z^r), y^*(w^r, z^r; \lambda^{r+1})$ are bounded. When the step-sizes are chosen such that*

$$\gamma \leq \min\left\{\frac{1}{L_g}, \frac{1}{2\mu_g}\right\}, \quad \eta \leq \min\left\{\frac{1}{L_K}, \frac{(p-L)^2}{136\tau(\ell_g^2 + \ell_g'^2)L_K^2}\right\},$$

$$\alpha \leq \min\left\{\frac{1}{L_K}, \frac{(p-L)^2}{36\tau\ell_g^2 L_K^2}\right\}, \quad \tau \leq \min\left\{\frac{(p-L)^2}{3456p\zeta(p+L)^2}, \frac{1/(126 \cdot 48)}{p\zeta\max\{\sigma_{strong}^2, \sigma_2^2\}}\right\},$$

$$T^r \geq \max\left\{-2\log_{1-2\gamma\mu_g}(\lambda^{r+1}\sqrt{\eta}r), -2\log_{1-2\gamma\mu_g}r\sqrt{\tau}\right\},$$

$$\beta \leq \frac{1/4}{\frac{1}{\zeta} + \frac{2p}{p-L} + \frac{36\zeta p^2}{(p-L)^2}} \leq 1, \quad \zeta \leq \sqrt{\frac{\varrho}{2}}\max\left\{\frac{1}{\zeta_1}, \frac{1}{\zeta_2}\right\}, \quad \vartheta \leq \frac{\varrho}{\tau\zeta_3}, \tag{143}$$

*then, we have*

$$Q^{r+1} - Q^r$$

$$\leq -\frac{1}{24\eta}\|x^{r+1} - x^r\|^2 - \frac{1}{12\alpha}\|y^{r+1} - y^r\|^2 - \frac{p}{12\beta}\|z^{r+1} - z^r\|^2 - \frac{p}{12\beta}\|w^{r+1} - w^r\|^2$$

$$- \frac{\tau}{189}|g(x^*(v^r; \lambda^{r+1})), y^*(v^r; \lambda^{r+1})) - g^*(x^*(v^r; \lambda^{r+1})) - \delta|^2$$

$$- \frac{1}{4\tau}\|\lambda^{r+1} - \lambda^r\|^2 + (\mu_g L_g^2 + 2\ell_g^2)\frac{D_S}{r^2} \tag{144}$$

*where $\sigma_{strong} \triangleq \sigma_{weak}\sigma_{aux}$, and $\zeta_1, \zeta_2, \zeta_3$ are defined in (171) and (174).*

*Proof.* From (104), it is clear that if we can select the step-sizes properly so that the coefficients in front of $\|x^{r+1} - x^r\|^2$, $\|y^{r+1} - y^r\|^2$, $\|z^{r+1} - z^r\|^2$, $\|w^{r+1} - w^r\|^2$ are strictly negative, then the potential function $Q^r$ will be decreasing. To be more specific, the step-sizes are chosen as follows.

1) Selection of $\eta$. Given the condition of (41), we request

$$\frac{1}{4\eta} - \frac{17\tau(\ell_g^2 + \ell_g'^2)\sigma_3^2}{\eta^2} = \frac{1}{4\eta} - \frac{17\tau(\ell_g^2 + \ell_g'^2)}{\eta^2(p-L)^2} > \frac{1}{8\eta} > 0, \tag{145}$$

i.e.,

$$\eta < \frac{(p-L)^2}{136\tau(\ell_g^2 + \ell_g'^2)L_K^2}. \tag{146}$$

2) Selection of $\alpha$.

$$\frac{1}{2\alpha} - \frac{9\tau\ell_g^2\sigma_3^2}{\alpha^2} = \frac{1}{2\alpha} - \frac{9\tau\ell_g^2}{\alpha^2(p-L)^2} > \frac{1}{4\alpha} > 0, \tag{147}$$

i.e.,

$$\alpha < \frac{(p-L)^2}{36\tau\ell_g^2 L_K^2}. \tag{148}$$

3) Selection of $\beta$.

$$\frac{1}{2\beta} - \left(\frac{1}{\zeta} + \frac{2p}{p-L} + 36\zeta\sigma_1^2\right) > \frac{1}{4\beta} > 0, \tag{149}$$

i.e.,

$$\beta < \frac{1}{4}\frac{1}{\frac{1}{\zeta} + \frac{2p}{p-L} + \frac{36\zeta p^2}{(p-L)^2}}. \tag{150}$$

4) Selection of $\tau$.

We need

$$\frac{1}{56} - 48p\zeta\sigma_2^2\tau \geq \frac{1}{63} > 0, \tag{151}$$

i.e.,

$$\tau \leq \frac{(p-L)^2}{3456p\zeta(p+L)^2}. \tag{152}$$

Then, consider the following two cases:

*Case 1.*

$$\frac{1}{3}\max\left\{\frac{1}{8\eta}\|x^{r+1}-x^r\|^2, \frac{1}{4\alpha}\|y^{r+1}-y^r\|^2, \frac{p}{4\beta}\|z^{r+1}-z^r\|^2, \frac{p}{4\beta}\|w^{r+1}-w^r\|^2,\right.$$

$$\left.\frac{\tau}{63}|g(x^*(w^r,z^r;\lambda^{r+1}),y^*(w^r,z^r;\lambda^{r+1})) - g^*(x^*(w^r,z^r;\lambda^{r+1})) - \delta|^2\right\}$$

$$> \max\left\{36p\zeta\big(\|y^*(w^r,z^r;\lambda_+^{r+1}(w^r,z^r)) - \bar{y}^*(w^r,z^r)\|^2\right.$$

$$\left.+ \|x^*(w^r,z^r;\lambda_+^{r+1}(w^r,z^r)) - \bar{x}^*(w^r,z^r)\|^2\big), \vartheta\tau\right\} \tag{153}$$

where $\vartheta$ is some parameter.

In this case, we can have

$$\mathcal{Q}^{r+1} - \mathcal{Q}^r \leq -\frac{1}{24\eta}\|x^{r+1}-x^r\|^2 - \frac{1}{12\alpha}\|y^{r+1}-y^r\|^2 - \frac{p}{12\beta}\|z^{r+1}-z^r\|^2 - \frac{p}{12\beta}\|w^{r+1}-w^r\|^2$$

$$- \frac{\tau}{189}|g(x^*(v^r;\lambda^{r+1})),y^*(v^r;\lambda^{r+1})) - g^*(x^*(v^r;\lambda^{r+1})) - \delta|^2$$

$$- \vartheta\tau - \frac{1}{4\tau}\|\lambda^{r+1}-\lambda^r\|^2 + \big(\mu_g L_g^2 + 2\ell_g^2\big)\frac{D_{\mathcal{S}}}{r^2}, \tag{154}$$

meaning that $\mathcal{Q}^r$ is decreasing up to some constant at each step.

*Case 2.*

$$\frac{1}{3}\max\left\{\frac{1}{8\eta}\|x^{r+1}-x^r\|^2, \frac{1}{4\alpha}\|y^{r+1}-y^r\|^2, \frac{p}{4\beta}\|z^{r+1}-z^r\|^2, \frac{p}{4\beta}\|w^{r+1}-w^r\|^2,\right.$$

$$\left.\frac{\tau}{63}|g(x^*(w^r,z^r;\lambda^{r+1}),y^*(w^r,z^r;\lambda^{r+1})) - g^*(x^*(w^r,z^r;\lambda^{r+1})) - \delta|^2\right\}$$

$$\leq \max\{36p\zeta\big(\|y^*(w^r,z^r;\lambda_+^{r+1}(w^r,z^r)) - \bar{y}^*(w^r,z^r)\|^2$$

$$+ \|x^*(w^r,z^r;\lambda_+^{r+1}(w^r,z^r)) - \bar{x}^*(w^r,z^r)\|^2\big), \vartheta\tau\}. \tag{155}$$

First, if $36p\zeta\big(\|y^*(w^r,z^r;\lambda_+^{r+1}(w^r,z^r)) - \bar{y}^*(w^r,z^r)\|^2 + \|x^*(w^r,z^r;\lambda_+^{r+1}(w^r,z^r)) - \bar{x}^*(w^r,z^r)\|^2\big) \geq \vartheta\tau$, then we have the following results.

Recall the weak error bound

$$\|y^*(w,z;\lambda_+(w,z)) - \bar{y}^*(w,z)\|^2 + \|x^*(w,z;\lambda_+(w,z)) - \bar{x}^*(w,z)\|^2$$

$$\leq \sigma_{\text{weak}}\|\lambda - \lambda_+(w,z)\|\|\lambda(w,z) - \lambda_+(w,z)\| \tag{156}$$

where $\lambda(v) \in \arg\max_{0\leq\lambda\leq\Lambda} K(\bar{x}^*(v),\bar{y}^*(v),v;\lambda)$.

We can get

$$\|y^*(w^r,z^r;\lambda_+^{r+1}(w^r,z^r)) - \bar{y}^*(w^r,z^r)\|^2 + \|x^*(w^r,z^r;\lambda_+^{r+1}(w^r,z^r)) - \bar{x}^*(w^r,z^r)\|^2$$

$$\leq \underbrace{\sigma_{\text{weak}}2\Lambda}_{\triangleq\bar{\sigma}_{\text{w}}}\tau\left\|g(x^*(v^r;\lambda^{r+1}),y^*(v^r;\lambda^{r+1})) - g(x^*(v^r;\lambda^{r+1})) - \delta\right\|, \tag{157}$$

which gives

$$\|g(x^*(w^r,z^r;\lambda^{r+1}),y^*(w^r,z^r;\lambda^{r+1})) - g(x^*(w^r,z^r;\lambda^{r+1})) - \delta\| \leq 3\cdot 63\cdot 36p\zeta\bar{\sigma}_{\text{w}}. \tag{158}$$

Then, we can have

$$\frac{p}{4\beta}\|z^{r+1} - z^r\|^2$$

$$\leq 3 \cdot 36 p\zeta \left(\|y^*(w^r, z^r; \lambda_+^{r+1}(w^r, z^r)) - \bar{y}^*(w^r, z^r)\|^2 + \|x^*(w^r, z^r; \lambda_+^{r+1}(w^r, z^r)) - \bar{x}^*(w^r, z^r)\|^2\right)$$

$$\leq 3 \cdot 36 p\zeta\bar{\sigma}_{\mathrm{w}} \left\|g(x^*(w^r, z^r; \lambda^{r+1}), y^*(w^r, z^r; \lambda^{r+1})) - g(x^*(w^r, z^r; \lambda^{r+1})) - \delta\right\| \quad (159)$$

$$\leq 3^2 \cdot 63 \cdot 36^2 p^2 \zeta^2 \bar{\sigma}_{\mathrm{w}}^2. \quad (160)$$

Similarly,

$$\left\{\frac{1}{8\eta}\|x^{r+1} - x^r\|^2, \frac{1}{4\alpha}\|y^{r+1} - y^r\|^2, \frac{p}{4\beta}\|w^{r+1} - w^r\|^2\right\} \leq 3^2 \cdot 63 \cdot 36^2 p^2 \zeta^2 \bar{\sigma}_{\mathrm{w}}^2. \quad (161)$$

These results imply that the iterates generated by SLM will converge to some point within a ball with a radius of $\mathcal{O}(p^2\zeta^2\bar{\sigma}_{\mathrm{w}}^2)$.

Further, note that

$$\|w^r - x^*(w^r, z^r)\|^2 + \|z^r - y^*(w^r, z^r)\|^2$$

$$= \|w^r - x^{r+1} + x^{r+1} - x^*(w^r, z^r)\|^2 + \|z^r - y^{r+1} + y^{r+1} - y^*(w^r, z^r)\|^2$$

$$\leq \frac{2}{\beta^2}\left(\|w^{r+1} - w^r\|^2 + \|z^{r+1} - z^r\|^2\right) + 2\left(\|x^{r+1} - x^*(w^r, z^r)\|^2 + \|y^{r+1} - y^*(w^r, z^r)\|^2\right)$$

$$= \frac{2}{\beta^2}\left(\|w^{r+1} - w^r\|^2 + \|z^{r+1} - z^r\|^2\right)$$

$$\quad + 2\|x^{r+1} - x^*(w^r, z^r; \lambda^{r+1}) + x^*(w^r, z^r; \lambda^{r+1}) - x^*(w^r, z^r)\|^2$$

$$\quad + 2\|y^{r+1} - y^*(w^r, z^r; \lambda^{r+1}) + y^*(w^r, z^r; \lambda^{r+1}) - y^*(w^r, z^r)\|^2 \quad (162)$$

$$\leq \frac{2}{\beta^2}\left(\|w^{r+1} - w^r\|^2 + \|z^{r+1} - z^r\|^2\right)$$

$$\quad + 4\|x^{r+1} - x^*(w^r, z^r; \lambda^{r+1})\|^2 + 4\|x^*(w^r, z^r; \lambda^{r+1}) - x^*(w^r, z^r)\|^2$$

$$\quad + 4\|y^{r+1} - y^*(w^r, z^r; \lambda^{r+1})\|^2 + 4\|y^*(w^r, z^r; \lambda^{r+1}) - y^*(w^r, z^r)\|^2. \quad (163)$$

By following the steps from (136) to (138), we have

$$\|y^*(w^r, z^r; \lambda^{r+1}) - \bar{y}^*(w^r, z^r)\|^2 + \|x^*(w^r, z^r; \lambda^{r+1}) - \bar{x}^*(w^r, z^r)\|^2$$

$$\leq 2\|y^*(w^r, z^r; \lambda_+^{r+1}(w^r, z^r)) - \bar{y}^*(w^r, z^r)\|^2 + 2\|x^*(w^r, z^r; \lambda_+^{r+1}(w^r, z^r)) - \bar{x}^*(w^r, z^r)\|^2$$

$$\quad + 2\|y^*(w^r, z^r; \lambda^{r+1}) - y^*(w^r, z^r; \lambda_+^{r+1}(w^r, z^r))\|^2$$

$$\quad + 2\|x^*(w^r, z^r; \lambda^{r+1}) - x^*(w^r, z^r; \lambda_+^{r+1}(w^r, z^r))\|^2 \quad (164)$$

$$\leq 2\|y^*(w^r, z^r; \lambda_+^{r+1}(w^r, z^r)) - \bar{y}^*(w^r, z^r)\|^2 + 2\|x^*(w^r, z^r; \lambda_+^{r+1}(w^r, z^r)) - \bar{x}^*(w^r, z^r)\|^2$$

$$\quad + 4\sigma_2^2\tau^2 \left\|g(x^*(w^r, z^r; \lambda^{r+1}), y^*(w^r, z^r; \lambda^{r+1})) - g(x^*(w^r, z^r; \lambda^{r+1})) - \delta\right\|^2. \quad (165)$$

Combining (163) and (165) yields

$$\|w^r - x^*(w^r, z^r)\|^2 + \|z^r - y^*(w^r, z^r)\|^2 \quad (166)$$

$$\leq 4\left(\left(\frac{1 + \eta(p - L)}{\eta(p - L)}\right)^2 \|x^{r+1} - x^r\|^2 + \left(\frac{1 + \alpha(p - L)}{\alpha(p - L)}\right)^2 \|y^{r+1} - y^r\|^2\right)$$

$$\quad + \frac{2}{\beta^2}\left(\|w^{r+1} - w^r\|^2 + \|z^{r+1} - z^r\|^2\right)$$

$$\quad + 8\|y^*(w^r, z^r; \lambda_+^{r+1}(w^r, z^r)) - \bar{y}^*(w^r, z^r)\|^2 + 8\|x^*(w^r, z^r; \lambda_+^{r+1}(w^r, z^r)) - \bar{x}^*(w^r, z^r)\|^2$$

$$\quad + 16\sigma_2^2\tau^2 \left\|g(x^*(w^r, z^r; \lambda^{r+1}), y^*(w^r, z^r; \lambda^{r+1})) - g(x^*(w^r, z^r; \lambda^{r+1})) - \delta\right\|^2 \quad (167)$$

$$\leq 8\left(4\eta\left(\frac{1 + \eta(p - L)}{\eta(p - L)}\right)^2 + 2\alpha\left(\frac{1 + \alpha(p - L)}{\alpha(p - L)}\right)^2 + \frac{8}{p\beta}\right) 3^2 \cdot 63 \cdot 36^2 p^2 \zeta^2 \bar{\sigma}_{\mathrm{w}}^2$$

$$\quad + \left(8\bar{\sigma}_{\mathrm{w}}\tau + 16\sigma_2^2\tau^2 \cdot (3 \cdot 63 \cdot 36 p\zeta\bar{\sigma}_{\mathrm{w}})\right)(3 \cdot 63 \cdot 36 p\zeta\bar{\sigma}_{\mathrm{w}}). \quad (168)$$

Therefore, once

$$8\left(4\eta\left(\frac{1+\eta(p-L)}{\eta(p-L)}\right)^2 + 2\alpha\left(\frac{1+\alpha(p-L)}{\alpha(p-L)}\right)^2 + \frac{8}{p\beta}\right)3^2\cdot 63\cdot 36^2 p^2\zeta^2\bar\sigma_{\rm w}^2$$

$$+ \left(8\bar\sigma_{\rm w}\tau + 16\sigma_2^2\tau^2\cdot(3\cdot 63\cdot 36 p\zeta\bar\sigma_{\rm w})\right)(3\cdot 63\cdot 36 p\zeta\bar\sigma_{\rm w}) < \varrho\sim\mathcal{O}(1), \tag{169}$$

i.e., $\zeta$ is small, then, Lemma 8 holds automatically. We can simply choose

$$\zeta \le \sqrt{\frac{\varrho}{2}}\max\left\{\frac{1}{\zeta_1},\frac{1}{\zeta_2}\right\} \tag{170}$$

where

$$\zeta_1 = 3\cdot 36 p\bar\sigma_{\rm w}\sqrt{8\left(4\eta\left(\frac{1+\eta(p-L)}{\eta(p-L)}\right)^2 + 2\alpha\left(\frac{1+\alpha(p-L)}{\alpha(p-L)}\right)^2 + \frac{8}{p\beta}\right) + 63\cdot 16\sigma_2^2\tau^2}, \tag{171a}$$

$$\zeta_2 = 8\cdot 3\cdot 63\cdot 36 p\tau\bar\sigma_{\rm w}^2. \tag{171b}$$

Second, if $36 p\zeta\big(\|y^*(w^r,z^r;\lambda_+^{r+1}(w^r,z^r)) - \bar y^*(w^r,z^r)\|^2 + \|x^*(w^r,z^r;\lambda_+^{r+1}(w^r,z^r)) - \bar x^*(w^r,z^r)\|^2\big) \le \vartheta\tau$, then we have the following results.

$$\|w^r - x^*(w^r,z^r)\|^2 + \|z^r - y^*(w^r,z^r)\|^2$$

$$\le 12\left(\left(\frac{1+\eta(p-L)}{\eta(p-L)}\right)^2\cdot 8\eta + \left(\frac{1+\alpha(p-L)}{\alpha(p-L)}\right)^2\cdot 4\alpha\right)\vartheta\tau + \frac{48\vartheta\tau}{\beta p} + \frac{2\vartheta\tau}{9\zeta p} + 63\cdot 48\sigma_2^2\vartheta\tau^2. \tag{172}$$

It is obvious that when $\vartheta$ is small, then the RHS of (172) is less than constant $\varrho\sim\mathcal{O}(1)$, i.e., when

$$\vartheta \le \frac{\varrho}{\tau\zeta_3} \tag{173}$$

where

$$\zeta_3 = 12\left(\left(\frac{1+\eta(p-L)}{\eta(p-L)}\right)^2\cdot 8\eta + \left(\frac{1+\alpha(p-L)}{\alpha(p-L)}\right)^2\cdot 4\alpha\right) + \frac{48}{\beta p} + \frac{2}{9\zeta p} + 63\cdot 48\sigma_2^2\tau, \tag{174}$$

then, Lemma 8 holds automatically.

Given the strong error bound, substituting (103) into (141) yields

$$\mathcal{Q}^{r+1} - \mathcal{Q}^r$$

$$\le -\frac{1}{8\eta}\|x^{r+1} - x^r\|^2 - \frac{1}{4\alpha}\|y^{r+1} - y^r\|^2 - \frac{p}{4\beta}\|z^{r+1} - z^r\|^2 - \frac{p}{4\beta}\|w^{r+1} - w^r\|^2$$

$$- \tau\left(\frac{1}{63} - 12 p\zeta(3\sigma_{\rm strong}^2 + 4\sigma_2^2)\tau\right)|g(x^*(v^r;\lambda^{r+1})), y^*(v^r;\lambda^{r+1})) - g^*(x^*(v^r;\lambda^{r+1})) - \delta|^2$$

$$+ \left(\mu_g L_g^2 + 2\ell_g^2\right)\frac{D_{\mathcal{S}}}{r^2} - \frac{1}{4\tau}\|\lambda^{r+1} - \lambda^r\|^2. \tag{175}$$

Following (152), we also require

$$\frac{1}{63} - 12 p\zeta(3\sigma_{\rm strong}^2 + 4\sigma_2^2)\tau) > \frac{1}{126} \tag{176}$$

i.e.,

$$\tau < \frac{1}{126\cdot 12 p\zeta(3\sigma_{\rm strong}^2 + 4\sigma_2^2)} < \frac{1}{126\cdot 48 p\zeta\max\{\sigma_{\rm strong}^2,\sigma_2^2\}}. \tag{177}$$

When the conditions shown in (146), (148), (150), (152), and (177) are satisfied, we can obtain

$$\mathcal{Q}^{r+1} - \mathcal{Q}^r$$

$$\le -\frac{1}{8\eta}\|x^{r+1} - x^r\|^2 - \frac{1}{4\alpha}\|y^{r+1} - y^r\|^2 - \frac{p}{4\beta}\|z^{r+1} - z^r\|^2 - \frac{p}{4\beta}\|w^{r+1} - w^r\|^2$$

$$- \frac{\tau}{126}|g(x^*(v^r;\lambda^{r+1})), y^*(v^r;\lambda^{r+1})) - g^*(x^*(v^r;\lambda^{r+1})) - \delta|^2$$

$$+ \left(\mu_g L_g^2 + 2\ell_g^2\right)\frac{D_{\mathcal{S}}}{r^2} - \frac{1}{4\tau}\|\lambda^{r+1} - \lambda^r\|^2. \tag{178}$$

This completes the proof. $\qquad\square$

### B.6 Equivalence of KKT Points between the Saddle Point Problem and Constrained Optimization

**Lemma 10.** *Consider the following two problems:*

$$\text{(P1)}: \quad \min_{x \in \mathcal{X}, y, g(x,y)-g^*(x) \leq \delta} f(x,y) + \frac{p}{2}\|y-z\|^2 + \frac{p}{2}\|x-w\|^2, \tag{179a}$$

$$\text{(P2)}: \quad \min_{x \in \mathcal{X}, y} \max_{0 \leq \lambda \leq \Lambda} f(x,y) + \lambda(g(x,y)-g^*(x)-\delta) + \frac{p}{2}\|y-z\|^2 + \frac{p}{2}\|x-w\|^2. \tag{179b}$$

*Let $x^*, y^*, w^*, z^*, \lambda^*$ denote the KKT solutions of P2. If $\lambda^* \neq \Lambda$, then the KKT solutions of P2 are also the KKT solutions of P1.*

*Proof.* It is easy to check that the primal stationary conditions of P1 and P2 are the same. Regarding the stationarity of dual variable $\lambda$, we can apply the saddle point theorem. Note that problem P2 is strongly convex w.r.t. $y$ and concave w.r.t. $y$ and both feasible sets of $x, y$ are compact. Then, there exists a saddle point such that

$$\lambda^*(g(x^*, y^*) - g^*(x^*) - \delta) \geq \lambda(g(x^*, y^*) - g^*(x^*) - \delta), \tag{180}$$

which gives $\lambda^*$ in following three cases

$$\lambda^* = \begin{cases} \Lambda & \text{if } g(x^*, y^*) - g^*(x^*) - \delta > 0, \\ \lambda & \text{if } g(x^*, y^*) - g^*(x^*) - \delta = 0, \\ 0 & \text{if } g(x^*, y^*) - g^*(x^*) - \delta < 0. \end{cases} \tag{181}$$

If $\lambda^* \neq \Lambda$, we can obtain

$$\lambda^* \geq 0, \quad \lambda^*(g(x^*, y^*) - g^*(x^*) - \delta) = 0, \tag{182}$$

which are the KKT conditions of P1. $\qquad\square$

Given this result, we can know that $P(w, z)$ defined in (10b) becomes

$$P(w, z) = \min_{x \in \mathcal{X}, y, g(x,y)-g^*(x) \leq \delta} f(x,y) + \frac{p}{2}\|x-w\|^2 + \frac{p}{2}\|y-z\|^2, \tag{183}$$

if $0 \leq \lambda < \Lambda$. Therefore, when the iterates converge, if the dual variable is strictly less than $\Lambda$, the converged point is the primal-dual (KKT) solution of the original problem.

## C  Boundedness of Dual Variable, LL Variables, and Potential Function

### C.1  Boundedness of $\mathcal{Q}^r$

From (13), we know that

$$\begin{aligned} &\mathcal{Q}^r(x^r, y^r, w^r, z^r; \lambda^r) \\ &= K(x^r, y^r, w^r, z^r; \lambda^r) - 2D(w^r, z^r; \lambda^r) + 2P(w^r, z^r) \qquad\qquad (184) \\ &= P(w^r, z^r) + K(x^r, y^r, w^r, z^r; \lambda^r) - D(w^r, z^r; \lambda^r) + (P(w^r, z^r) - D(w^r, z^r; \lambda^r)) \quad (185) \\ &\overset{(a)}{\geq} P(w^r, z^r) \geq \underline{f} \qquad\qquad (186) \end{aligned}$$

where $(a)$ holds due to 1) $K(x^r, y^r, w^r, z^r; \lambda^r) - D(w^r, z^r; \lambda^r) \geq 0$ based on the definition of $D(w^r, z^r; \lambda^r)$ and 2) note that $P(w, z) = \min_{x \in \mathcal{X}, y} \max_{0 \leq \lambda \leq \Lambda} f(x, y) + \lambda(g(x, y) - g^*(x) - \delta) + \frac{p}{2}\|x-w\|^2 + \frac{p}{2}\|y-z\|^2$ and $P(w^r, z^r) - D(w^r, z^r; \lambda^r) \geq 0$, which is true because the minimax equality theorem [49, 50] holds when $K(x, y, w, z; \lambda)$ is strongly convex in $x, y$ and linear (concave) in $\lambda$.

### C.2  Boundedness of Dual Variable

**Lemma 11.**

*Under A1-A5, suppose that the sequence $\{x^r, y^r, w^r, z^r, u^r, \lambda^r, \forall r\}$ is generated by SLM. Assume that $\lambda^r \sim \mathcal{O}(1/\sqrt{\delta}), y^r \sim \mathcal{O}(1)$ are bounded. When $0 < \delta < 1$, choose $p = \Theta(\lambda^{r+1})$, $\tau = \mathcal{O}(1), \alpha = \mathcal{O}(1/\lambda^{r+1}), \zeta, \beta = \Theta(\delta^{1.5})$ such that $p > L, \alpha \le 1/L_K, \beta \le 1$, then, $\lambda^{r+1}$ will either be upper bounded by $\Lambda$ for at most $\mathcal{O}(1/\delta^{2.5})$ steps or there exists a constant $c$ such that*

$$\lambda^{r+1} \le \frac{c\ell_f}{\sqrt{\delta}} \sim \mathcal{O}\left(\frac{1}{\sqrt{\delta}}\right), \quad \Lambda > \frac{c\ell_f}{\sqrt{\delta}}. \tag{187}$$

*In addition, when $\delta > 1$, choose $\beta = \mathcal{O}(1)$, then, $\lambda^{r+1} = \mathcal{O}(1)$.*

*Proof.* From (7b), we have

$$y^{r+1} = y^r - \alpha \left(\nabla_y f(x^r, y^r) + \lambda^{r+1} \nabla_y g(x^r, y^r) + p(y^r - z^r)\right). \tag{188}$$

Applying the triangle inequality gives

$$\|\lambda^{r+1} \nabla_y g(x^r, y^r)\| \le \frac{1}{\alpha} \|y^{r+1} - y^r\| + \|\nabla_y f(x^r, y^r)\| + p\|y^r - z^r\|. \tag{189}$$

If $\lambda^{r+1} < \lambda^r$, then $\lambda^{r+1}$ is bounded immediately. Otherwise, note that

$$\|\lambda^{r+1} \nabla_y g(x^r, y^r)\|$$

$$\overset{(a)}{\ge} \lambda^{r+1} \sqrt{2\mu_g(g(x^r, y^r) - g(x^r, y^*(x^r)))} \tag{190}$$

$$= \lambda^{r+1} \sqrt{2\mu_g(g(x^r, y^r) - g(x^r, u^r) + g(x^r, u^r) - g(x^r, y^*(x^r)))} \tag{191}$$

$$\overset{(b)}{=} \lambda^{r+1} \sqrt{\frac{2\mu_g}{\tau}(\lambda^{r+1} - \lambda^r) + \delta + g(x^r, u^r) - g(x^r, y^*(x^r))} \tag{192}$$

where $(a)$ holds due to the PŁ condition and nonnegativity of $\lambda$, in $(b)$ we use the rule of the dual variable update.

Then, we can obtain

$$\lambda^{r+1} \le \frac{\frac{1}{\alpha}\|y^{r+1} - y^r\| + \|\nabla_y f(x^r, y^r)\| + p\|y^r - z^r\|}{\sqrt{\frac{2\mu_g}{\tau}(\lambda^{r+1} - \lambda^r) + \delta + g(x^r, u^r) - g(x^r, y^*(x^r))}} \tag{193}$$

$$\overset{(a)}{\le} \frac{\frac{1}{\alpha}\|y^{r+1} - y^r\| + \|\nabla_y f(x^r, y^r)\| + p\|y^r - z^r\|}{\sqrt{\frac{2\mu_g}{\tau}(\lambda^{r+1} - \lambda^r) + \delta}} \tag{194}$$

where $(a)$ follows from the definition $g^*(x^r)$, i.e., $g(x^r, u^r) \ge g(x^r, y^*(x^r))$. As $\lambda^{r+1} > \lambda^r$, we can further have

$$\lambda^{r+1} \le \frac{\frac{1}{\alpha}\|y^{r+1} - y^r\| + \|\nabla_y f(x^r, y^r)\| + p\|y^r - z^r\|}{\sqrt{\delta}}. \tag{195}$$

Note that the LHS of (153) decreases on the order of $(\mathcal{Q}^1 - \underline{f})/r$, with the lower bound of this quantity being $\vartheta\tau$. Therefore, according to (153) and (155), we can know that the iterates will be either in Case 1 for a certain number of iterations (i.e., at most $\mathcal{O}((\mathcal{Q}^1 - \underline{f})/(\vartheta\tau))$ steps) or in Case 2. In Case 1, the projection operation (i.e., $\mathcal{P}_+[\cdot]$) can always ensure that $\lambda^{r+1}$ is bounded by $\Lambda$ so that we can choose $p > L$. In the following, it can be seen that the dual variable is automatically bounded, implying that the projection of the iterates onto the upper bound will be always inactive when the algorithm is in Case 2 (i.e., $\lambda^r < \Lambda$).

In Case 2, when $0 < \delta < 1$ and the following conditions are met

$$\eta, \alpha = \mathcal{O}\left(\frac{1}{\lambda^r}\right), \quad p = \Theta(\lambda^r), \quad \beta = \mathcal{O}(\delta^{1.5}), \quad \tau = \Theta(1), \tag{196}$$

and assuming that $\lambda^r = \mathcal{O}(1/\sqrt{\delta})$, we can have $\bar{\sigma}_w = \mathcal{O}(1)$ due to (157), $\vartheta = \mathcal{O}(\delta^2)$ due to (153), $\mathcal{Q}^1 = \mathcal{O}(1/\sqrt{\delta})$, and

$$\|y^r - z^r\|^2 \overset{(198)}{=} \mathcal{O}(\|y^{r+1} - y^r\|^2 + \|z^{r+1} - z^r\|^2/\beta^2) \overset{(161)}{\leq} \mathcal{O}(\delta), \tag{197a}$$

$$\|y^{r+1} - y^r\|^2 \overset{(161)}{\leq} \mathcal{O}(\delta^2) \tag{197b}$$

where we use

$$\|y^r - y^{r+1} + y^{r+1} - z^r\| \leq \|y^{r+1} - y^r\| + \frac{1}{\beta}\|z^{r+1} - z^r\|. \tag{198}$$

Then, (195) can be written as

$$\lambda^{r+1} \leq \frac{\lambda^{r+1}\|y^{r+1} - y^r\|}{\alpha_0\sqrt{\delta}} + \frac{\ell_f}{\sqrt{\delta}} + \frac{p_0\lambda^{r+1}\|y^r - z^r\|}{\sqrt{\delta}} \tag{199}$$

where we choose

$$p = p_0\lambda^{r+1}, \quad \alpha = \frac{\alpha_0}{\lambda^{r+1}}. \tag{200}$$

By utilizing the step-sizes as specified in (200) and substituting (197) back to (195), it can be easily checked that the first term of (199) is $\mathcal{O}(\delta^{1/2}\lambda^{r+1}/\alpha_0)$, and the last term is $\mathcal{O}(p_0\lambda^{r+1})$. From (161), it can be seen that when $\zeta$ is small (or equivalently $\beta_0$ is small, where $\beta \overset{(150)}{=} \beta_0\delta^{1.5}$), there must exist a constant $c$ such that

$$\lambda^{r+1} \leq \frac{c\ell_f}{\sqrt{\delta}} \sim \mathcal{O}\left(\frac{1}{\sqrt{\delta}}\right). \tag{201}$$

When $\delta > 1$, it is easy to check that when $\beta, \tau = \mathcal{O}(1), p = \Omega(\lambda^{r+1})$, we have $\lambda^{r+1} = \mathcal{O}(1)$.

$\square$

Given this result and Lemma 10, we can set the tunable variable $\Lambda$ as $\Theta(1/\sqrt{\delta})$. After running the algorithm for $T$ steps, if the last dual variable is $\Lambda$, i.e., the projection of the last iterate on $\Lambda$ is active, then, we need to increase $\Lambda$ and regenerate the sequence.

### C.3 Boundedness of $y^r$, $\bar{y}^*(w^r, z^r)$, and $y^*(w^r, z^r; \lambda^{r+1})$

**Lemma 12.**

*Under A1-A5 and the local regularity condition, suppose that the sequence $\{x^r, y^r, w^r, z^r, u^r, \lambda^r, \forall r\}$ is generated by SLM. Assume that $y^r, \bar{y}^*(w^r, z^r)$ are bounded. Then, we have $\bar{y}^*(w^{r+1}, z^{r+1}), y^*(w^r, z^r; \lambda^{r+1}), y^{r+1}$ are also bounded.*

*Proof.* We prove these results by induction. First, we assume that $y^r, \bar{y}^*(w^r, z^r)$ are bounded, which gives the (gradient) Lipschitz continuity of $g(x, \cdot)$ at these points.

Let $\Delta R^r \triangleq \mu_g(L_g^2 + \ell_g^2)D_{\mathcal{S}}/r^2, r \geq 1$. Recall the bounded set level set assumption that let

$$\psi(x, y) = f(x, y), x \in \mathcal{X}, y \in \{y|g(x, y) - g^*(x) \leq \delta\}. \tag{202}$$

Under the assumption that $y^r, \bar{y}^*(w^r, z^r)$ are bounded, we can have the descent of the potential function up to $\Delta R^r$ that is a decreasing sequence. By the fact that $\psi(\bar{x}^*(w^{r+1}, z^{r+1}), \bar{y}^*(w^{r+1}, z^{r+1})) \leq P(w^{r+1}, z^{r+1})$, for any $(x^1, y^1, w^1, z^1; \lambda^1)$, there exists a constant $R$ such that

$$\{\bar{y}^*(w^{r+1}, z^{r+1})|P(w^{r+1}, z^{r+1}) \leq \mathcal{Q}^{r+1} + \Delta R^1\} \subseteq \mathcal{B}(R(x^1, y^1, z^1; \lambda^1)) \tag{203}$$

which gives that $\bar{y}^*(w^{r+1}, z^{r+1})$ is bounded.

Applying (14) shown in Lemma 2 gives

$$\|y^*(x^r, z^r; \lambda_+^{r+1}(w^r, z^r)) - \bar{y}^*(w^r, z^r)\|$$

$$\overset{(14)}{\leq} \sqrt{\sigma_{\text{weak}} \|\lambda^{r+1} - \lambda_+^{r+1}(w^r, z^r)\| \|\lambda(w^r, z^r) - \lambda_+^{r+1}(w^r, z^r)\|} \tag{204}$$

$$\overset{(157)}{\leq} \mathcal{O}\left(\sqrt{\bar{\sigma}_{\text{w}} \tau \|g(x^*(w^r, z^r; \lambda^{r+1}), y^*(w^r, z^r; \lambda^{r+1})) - g(x^*(w^r, z^r; \lambda^{r+1})) - \delta\|}\right) \tag{205}$$

$$\overset{(a)}{\leq} \mathcal{O}(\sqrt{\delta}) \tag{206}$$

where in $(a)$ we choose $\tau = \mathcal{O}(1)$ when $\delta > 1$, so we have $\lambda, \bar{\sigma}_{\text{w}} = \mathcal{O}(1)$, and we choose step-sizes according to (196) and also apply (158) when $0 < \delta \leq 1$. So, we can have that $y^*(w^r, z^r; \lambda_+^{r+1}(w^r, z^r)) \sim \mathcal{O}(\sqrt{\delta})$ is bounded. Also, it can be checked that

$$\|y^*(w^r, z^r; \lambda^{r+1}) - y^*(x^r, z^r; \lambda_+^{r+1}(w^r, z^r))\|$$

$$\overset{(28a)}{\leq} \frac{p+L}{p-L} \|\lambda^{r+1} - \lambda_+^{r+1}(w^r, z^r)\| \tag{207}$$

$$\overset{(a)}{\leq} \frac{p+L}{p-L} \tau \|g(x^*(v^r; \lambda^{r+1}), y^*(v^r; \lambda^{r+1})) - g^*(x^*(v^r; \lambda^{r+1})) - \delta)\|$$

$$\overset{(155)}{\leq} \mathcal{O}(\tau\delta) \sim \mathcal{O}(1) \tag{208}$$

where $(a)$ holds due to the nonexpansiveness of the nonnegative projection operator. Therefore, $y^*(w^r, z^r; \lambda^{r+1})$ is bounded.

Note that $K(x, \cdot, z; \lambda^{r+1})$ is strongly convex with modulus $p - L$ and gradient Lipschitz continuous with parameter $p + L$. From [51, Theorem 3.5.], we have

$$\|y^{r+1} - y^*(w^r, z^r; \lambda^{r+1})\| \leq \left(1 - \frac{p-L}{p+L}\right) \|y^r - y^*(w^r, z^r; \lambda^{r+1})\|, \tag{209}$$

which directly gives the boundedness of $y^{r+1}$.

$\square$

# D  Theoretical Convergence Results

## D.1  Proof of Theorem 1

*Proof.* **KKT Conditions**.

For $x, y, w, z, \lambda$, we define $\mathcal{F}$ as a map such that $\mathcal{F}(x, y, w, z, u, \lambda) = (x^+, y^+, w^+, z^+, u^+, \lambda^+)$, where $(x^+, y^+, w^+, z^+, u^+, \lambda^+)$ is the next iteration of Algorithm 1. It can be easily checked that the map $\mathcal{F}$ is continuous and if $x, y, w, z, u, \lambda$ is a fixed point of $\mathcal{F}$, i.e., $\mathcal{F}(x, y, w, z, u, \lambda) = (x, y, w, z, u, \lambda)$, then $(x, y, \lambda)$ is a pair of primal-dual stationary solution of problem (1). Suppose that

$$(x^r, y^r, w^r, z^r, u^r, \lambda^r) \to (\bar{x}, \bar{y}, \bar{w}, \bar{z}, \bar{u}, \bar{\lambda}) \text{ along a subsequence } r \in \mathcal{T}. \tag{210}$$

Notice the lower boundedness of $\mathcal{Q}^r$ shown in (186) and Lemma 9, we can get

$$\|x^{r+1} - x^r\| \to 0, \quad \|y^{r+1} - y^r\| \to 0, \quad \|z^{r+1} - z^r\| \to 0 \tag{211}$$

and

$$\|g(x^*(w^r, z^r; \lambda^{r+1}), y^*(w^r, z^r; \lambda^{r+1})) - g^*(x^*(w^r, z^r; \lambda^{r+1})) - \delta\| \to 0, \; |g(x^r, u^r) - g^*(x^r)| \to 0,$$

which gives $\|x^{r+1} - w^r\| \to 0, \|y^{r+1} - z^r\| \to 0, \|\lambda^{r+1} - \lambda^r\| \to 0$, and further implies

$$\left\|(x^{r+1}, y^{r+1}, w^{r+1}, z^{r+1}, u^{r+1}, \lambda^{r+1}) - (x^r, y^r, w^r, z^r, u^r, \lambda^r)\right\| \to 0. \tag{212}$$

Therefore, we obtain

$$\|\mathcal{F}(\bar{x}, \bar{y}, \bar{w}, \bar{z}, \bar{u}, \bar{\lambda}) - (\bar{x}, \bar{y}, \bar{w}, \bar{z}, \bar{u}, \bar{\lambda})\|$$

$$\overset{(a)}{=} \lim_{t \to \infty, t \in \mathcal{T}} \|(x^r, y^r, w^r, z^r, u^r, \lambda^r) - \mathcal{F}(x^r, y^r, w^r, z^r, u^r, \lambda^r)\| \tag{213}$$

$$\overset{(b)}{=} \lim_{t \to \infty, t \in \mathcal{T}} \|(x^{r+1}, y^{r+1}, w^{r+1}, z^{r+1}, u^{r+1}, \lambda^{r+1}) - \mathcal{F}(x^r, y^r, w^r, z^r, u^r, \lambda^r)\| \tag{214}$$

$$= 0$$

where $(a)$ holds due to the continuity of $\mathcal{F}$ and $(b)$ follows from (212). Hence, $(\bar{x}, \bar{y}, \bar{\lambda})$ satisfies

$$\text{stationarity}: \quad \left\| \begin{matrix} \frac{1}{\eta}\left(\bar{x} - \mathcal{P}_{\mathcal{X}}(\bar{x} - \eta \nabla_x \mathcal{L}(\bar{x}, \bar{y}; \bar{\lambda}))\right) \\ \nabla_y \mathcal{L}(\bar{x}, \bar{y}; \bar{\lambda}) \end{matrix} \right\| = 0, \quad \bar{x} \in \mathcal{X}, \forall \eta > 0, \tag{215a}$$

$$\text{feasibility}: \quad g(\bar{x}, \bar{y}) - g^*(\bar{x}) - \delta \leq 0, \bar{\lambda} \geq 0, \tag{215b}$$

$$\text{complementary slackness}: \quad (g(\bar{x}, \bar{y}) - g^*(\bar{x}) - \delta)\bar{\lambda} = 0, \tag{215c}$$

i.e., every limit point $(\bar{x}, \bar{y}, \bar{\lambda})$ is a primal-dual stationary solution (KKT point) of problem (1).

**Stationarity of Primal Variables**. Recall

$$\mathcal{G}(x^r, y^r) = \begin{bmatrix} \frac{1}{\eta}\left(x^r - \mathcal{P}_{\mathcal{X}}(x^r - \eta \nabla_x \mathcal{L}(x^r, y^r; \lambda^r))\right) \\ \nabla_y \mathcal{L}(x^r, y^r; \lambda^r) \end{bmatrix}. \tag{216}$$

For the block-$x$, we have

$$\left\| \frac{1}{\eta}\left(x^r - \mathcal{P}_{\mathcal{X}}(x^r - \eta \nabla_x \mathcal{L}(x^r, y^r; \lambda^r))\right) \right\|$$

$$\leq \frac{1}{\eta}\left(\|x^{r+1} - x^r\| + \|x^{r+1} - \mathcal{P}_{\mathcal{X}}(x^r - \eta(\nabla_x f(x^r, y^r) + \lambda^r(\nabla_x g(x^r, y^r) - \nabla_x g^*(x^r)))\|\right)$$

$$\overset{(a)}{\leq} \frac{1}{\eta}\left(3\|x^{r+1} - x^r\| + \eta\|\nabla_x f(x^r, y^r) - \nabla_x f(x^r, y^{r+1})\| + \eta\lambda^{r+1}\|\nabla_x g(x^r, y^r) - \nabla_x g(x^r, y^{r+1})\|\right)$$

$$+ \lambda^{r+1}\|\nabla_x g(x^r, u^{r+1})) - \nabla_x g^*(x^r)\| + |\lambda^{r+1} - \lambda^r|\|\nabla_x g(x^r, y^r) - \nabla_x g^*(x^r)\| + p\|x^r - w^r\|$$

$$\leq \left(\frac{3}{\eta} + p\right)\|x^{r+1} - x^r\| + \frac{p}{\beta}\|w^{r+1} - w^r\|$$

$$+ (L_f + L_g\lambda^{r+1})\|y^{r+1} - y^r\| + \lambda^{r+1}L_g d_{\mathcal{S}(x^r)}(u^{r+1}) + 2\ell_g|\lambda^{r+1} - \lambda^r| \tag{217}$$

$$\overset{(b)}{\leq} \left(\frac{3}{\eta} + p\right)\|x^{r+1} - x^r\| + \frac{p}{\beta}\|w^{r+1} - w^r\| + (L_f + L_g\lambda^{r+1})\|y^{r+1} - y^r\|$$

$$+ 2\ell_g\tau|g(x^r, y^r) - g(x^r, u^{r+1}) - \delta| + \lambda^{r+1}L_g d_{\mathcal{S}(x^r)}(u^{r+1}) \tag{218}$$

$$\overset{(c)}{\leq} \left(\frac{3}{\eta} + p + \frac{2\ell_g^2\tau}{\eta(p - L)}\right)\|x^{r+1} - x^r\| + \frac{p}{\beta}\|w^{r+1} - w^r\|$$

$$+ \left(L_f + L_g\lambda^{r+1} + \frac{\ell_g^2\tau}{\alpha(p - L)}\right)\|y^{r+1} - y^r\|$$

$$+ 2\ell_g\tau|g(x^*(w^r, z^r; \lambda^{r+1}), y^*(w^r, z^r; \lambda^{r+1})) - g^*(x^*(w^r, z^r; \lambda^{r+1})) - \delta|$$

$$+ \left(2\ell_g\tau + \lambda^{r+1}L_g\right)d_{\mathcal{S}(x^r)}(u^{r+1}) \tag{219}$$

where in $(a)$ we apply the following optimality condition of the $x$-subproblem

$$x^{r+1} = \mathcal{P}_{\mathcal{X}}\left[x^{r+1} - \eta\left(\nabla_x f(x^r, y^{r+1})\right.\right.$$

$$\left.\left. + \lambda^{r+1}(\nabla_x g(x^r, y^{r+1}) - \nabla_x g(x^r, u^{r+1})) + p(x^r - w^r) + \frac{1}{\eta}(x^{r+1} - x^r)\right)\right],$$

$(b)$ results from the nonexpansiveness of the nonnegative projection operator, and $(c)$ holds due to

$$|g(x^*(w^r, z^r; \lambda^{r+1}), y^*(w^r, z^r; \lambda^{r+1})) - g(x^r, y^r)|$$
$$\leq \frac{\ell_g}{p-L} \left( \frac{1}{\alpha} \|y^{r+1} - y^r\| + \frac{1}{\eta} \|x^{r+1} - x^r\| \right), \tag{220}$$

and

$$|g^*(x^*(w^r, z^r; \lambda^{r+1})) - g^*(x^r)| \leq \frac{\ell_g}{\eta(p-L)} \|x^{r+1} - x^r\|. \tag{221}$$

For the block-$y$, we have

$$\|\nabla_y \mathcal{L}(x^r, y^r; \lambda^r)\|$$
$$\leq \|\nabla_y f(x^r, y^r) + \lambda^r \nabla_y g(x^r, y^r)\| \tag{222}$$
$$\overset{(a)}{\leq} \frac{1}{\alpha} \|y^{r+1} - y^r\| + \ell_g |\lambda^{r+1} - \lambda^r| + p\|y^r - z^r\| \tag{223}$$
$$\overset{(7c)}{\leq} \left( \frac{1}{\alpha} + p \right) \|y^{r+1} - y^r\| + \ell_g |\lambda^{r+1} - \lambda^r| + \frac{p}{\beta} \|z^{r+1} - z^r\| \tag{224}$$
$$\overset{(b)}{\leq} \left( \frac{1}{\alpha} + p \right) \|y^{r+1} - y^r\| + \ell_g \tau |g(x^r, y^r) - g(x^r, u^{r+1}) - \delta| + \frac{p}{\beta} \|z^{r+1} - z^r\| \tag{225}$$
$$\leq \left( \frac{1}{\alpha} + p + \frac{\ell_g^2 \tau}{\alpha(p-L)} \right) \|y^{r+1} - y^r\| + \frac{2\ell_g^2 \tau}{\eta(p-L)} \|x^{r+1} - x^r\| + \frac{p}{\beta} \|z^{r+1} - z^r\|$$
$$+ \ell_g \tau |g(x^*(w^r, z^r; \lambda^{r+1}), y^*(w^r, z^r; \lambda^{r+1})) - g^*(x^*(w^r, z^r; \lambda^{r+1})) - \delta|$$
$$+ \ell_g \tau d_{\mathcal{S}(x^r)}(u^{r+1}) \tag{226}$$

where in $(a)$ we apply the following optimality condition of the $y$-subproblem

$$\nabla_y f(x^r, y^r) + \lambda^{r+1} \nabla_y g(x^r, y^r) + p(y^r - z^r) + \frac{1}{\alpha}(y^{r+1} - y^r) = 0, \tag{227}$$

and $(b)$ comes from the update rule of the dual variable, i.e., the second line of Algorithm 1

Therefore, the primal optimality gap can be quantified as follows:

$$\|\mathcal{G}(x^r, y^r)\|^2$$
$$\leq 5 \left( \left( \frac{3}{\eta} + p + \frac{2\ell_g^2 \tau}{\eta(p-L)} \right)^2 + \left( \frac{2\ell_g^2 \tau}{\eta(p-L)} \right)^2 \right) \|x^{r+1} - x^r\|^2$$
$$+ 5 \left( \left( L_f + L_g \lambda^{r+1} + \frac{2\ell_g^2 \tau}{\alpha(p-L)} \right)^2 + \left( \frac{1}{\alpha} + p + \frac{\ell_g^2 \tau}{\alpha(p-L)} \right)^2 \right) \|y^{r+1} - y^r\|^2$$
$$+ \frac{5p^2}{\beta^2} \|w^{r+1} - w^r\|^2 + \frac{5p^2}{\beta^2} \|z^{r+1} - z^r\|^2$$
$$+ 25\ell_g^2 \tau^2 |g(x^*(w^r, z^r; \lambda^{r+1}), y^*(w^r, z^r; \lambda^{r+1})) - g^*(x^*(w^r, z^r; \lambda^{r+1})) - \delta|^2$$
$$+ 5 \left( (2\ell_g \tau + \lambda^{r+1} L_g)^2 + \ell_g^2 \tau^2 \right) d_{\mathcal{S}(x^r)}^2(u^{r+1}). \tag{228}$$

From (144), we have the following three inequalities.

The first one is

$$\min \left\{ \frac{1}{24\eta}, \frac{1}{12\alpha} \right\} \left( \|x^{r+1} - x^r\|^2 + \|y^{r+1} - y^r\|^2 \right) \leq \mathcal{Q}^r - \mathcal{Q}^{r+1} + \left( \mu_g L_g^2 + 2\ell_g^2 \right) \frac{D_{\mathcal{S}}}{r^2}. \tag{229}$$

The second one is

$$\frac{p}{12\beta} \left( \|w^{r+1} - w^r\|^2 + \|z^{r+1} - z^r\|^2 \right) \leq \mathcal{Q}^r - \mathcal{Q}^{r+1} + \left( \mu_g L_g^2 + 2\ell_g^2 \right) \frac{D_{\mathcal{S}}}{r^2}. \tag{230}$$

The third one is

$$\frac{\tau}{189}|g(x^*(v^r;\lambda^{r+1})), y^*(v^r;\lambda^{r+1})) - g^*(x^*(v^r;\lambda^{r+1})) - \delta|^2$$

$$\leq \mathcal{Q}^r - \mathcal{Q}^{r+1} + \left(\mu_g L_g^2 + 2\ell_g^2\right)\frac{D_\mathcal{S}}{r^2}. \tag{231}$$

Then, we let

$$\rho_1 \triangleq \min\left\{\frac{1}{24\eta}, \frac{1}{12\alpha}\right\} \tag{232a}$$

$$\rho_2 \triangleq \max\left\{5\left(\left(\frac{3}{\eta} + p + \frac{2\ell_g^2\tau}{\eta(p-L)}\right)^2 + \left(\frac{2\ell_g^2\tau}{\eta(p-L)}\right)^2\right),\right.$$

$$\left. 5\left(\left(L_f + L_g\lambda^{r+1} + \frac{2\ell_g^2\tau}{\alpha(p-L)}\right)^2 + \left(\frac{1}{\alpha} + p + \frac{\ell_g^2\tau}{\alpha(p-L)}\right)^2\right)\right\}. \tag{232b}$$

Plugging (229), (230), and (231) into (228) along with (232a) and (232b), we can have

$$\|\mathcal{G}(x^r, y^r)\|^2 \leq \left(\frac{\rho_2}{\rho_1} + \frac{60p}{\beta} + 25 \cdot 189\ell_g^2\tau\right)\left(\mathcal{Q}^r - \mathcal{Q}^{r+1} + \left(\mu_g L_g^2 + 2\ell_g^2\right)\frac{D_\mathcal{S}}{r^2}\right)$$

$$+ 5\left(\left(2\ell_g\tau + \lambda^{r+1}L_g\right)^2 + \ell_g^2\tau^2\right)d_{\mathcal{S}(x^r)}^2(u^{r+1}).$$

Note that according to (32) we have

$$L_g^2(\lambda^{r+1})^2\left(1 - 2\gamma\mu_g\right)^{T^r}D_\mathcal{S} \leq \frac{L_g^2 D_\mathcal{S}}{r^2}, \tag{233}$$

when $T^r \geq -2\log_{1-2\gamma\mu_g}\lambda^{r+1}r$.

Applying the telescoping sum over $r = 1, \ldots, T$ yields

$$\frac{1}{T}\sum_{r=1}^T \|\mathcal{G}(x^r)\|^2$$

$$\leq \frac{1}{T}\left(\frac{\rho_2}{\rho_1} + \frac{60p}{\beta} + 25 \cdot 189\ell_g^2\tau\right)\sum_{r=1}^T\left(\mathcal{Q}^r - \mathcal{Q}^{r+1}\right)$$

$$+ \frac{1}{T}\left(\frac{\rho_2}{\rho_1} + \frac{60p}{\beta} + 25 \cdot 189\ell_g^2\tau\right)\left(\mu_g L_g^2 + 2\ell_g^2\right)\sum_{r=1}^T\frac{D_\mathcal{S}}{r^2} + \left(10 + 45\ell_g^2\tau^2\right)\sum_{r=1}^T\frac{D_\mathcal{S}}{r^2}$$

$$\overset{(a)}{\leq} \frac{1}{T}\left(\frac{\rho_2}{\rho_1} + \frac{60p}{\beta} + 25 \cdot 189\ell_g^2\tau\right)\left(\mathcal{Q}^1 - \underline{f}\right)$$

$$+ \frac{1}{T}\left(\left(\frac{\rho_2}{\rho_1} + \frac{60p}{\beta} + 25 \cdot 189\ell_g^2\tau\right)\left(\mu_g L_g^2 + 2\ell_g^2\right) + \left(10 + 45\ell_g^2\tau^2\right)\right)D_\mathcal{S}$$

where $(a)$ results from the fact $\sum_{r=1}^T r^{-2} \leq 1 + \int_1^T x^{-2}dx = 2 - T^{-1}$.

We can obtain the convergence rate based on the following two cases: 1) $\delta > 1$ and 2) $\delta \leq 1$.

If $\delta > 1$, we have $\lambda^{r+1} = \mathcal{O}(1)$. We choose

$$\eta, \alpha = \mathcal{O}\left(\frac{1}{\lambda^{r+1}}\right), \ p = \Theta(\lambda^{r+1}), \ \gamma, \tau = \mathcal{O}(1), \ \zeta, \beta = \mathcal{O}\left(\frac{\delta}{p}\right), \ T^r = \Omega(\log\lambda^{r+1}). \tag{234}$$

such that the step-sizes satisfy (143). Note that as $\zeta$ satisfies (169) and (173), it leads to $\varrho = \mathcal{O}(\sqrt{\delta})$ and the strong error bound being true automatically.

If $0 < \delta \leq 1$, we choose

$$\eta, \alpha = \mathcal{O}\left(\frac{1}{\lambda^{r+1}}\right), \ p = \Theta\left(\lambda^{r+1}\right), \ \zeta, \beta = \Theta(\delta^{1.5}), \ \tau = \mathcal{O}(1), \ T^r = \Omega(\log \lambda^{r+1}), \quad (235)$$

such that the step-sizes satisfy (143), (169), (173) and (177), which leads to $\varrho = \mathcal{O}(\sqrt{\delta})$ and the strong error bound being true again.

Finally, it can be easily checked that when $\delta > 1$, according to (234), we can have $\frac{1}{T}\sum_{r=1}^{T}\|\mathcal{G}(x^r, y^r)\|^2 \leq \mathcal{O}(1/T)$, when $\delta \leq 1$, we can obtain

$$\frac{\rho_2}{\rho_1} = \mathcal{O}(\lambda^{r+1}), \quad \frac{p}{\beta} = \Theta\left(\frac{\lambda^{r+1}}{\delta^{2.5}}\right), \quad \mathcal{Q}^1 = \mathcal{O}(p) \tag{236}$$

which gives $\frac{1}{T}\sum_{r=1}^{T}\|\mathcal{G}(x^r, y^r)\|^2 \leq \mathcal{O}(1/(\delta^{2.5}T))$.

**Constraint Violation**. Similarly, we can quantify

$$
\begin{aligned}
&|g(x^r, y^r) - g^*(x^r) - \delta|_+^2 \\
&\leq |g(x^r, y^r) - g(x^r, u^{r+1}) + g(x^r, u^{r+1}) - g^*(x^r) - \delta|_+^2 \\
&\overset{(a)}{\leq} 2|g(x^r, y^r) - g(x^r, u^{r+1}) - \delta|_+^2 + 2|g(x^r, u^{r+1}) - g^*(x^r)|_+^2 \\
&\leq \frac{2}{\tau^2}\|\lambda^{r+1} - \lambda^r\|^2 + 2|g(x^r, u^{r+1}) - g^*(x^r)|_+^2
\end{aligned}
$$

where $(a)$ holds due to $g(x^r, u^{r+1}) - g^*(x^r) \geq 0$.

From (144), we have

$$\frac{2}{\tau^2}\|\lambda^{r+1} - \lambda^r\|^2 \leq \frac{8}{\tau}\left(\mathcal{Q}^r - \mathcal{Q}^{r+1} + \left(\mu_g L_g^2 + 2\ell_g^2\right)\frac{D_{\mathcal{S}}}{r^2}\right).$$

Applying the telescoping sum yields

$$
\begin{aligned}
&\frac{1}{T}\sum_{r=1}^{T}|g(x^r, y^r) - g^*(x^r) - \delta|_+^2 \\
&\overset{(a)}{\leq} \frac{8}{\tau}\frac{\mathcal{Q}^1 - \underline{f} + \left(\mu_g L_g^2 + 2\ell_g^2\right)D_{\mathcal{S}}}{T} + \frac{4\ell_2 D_{\mathcal{S}}}{T}
\end{aligned}
\tag{237}
$$

where $(a)$ results from the fact $\sum_{r=1}^{T} r^{-2} \leq 1 + \int_1^T x^{-2}dx = 2 - T^{-1}$. Therefore, when $\delta > 1$, we select step-sizes by (234), which gives $\frac{1}{T}\sum_{r=1}^{T}|g(x^r, y^r) - g^*(x^r) - \delta|_+^2 = \mathcal{O}(1/T)$; when $0 < \delta \leq 1$, we choose step-sizes by (235), which gives $\frac{1}{T}\sum_{r=1}^{T}|g(x^r, y^r) - g^*(x^r) - \delta|_+^2 = \mathcal{O}(1/(\sqrt{\delta}T))$.

**Slackness**. If $\lambda^r = 0$, then it is trivial that $|g(x^r, y^r) - g^*(x^r) - \delta|\lambda^r$ is zero. So, we only need to consider the case where $\lambda^r > 0$ as follows.

*Step 1.* Note that

$$|g(x^r, y^r) - g^*(x^r) - \delta|^2$$

$$\leq 3|g(x^r, y^r) - g(x^*(w^r, z^r; \lambda^{r+1}), y^*(x^r, z^r; \lambda^{r+1}))|^2$$

$$+ 3|g(x^*(w^r, z^r; \lambda^{r+1}), y^*(w^r, z^r; \lambda^{r+1})) - g^*(x^*(w^r, z^r; \lambda^{r+1})) - \delta|^2$$

$$+ 3|g^*(x^*(w^r, z^r; \lambda^{r+1})) - g^*(x^r)|^2 \tag{238}$$

$$\leq 6\ell_g^2 \|y^r - y^*(w^r, z^r; \lambda^{r+1})\|^2 + 3(2\ell_g^2 + \ell_g'^2)\|x^r - x^*(w^r, z^r; \lambda^{r+1})\|^2$$

$$+ 3|g(x^*(w^r, z^r; \lambda^{r+1}), y^*(x^r, z^r; \lambda^{r+1})) - g^*(x^*(w^r, z^r; \lambda^{r+1})) - \delta|^2 \tag{239}$$

$$\leq \frac{6\ell_g^2}{\alpha^2(p-L)^2}\|y^{r+1} - y^r\|^2 + \frac{3(2\ell_g^2 + \ell_g'^2)}{\eta^2(p-L)^2}\|x^{r+1} - x^r\|^2$$

$$+ 3|g(x^*(w^r, z^r; \lambda^{r+1}), y^*(x^r, z^r; \lambda^{r+1})) - g^*(x^*(w^r, z^r; \lambda^{r+1})) - \delta|^2 \tag{240}$$

$$\leq \underbrace{\frac{3}{(p-L)^2} \max\left\{ \frac{2\ell_g^2}{\alpha^2}, \frac{2\ell_g^2 + \ell_g'^2}{2\eta^2} \right\}}_{\triangleq \rho_3} \left( \|x^{r+1} - x^r\|^2 + \|y^{r+1} - y^r\|^2 \right)$$

$$+ 3|g(x^*(w^r, z^r; \lambda^{r+1}), y^*(w^r, z^r; \lambda^{r+1})) - g^*(x^*(w^r, z^r; \lambda^{r+1})) - \delta|^2. \tag{241}$$

*Step 2.* Then, we can get

$$\frac{1}{T} \sum_{r=1}^{T} \|g(x^{r+1}, y^{r+1}) - g^*(x^{r+1}) - \delta\|^2 \|\lambda^{r+1}\|^2$$

$$\leq \frac{1}{T} \sum_{r=1}^{T} \rho_3 \left( \|x^{r+2} - x^{r+1}\|^2 + \|y^{r+2} - y^{r+1}\|^2 \right) \|\lambda^{r+1}\|^2$$

$$+ 3|g(x^*(v^{r+1}; \lambda^{r+2}), y^*(v^{r+1}; \lambda^{r+2})) - g^*(x^*(v^{r+1}; \lambda^{r+2})) - \delta|^2 \|\lambda^{r+1}\|^2 \tag{242}$$

$$\leq \left( \frac{\rho_3}{\rho_1} + \frac{3 \cdot 189}{\tau} \right) \max\{\|\lambda^{r+1}\|^2\} \frac{\mathcal{Q}^1 - \underline{f} + \left( \mu_g L_g^2 + 2\ell_g^2 \right) D_{\mathcal{S}}}{T}$$

where $(a)$ holds as when $\delta > 1$ we have $\lambda^{r+1} = \mathcal{O}(1)$, which gives $\|g(x^{r+1}, y^{r+1}) - g^*(x^{r+1}) - \delta\|^2 \|\lambda^{r+1}\|^2 = \mathcal{O}(1/T)$; when $\delta \leq 1$, we have $\rho_3/\rho_1 = \mathcal{O}(1/\lambda^{r+1})$, $\|\lambda^{r+1}\|^2/\tau \leq \mathcal{O}(1/\delta)$, so we can obtain $\|g(x^{r+1}, y^{r+1}) - g^*(x^{r+1}) - \delta\|^2 \|\lambda^{r+1}\|^2 = \mathcal{O}(1/(\delta^{3/2}T))$. $\qquad\square$

# E  Additional Numerical Results

In this section, we present additional numerical results obtained from the Fashion MNIST data set and compare the performance of SLM with other algorithms, such as BOME [15], V-PBGD [16], BVFSM [52], and ITD[13].

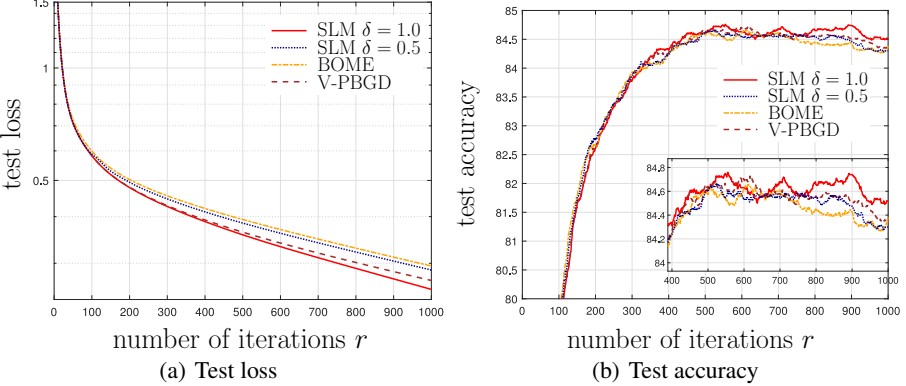

| (a) Test loss | (b) Test accuracy |

Figure 4: Convergence performance of SLM, BOME [15], and V-PBGD [16] on the neural network with the fashion MNIST data set.

## E.1  Fashion MNIST Data Set

We also evaluate the performance of these algorithms using the fashion MNIST dataset. The neural network architecture employed is the same as in the previous case. For all the compared algorithms, we set the step-sizes of the block-$x$ and block-$y$ updates to $0.1$. The step-size $\gamma$ for the auxiliary block-$u$ update is set to $0.01$. Figure 4 illustrates that SLM with $\delta = 1$ consistently outperforms the other algorithms in this example. This suggests that allowing for a certain tolerance level for the LL objective can enhance generalization performance. The convergence rates of all the algorithms are similar. However, it is important to note that both BOME and V-PBGD fail to achieve the optimal solution at the LL level since there exists a discrepancy of $c_{\text{error}} = 0.78$ for V-PBGD and $c_{\text{error}} = 0.42$ for BOME. Consequently, none of the methods can find the KKT points of problem (2) in this particular case. Nonetheless, SLM can attain the KKT solution of problem (1) with $c_{\text{error}}$ values below $1$ and $0.5$ respectively. This further underscores the advantages of utilizing this class of structured constrained problems compared to the BO formulation.

## E.2  Comparison with BVFSM and ITD w.r.t. Runtime

In addition, we conduct a numerical performance evaluation of our problem formulation and SLM algorithm on a different data hyper-cleaning task described in [15]. In this particular setup, a clipping function is applied at the lower level, and the training data consists of $50,000$ samples. We compare the performance of SLM with two other baseline algorithms: the bi-level value-function-based sequential minimization (BVFSM) algorithm introduced in [52], and the iterative differentiation (ITD) based bilevel optimizer as described in [13].

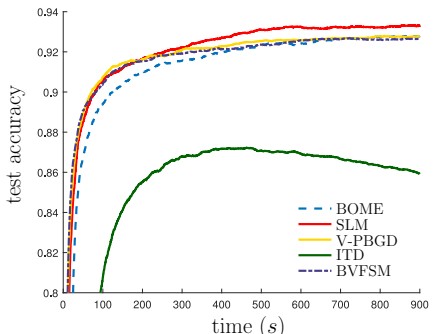

Figure 5: Convergence performance (test accuracy) of BOME, SLM, V-PBGD, ITD, and BVFSM.

The results of these experiments are presented in Figure 5. It is important to note that in this presentation, we plot test accuracy against runtime to ensure a fair comparison. The outcomes of the evaluation demonstrate that SLM achieves higher test accuracy compared to these existing benchmarks. Furthermore, the convergence speed of SLM closely aligns with that of V-PBGD and BVFSM.

# F  Ethical Considerations

This work primarily focuses on the design of optimization algorithms and the theoretical analysis of convergence. It does not involve any negative societal or ethical issues.

