# OpenReview forum: "SLM: A Smoothed First-Order Lagrangian Method for Structured Constrained Nonconvex Optimization"
_NeurIPS.cc/2023/Conference — NeurIPS 2023 poster_

### Official Review · Reviewer_VNHZ · 2023-07-04

**Soundness:** 2 fair
**Presentation:** 2 fair
**Contribution:** 2 fair
**Rating:** 4
**Confidence:** 3

**Summary:**

The paper proposes a smoothed first-order Lagrange method(SLM) with the help of a primal-dual optimization framework for solving a class of structured nonconvex functional constrained optimization (FCO) problems. It can be used to deal with lower-level (LL) nonconvex bilevel optimization(BO) problems.  Theoretically, the paper proves that the algorithm converges to an $\epsilon$-KKT points of these nonconvex constrained problems. Under some assumptions, the convergence rate of SLM to an $\epsilon$-KKT point is $\mathcal{O}(\epsilon^{-1})$ for FCO problems. SLM can achieve a convergence rate of $\mathcal{O}(\epsilon^{-2})$ to solve nonconvex BO problems when the LL problem satisfies the PŁ condition.	The paper compares the proposed method with existing ones on a toy example and a data hyper-cleaning problem. In the toy example, SLM performs better than BOME and V-PBGD. As for the data hyper-cleaning problem, it performs better than the V-PBGD.

**Strengths:**

This paper proposes a new algorithm to deal with a class of structured FCO problems, and also nonconvex BO problems.
The authors give convergence analysis when solving FCO and BO.

**Weaknesses:**

1. There are no formal definitions for $\epsilon$-KKT points and $\epsilon$-stationary points. This makes the paper difficult to follow. Moreover, when $\delta=0$, I don't think the KKT condition for bilevel optimization is well defined.

2. Assumption A4 needs more explanations.   Why can it hold for the case of $\delta=0$? It will be beneficial if the authors can give such an application.

1. There are numerous errors in the proof. Some are minor details that need to be carefully checked, but others lead to incorrect conclusions.
   - In Line 532, inequality (59),  there should be a $p$ in the gradient Lipschitz constant of $K$ with respect to $y$.
   - I cannot follow the proof of Lemma 6. Do we need a bound for $D(x^{r+1}, z^{r+1}; \lambda^{r+1}) - D(x^{r}, z^{r}; \lambda^{r+1})$?
   - In Line 567, why is $\lambda_+$  the maximizer in the expression of the objective function?


4.	The author should clarify the theoretical novelty of their compared to reference [1].

[1] Zhang, J., Xiao, P., Sun, R., & Luo, Z. (2020). A single-loop smoothed gradient descent-ascent algorithm for nonconvex-concave min-max problems. Advances in neural information processing systems, 33, 7377-7389.

**Questions:**

1. Equation (9): is there a missing \{\}?
1.  In Line 617, why does inequality (111) hold?
1.  In Line 721, E.1 Neural Network on MNIST Data Set. I am wondering why the performance is better when $\delta$ is larger. You say "The constraint tolerance parameter $\delta$ and the dual variable implicitly play the tradeoff between the training and validation losses." Is that an explanation for this phenomenon? Could you explain it more clearly?

1.  In the conditions of Theorem 1, step-sizes are chosen as $\tau, \eta, \alpha = \mathcal{O}(\frac{1}{\lambda^r})$. These step-sizes will be very small when $\lambda$ is getting large. Will this setting of step-sizes affect the effectiveness of the algorithm?
1.  In Figure 3,  why does the test accuracy of BOME get down after 400 iterations?

---

> ### Author Rebuttal · Authors · 2023-08-10
>
> We thank reviewer VNHZ for your helpful comments and questions.
>
> **Response to Weaknesses 1 (RW1)**
> $\epsilon$-KKT points and $\epsilon$-stationary points essentially refer to a point $x^*$ that satisfies the condition:
>
> \begin{equation}
> \max \\{|| \mathcal{G}(x^*,y^*)||,|g(x^*, y^*) - g^*(x^*)-\delta|_+\\}\le\epsilon.
> \end{equation}
>
> It is true that when $\delta=0$, the KKT condition of our formulation lacks a clear definition for bilevel optimization, given that the Slater condition is not satisfied. Consequently, we only discuss the method's convergence toward approximate KKT points.
>
>
> **RW2**.
> When $\delta=0$, our considered problem is not well-defined. Therefore, our focus remains solely on cases where $\delta>0$. While A4 is reasonable, since neither the feasible set nor the function value can be infinity in practice.
>
>
> **RW3.1**
> Yes, the $p$ was missing in the gradient Lipschitz constant. However, this omission will not impact the convergence result as $\lambda\sim p$.
>
>
> **W3.2**
> There were typos in the proof. The $y^{r+1}$ and $y^r$ in $D()$ should be $x^{r+1}$ and $x^r$.
>
> Yes. We also need a bound of $D(x^{r+1},z^{r+1};\lambda^{r+1})-D(x^{r},z^{r+1};\lambda^{r+1})$.
>
> To be more specific, as function $K(\cdot)$ is $L_x=L+p$-smooth and $p-L$-strongly convex, we can apply Lemma 4.3 in [Lin et al., 20 or 37] and have
>
> \begin{align}
>   D(x^{r+1},z^{r+1};\lambda^{r+1}) - D(x^r,z^{r+1};\lambda^{r+1})
> \le  \langle \nabla K(x^r,y^*(x^r,z^{r+1};\lambda^{r+1}), z^{r+1};\lambda^{r+1}), x^{r+1}-x^r\rangle + \frac{L_{K_x}}{2}||x^{r+1}-x^r||^2
> \end{align}
>
> where $L_{K_x} = L_x+L_x^2/(p-L)$. Then, following the similar steps shown from equations (47) to (54), we can first quantify
>
> \begin{align}
> & ||\nabla \hat{K}(x^r,y^{r+1},z^{r+1};\lambda^{r+1}) - \nabla K(x^r,y^*(x^r,z^{r+1};\lambda^{r+1}), z^{r+1};\lambda^{r+1}) ||
> \\\\
> \le & 3(L^2_f+\lambda^{r+1}L^2_g)||y^*(x^r,z^{r+1};\lambda^{r+1}) - y^{r+1}||^2 + 3(\lambda^{r+1})^2L^2_gd^2_{\mathcal{S}(x^r)}(u^{r+1})
> \\\\
> \le &   3(L^2_f+\lambda^{r+1}L^2_g)(1+\sigma_1)^2/\sigma^2_1||y^{r+1}-y^r||^2 + \frac{3\mu_g L^2_gD_{\mathcal{S}}}{\eta r^2}
> \end{align}
> where the first inequality holds due to gradient Lipschitz continuity; in the second inequality we utilize the primal error bound from Lemma 3.10 in [Zhang et al., 20 or 32] and the facts from equations (50) to (53). Here, $\sigma_1=\alpha (p+L)$.
>
> Combing this result with gradient Lipschitz continuity of $D(\cdot)$ mentioned above, we can have
> \begin{align}\notag
> &\quad D(x^{r+1},z^{r+1};\lambda^{r+1}) - D(x^r,z^{r+1};\lambda^{r+1})
> \\\\
> \le & -\frac{1}{4\eta}||x^{r+1} -x^r||^2+\frac{3\mu_g L^2_gD_{\mathcal{S}}}{r^2} +3\eta (L^2_f+\lambda^{r+1}L^2_g)(1+\sigma_1)^2/\sigma^2_1||y^{r+1}-y^r||^2
> \end{align}
> where $\eta\le 1/L_{K_x}$.
> It can be easily seen that the second term can be combined with the bias. The third term can be dominated by the descent accomplished by updating $y$ (given that the coefficient in eq.(60) is proportional to $1/\alpha$, whereas here, the coefficient is only proportional to $1$).
>
>
> **RW3.3** Here, we employ the optimality condition of the fixed point, which implies that $x^*=P_{\mathcal{X}}(x^*+\tau \nabla f(x^*))$ represents the optimal solution for $\max_{x\in\mathcal{X}} f(x)$. It is straightforward to verify that the function $\lambda_+(z)$ defined in equation (76) corresponds to the optimal solution of the problem presented in equation (78).
>
> To elaborate further, we can calculate the gradient of the objective shown in equation (78) at $\lambda_+(x)$, yielding $-\lambda_+(z)/\tau+\lambda/\tau+\nabla_{\lambda}K(x,y^*(x,z;\lambda),z;\lambda)$ (please note that we are scaling the objective by a constant $1/\tau$, which does not alter the optimal solution). Subsequently, we multiply this gradient by $\tau$ and examine the optimality condition. It becomes evident that $\lambda_+(z)$ defined in eq.(138) indeed stands as the optimal solution to this maximization problem.
>
> **RW4**
> Please note that the two papers address distinct optimization problems. We do utilize the primal error bounds obtained in [1], as they are independent of the dual problem. The key technical differences are as follows:
> - Here, we consider a constrained problem where the dual variable is unbounded, while in [1], the authors assume the boundedness of the maximizer directly, making the analysis more tractable.
> - The proofs of the strong error bound are entirely distinct. Here, we consider a constraint that follows the PL condition, while in [1], this bound is established based on a specific min-max problem, i.e., the minimization of the point-wise maximum of finite functions.
> - The constraint considered here belongs to a class of equilibrium constraints that did not appear in [1]. Demonstrating the convergence of the primal-dual algorithm in this problem involves new analysis for showing the descent lemma.
>
>
> **Response to Q1 (RQ1)** Yes.
>
> **RQ2** Sorry for the errors in the expressions of the error bounds. Due to the space limit, please see our response to Q1 of Reviewer aS1n.
>
> **RQ3**  Indeed, an optimal choice of $\delta$ exists. If $\delta$ is small, the constraint may limit the search space. If $\delta$ is too large, the problem essentially reduces to an unconstrained or single-level problem, lacking the capacity to perform the LL learning task. In the example, $\delta$ essentially adjusts the model's reliance on training and validation losses.
>
> **RQ4**  It holds true that if $\lambda$ is large and the step-sizes theoretically need to be small, which is reasonable. In this scenario, it implies that the constraint is not satisfied, leading to a greater value on the Lagrangian. Consequently, this results in a higher Lipschitz constant.
>
>  **RQ5**  The model optimized by BOME enters the overfitting regime after 400 iterations. In this instance, the LL problem is optimized using training data, where a distribution shift between the training and testing datasets could exist (actually indeed exist).

---

> > ### Comment · Reviewer_VNHZ · 2023-08-19
> >
> > Thanks for the rebuttal and clarification. I still have two questions on A4 and A5.
> >
> > Does A4 assume a uniform bound for any given $x$?  If so, then A4 is too strong.
> >
> > Similar question on A5: is the parameter $\mu_g$ in PL condition independent of variable $x$? If so, then A5 is too strong.
> >
> > The authors should give an example of bilevel optimization that satisfies A1-A5.

---

> > > ### Author Response · Authors · 2023-08-20
> > > **Regarding the Assumptions**
> > >
> > > Dear Reviewer VNHZ
> > >
> > > Thank you very much for acknowledging our response. Regarding your questions on A4 and A5, we are more than happy to provide further explanations as follows.
> > >
> > > *Does A4 assume a uniform bound for any given $x$? If so, then A4 is too strong.*
> > >
> > > > We assume that A4 holds true for any $x\in\mathcal{X}$. Please kindly note that this bounded level set assumption （aka coerciveness assumption), is primarily utilized to ensure the boundedness of the iterates. It has been extensively adopted in existing literature, such as A3 in [R1], A1 in [R2], and A5 in [R3], to analyze the convergence of iterative algorithms. This is a widely accepted assumption in the optimization community, commonly employed for conducting theoretical convergence analyses. Notably, this assumption can be easily satisfied. For instance, as suggested in [R1], introducing a small $\ell_2$-penalty to a non-negative loss (e.g. cross-entropy or mean-squared loss) can establish boundedness for both the level set and the iterates. An example involving neural networks can be found in Theorem 2 of [R4].
> > >
> > > > [R1] M.Arbel, J.Mairal, "Non-convex bilevel games with critical point selection maps," NeurIPS, 2022.
> > >
> > > > [R2] Y. Wang, W. Yin, and J. Zeng, "Global convergence of admm in nonconvex nonsmooth optimization," Journal of Scientific Computing, vol. 78, no. 1, pp. 29–63, 2019.
> > >
> > > > [R3] L. Cannelli, F. Facchinei, V. Kungurtsev, and G. Scutari, "Asynchronous parallel algorithms
> > > for nonconvex optimization," Mathematical Programming, vol. 184, pp. 121–154, 2020.
> > >
> > > > [R4] S. Liang, R. Sun, and R. Srikant. "Revisiting landscape analysis in deep neural networks: Eliminating decreasing paths to infinity," SIAM Journal on Optimization, vol. 32, no. 4, pp 2797-2827, 2022.
> > >
> > >
> > > We will cite the aforementioned references in the updated version.
> > >
> > >
> > > *Similar question on A5: is the parameter $\mu_g$ in PL condition independent of variable $x$? If so, then A5 is too strong.*
> > >
> > > > Yes, it is independent of $x$. Please note that this assumption is not overly strong, for the following reasons:
> > >
> > > >- A function satisfying the PL condition can be nonconvex.
> > >
> > > > - The PL condition stands as a weaker criterion when compared to various other classes of conditions, such as the weakly strongly convex condition and the restricted secant inequality, among others (as detailed in Theorem 2 in [35]).
> > >
> > > > - In the context of bilevel optimization, numerous existing studies, such as [1, 5, 8, 9, 10, 13, 17, 18, 21, 23, 24, 25], assume strong convexity for the lower-level function. Notably, strong convexity is a subset of the functions that satisfy the PL condition. In these prior works, the uniqueness of the lower-level optimal solution is typically required for algorithm design and convergence analysis. In contrast, our work does not necessitate such uniqueness (a function satisfying the PL condition can possess multiple optimal solutions). From this point of view, the assumption made at the lower level is less stringent than that of these existing works.
> > >
> > > > - Multiple loss or objective functions of machine learning problems satisfy the PL condition. For instance, the loss function of overparametrized neural networks [40], the nonconvex discounted return objective in reinforcement learning [R1], and the loss function of a specific class of Linear Quadratic Regulator (LQR) models (refer to Lemma 3 in [R2]).
> > >
> > >
> > > > [R1] J. Mei, C. Xiao, C. Szepesvari, and D. Schuurmans. "On the global convergence rates of softmax policy gradient methods," ICML, 2020.
> > >
> > > > [R2]  Fazel, Maryam, et al. "Global convergence of policy gradient methods for the linear quadratic regulator." ICML, 2018.
> > >
> > > > We will also cite the aforementioned references in the updated version.
> > >
> > >
> > >
> > > *The authors should give an example of bilevel optimization that satisfies A1-A5.*
> > >
> > > > The example of data hyper-cleaning can serve as an illustrative case of bilevel optimization that satisfies A1-A5. The loss function at the upper level can be a smooth and non-negative function, such as cross-entropy or mean-squared loss, thereby satisfying A3. By imposing a box constraint of $x\in[0,1]$, the conditions A1 for the upper-level loss function and A2 can also be satisfied. As previously noted, the introduction of a small $\ell_2$-penalty can ensure that the Lagrangian function satisfies the bounded level set assumption (A4). Moreover, when the lower-level model is overparameterized, the loss function satisfies the PL condition (A5). A concrete example can be a wide neural network, as detailed in Section 4.2 or Theorem 4 of [40 or R0]. Also, if the activation function is smooth, then the lower-level function is smooth (A1).
> > >
> > > > [R0] C. Liu, L. Zhu, and M. Belkin, "Loss landscapes and optimization in over-parameterized nonlinear systems and neural networks," Applied and Computational Harmonic Analysis, vol. 59,  pp. 85–116, 2022.
> > >
> > > Hope we have effectively addressed these concerns. Thank you for dedicating your time to reviewing our work.

---

> > > > ### Comment · Reviewer_VNHZ · 2023-08-20
> > > >
> > > > Thank you for your response. I now believe that the assumptions are reasonable, albeit somewhat limited.

---

> > > > > ### Author Response · Authors · 2023-08-20
> > > > > **Thank you for increasing the score**
> > > > >
> > > > > We are delighted to know that you have been generally satisfied with our response, and we are truly grateful that you increased the score. We hope you will provide more positive feedback on the contributions of this work during the reviewer-AC discussions. Thank you very much.
> > > > >
> > > > > Sincerely,
> > > > >
> > > > > Authors of paper #8513

---

### Official Review · Reviewer_Ggqj · 2023-07-06

**Soundness:** 3 good
**Presentation:** 3 good
**Contribution:** 3 good
**Rating:** 4
**Confidence:** 3

**Summary:**

In this paper, a smooth first-order Lagrangian method (SLM) is proposed to solve a class of structured nonconvex functional constrained optimization problems, where the two optimization variable are nonlinearly coupled. The convergence rates are established by the dual error bounds. The proposed method is used to solve bilevel optimization problems with nonconvex lower-level problems. Numerical experiments are used to show that the proposed method outperforms the compared methods.

**Strengths:**

(1) This method only uses the gradient information and obtains the optimal iteration complexity.
(2) SLM can be used to solve bilevel optimization problems with the lower-level problem satisfying the PL condition.

**Weaknesses:**

These are given in the question section.

**Questions:**

(1) This paper is able to handle nonconvexity and non-singleton at the lower level. However, the assumption that the PL condition is quite strong. If there exist multiple isolated minimizers, then such a condition does not hold.
Though [40] give a condition that guarantees the PL condition, overparameterization is not practical. Is there any example
that is practical and satisfies the PL condition?
(2) In the toy example, SLM is notably faster than V-PBGD method. However, both V-PBGD and SLM have the same complexity O(\eps^{-1}). Can the authors explain this?
(3) How to choose the parameters in Algorithm 1? Any suggestions to improve its performance? For many optimization algorithms, their performance highly depends on the choice of parameters. It is important to choose appropriate parameters when compared to other algorithms.


**Limitations:**

This paper discussed the limitations of the proposed algorithm, which is to only consider a single constraint problem.

---

> ### Author Rebuttal · Authors · 2023-08-10
>
> We thank reviewer Ggqj for your helpful comments and questions.
>
> *Q1: This paper is able to handle nonconvexity and non-singleton at the lower level. However, the assumption that the PL condition is quite strong. If there exist multiple isolated minimizers, then such a condition does not hold. Though [40] give a condition that guarantees the PL condition, overparameterization is not practical. Is there any example that is practical and satisfies the PL condition?*
>
> **R1.**
> Please kindly note that our assumption does not exclude the case where there are multiple isolated minimizers. For example, the function $\sin^2x$ satisfies the PL condition, yet it possesses multiple isolated minimizers.
>
> Overparameterized networks are indeed a common scenario, particularly in applications involving large-scale fundation models. Reference [35] demonstrates that the PL condition is relatively weak compared to other existing conditions. In the realm of bilevel optimization applications, many studies assume the strong convexity of the lower-level problem. It's worth noting that strong convexity implies the PL condition, but the reverse is not necessarily true.
>
> An example where the objective satisfies the PL condition can be found in the context of learning LQR models (please refer to Lemma 3 in [R]).
>
> [R] Fazel, Maryam, et al. "Global convergence of policy gradient methods for the linear quadratic regulator." ICML, 2018.
>
> *Q2: In the toy example, SLM is notably faster than V-PBGD method. However, both V-PBGD and SLM have the same complexity $\mathcal{O}(\epsilon^{-1}$). Can the authors explain this?*
>
> **R2.**
> It's generally understood that primal-dual methods tend to converge more quickly in numerical experiments compared to penalty-based approaches even they have the same order of the theoretical convergence rate. This is because the dual variable or Lagrange multiplier adaptively enforces constraint satisfaction throughout the optimization process.
>
>
> *Q3: How to choose the parameters in Algorithm 1? Any suggestions to improve its performance? For many optimization algorithms, their performance highly depends on the choice of parameters. It is important to choose appropriate parameters when compared to other algorithms.*
>
> **R3.**
> In the numerical results, we opt for step sizes as large as possible to anticipate a faster convergence rate. Similar to other algorithms, these step sizes are generally determined through cross-validation.

---

### Official Review · Reviewer_aS1n · 2023-07-06

**Soundness:** 3 good
**Presentation:** 3 good
**Contribution:** 2 fair
**Rating:** 6
**Confidence:** 4

**Summary:**

This work studies a class of structured nonconvex functional constrained optimization problems, where the functional constraint satisfies the PL condition w.r.t. one block of variables. The authors propose a smoothed first-order Lagrangian method (SLM) and establish the theoretical convergence rates of SLM to the KKT solutions. The authors demonstrate the application of the proposed SLM to bilevel optimization problems where the lower-level problem is nonconvex. Several experiments are provided.

**Strengths:**

(1) The paper is easy to follow. The structured constrained optimization and then the bilevel optimization has not been well studied. Finding an effective first-order method is both interesting and important.

(2) The proposed SLM algorithm achieves a near-optimal convergence rate for finding the $\epsilon$-KKT points under certain conditions. The three-layer structure of the propsoed method is interesting. The experiments in Fig. 1 and Fig. 2 support much better performance of SLM than other same-type methods in this area.

**Weaknesses:**

Major comment 1: BOME in Table 1 also covers the nonconvexity and non-singleton case, as described in Section 4.3 of their paper. So SLM is not the first work where a first-order method is capable of finding the KKT points of BO problems in the presence of nonconvexity and non-singleton at the lower level, as claimed in the main contributions of this work.

Major comment 2: For Regularity Condition in Line 201, why does the KKT solution is unique? Is the feasible set convex, and if so, why? Why is the constant $\underline{\delta}$ independent of $(x,z)$? Is the Regularity Condition too strong?

**Questions:**

Major comment 2 above is my first question. The following are others:

Q1: In Line 268, why does the strong error bound immediately come from (14) and (16)?

Q2: In Section 5.2, why do we set $\delta=1.2$ in Line 308 for the data hyper-cleaning problem? Why not choose a smaller one? Is the performance sensitive to such parameter?

Q3: Could the authors add the figure of the test accuracy and F1 score in the comparison between SLM and V-PBGD on the data hyper-cleaning problem?

Minor comments:
(1) In Line 124, is the ball open or closed?
(2) In Equation (4a), why is $g^*(x)$ smooth? It is generally not smooth when non-singleton at the lower level. When $g(x,y)$ is strongly convex w.r.t. $y$, $g^*(x)$ is smooth and its gradient does NOT require the computation of the Jacobian matrix and inverse of Hessian by Danskin’s theorem.
(3) In Equation (5), $u_{t}^{r+1}$ should be $u_{t+1}^{r}$.
(4) In Line 165, does $y^r$ be $y^{r+1}$?
(5) In Line 1 of Algorithm 1, for $r=0,1,\cdots,T-1$ do.
(6) In Corollary 1, the $\epsilon$-stationary solution of the bilevel problem (2) does not defined.
(7) In Line 233, “result in 1” should be “result in Theorem 1”.
(8) Correct the expression of $y^*(x^r, \lambda^{r+1}; z^r)$ between Lines 249 and 250, and in Line 252. Also verify the expression of $y^*$ in other parts of the paper.
(9) In Line 258, correct the definition of $\lambda_{+}(z)$.
(10) In Line 322, “optimal solution” should be “optimal value”.
(11) Please update the references.




**Limitations:**

The authors discussed part of the limitations in the conclusion section of the paper.

---

> ### Author Rebuttal · Authors · 2023-08-10
>
> We thank reviewer aS1n for your helpful comments and questions.
>
> *W1: Major comment 1: BOME in Table 1 also covers the nonconvexity and non-singleton case. So SLM is not the first work.*
>
> We have thoroughly reviewed reference [14] (i.e., BOME paper). We acknowledge that SLM is not the first work capable of finding the $\epsilon$-KKT points of BO points in the presence of nonconvexity and non-singleton at the lower level. The primary contribution lies in the fact that our derived achievable rate of SLM is $\mathcal{O}(\epsilon^{-2})$, whereas BOME achieves $\mathcal{O}(\epsilon^{-4})$. We will rectify this mistake and the associated statements.
>
> *W2: For Regularity Condition in Line 201, why does the KKT solution is unique? Is the feasible set convex, and if so, why? Why is the constant $\underline{\delta}$ independent of $(x,z)$? Is the Regularity Condition too strong?*
>
> We did not mention that the KKT solution is unique. This condition only intends to convey that given the point $(x,z)$, the constraint can be satisfied.
>
> We did not make the assumption that the feasible set of $y$ is convex. We are considering the case where $g(x,y)$ satisfies the PL condition w.r.t. $y$.
>
> Strictly speaking, $\underline{\delta}$ depends on $(x,z)$. We partially agree that this assumption is strong, as it involves a global regularity condition. However, it is not excessively stringent, given that this constraint is nonconvex. To the best of our knowledge, all the existing approaches for solving functionally nonconvex constraints require a certain type of constraint qualifications, such as the CRCQ required in BOME. Therefore, it is necessary.
>
> Thank you for bringing up this concern. In the revised version, we will address this issue by including a discussion along with a comparison to existing works.
>
> *Q1: In Line 268, why does the strong error bound immediately come from (14) and (16)?*
>
> We apologize for the errors in the expressions of the error bounds.
>
> The correct weak error bound or equation (14) is as follows:
> \begin{equation}
> ||y^*(x^r,z^r;\lambda^{r+1})-\bar{y}^*(x^r, z^r)||^2
> \le \sigma_{\textrm{weak}}\|\lambda^r -\lambda_+(z^r)\|\|\lambda(z)-\lambda_+(z)\|
> \end{equation}
> which has been already shown in equation (87). And $\sigma_{\textrm{weak}}:=(1+\tau L + \tau L\sigma_2)/(\tau (p-L))$.
>
> The correct strong error bound or equation (16) should be:
> \begin{equation}
> |\lambda^r(z^r) -\lambda_+(z^r)| \le \sigma_{\textrm{strong}}||y^*(x^r,z^r;\lambda^{r+1})-\bar{y}^*(x^r, z^r)||
> \end{equation}
> The reason is $\lambda$ in equation (148a) should be $\lambda(z)$ based on the defintion of $\bar{y}^*(x,z)$ shown in equation (10d)
>
> By combining these two results, we can immediately derive the strong error bound
> \begin{equation}\notag
> ||y^*(x^r,z^r;\lambda^{r+1})-\bar{y}^*(x^r, z^r)||\le  \sigma_{\textrm{weak}}\sigma_{\textrm{strong}}\tau |g(x^r, y^*(x^r,z^r;\lambda^{r+1})) - g^*(x^r)-\delta|,
> \end{equation}
> based on the update rule of the dual variable (or please see equations (112) and (113) ).
>
> *Q2: In Section 5.2, why do we set $\delta=1.2$ in Line 308 for the data hyper-cleaning problem? Why not choose a smaller one? Is the performance sensitive to such parameter?*
>
> The hyperparameter is determined through cross-validation. The proposed method with $\delta=1.2$ exhibits superior generalization performance compared to cases with smaller $\delta$ values. We have delved into this topic more comprehensively in the appendix; please refer to section E for a detailed discussion. We have also generated explicit plots depicting the test loss across various $\delta$ values. These plots clearly illustrate that the method with $\delta=1$ yields higher test accuracy than when $\delta=0.5$. This observation further underscores the advantages of our proposed model over existing bilevel formulations. In essence, our model offers greater flexibility for enhancing the generalization performance of the tested machine learning models.
>
> *Q3: Could the authors add the figure of the test accuracy and F1 score?*
>
> Please see the general response.
>
> *MQ1: In Line 124, is the ball open or closed?*
>
> Closed.
>
> *MQ2: In Equation (4a), why is $g^{\*}(x)$ smooth? It is generally not smooth when non-singleton at the lower level. When $g(x,y)$ is strongly convex w.r.t. $y$, $g^{\*}(x)$ is smooth and its gradient does NOT require the computation of the Jacobian matrix and inverse of Hessian by Danskin’s theorem.*
>
> That's a valid question. Even in scenarios where multiple optimal solutions exist at the lower level, the gradient of $g^*(x)$ remains unique as $g(x,y)$ satisfies the PL condition w.r.t. $y$. For more details, you can refer to Lemma A.5 in [Nouiehed et al., 2019 or 34] or Lemma 2 in [Shen et al., 2023 or 15].
>
> Exactly. The computation of this gradient doesn't necessitate the calculation of the Jacobian matrix or the inverse of the Hessian. This characteristic is why our method qualifies as a first-order approach. In contrast, the conventional bilevel formulation mandates the computation of the gradient of $f(x,y^{\*}(x))$, even in cases where $g(x,y)$ is strongly convex with respect to $y$. Additional information can be found in Lemma 2.1 in [Ghadimi et al., 2018 or 17].
>
> *MQs 3-5:* Yes. We will correct them. Many thanks.
>
> *MQ6: In Corollary 1, the $\epsilon$-stationary solution of the bilevel problem (2) does not defined.*
>
> The concept of an $\epsilon$-stationary solution for our considered problem primarily pertains to a point $x^*$ that satisfies the following inequality:
> \begin{equation}
> \max  \\{ || \mathcal{G}(x^*,y^*)||,|g(x^*, y^*) - g^*(x^*)-\delta|_+ \\}\le\epsilon.
> \end{equation}
>
> *MQs 7-11:*  We sincerely thank you for pointing out these typos. We will correct them in the updated version

---

> > ### Comment · Reviewer_aS1n · 2023-08-15
> >
> > Thanks for the rebuttal and the newly added numerical experimental results. I have updated my scores based on the rebuttal.

---

> > > ### Author Response · Authors · 2023-08-15
> > > **Thank you for increasing the score**
> > >
> > > Dear Reviewer aS1n,
> > >
> > > We truly appreciate your feedback and recognition of the importance of the newly incorporated numerical results. Your comments have undeniably elevated the quality of this work. In the updated revision, we will correct all the typos and misunderstandings as discussed above.
> > >
> > > We extend our heartfelt gratitude for your role in enhancing the score.
> > >
> > > Best regards,
> > > Authors

---

### Official Review · Reviewer_7siX · 2023-07-11

**Soundness:** 3 good
**Presentation:** 3 good
**Contribution:** 3 good
**Rating:** 6
**Confidence:** 2

**Summary:**

This paper studies structured nonconvex functional constrained problems and proposed a smoothed first-order Lagrangian method to solve it. The proposed method decomposes the complexity of the problem by introducing a proximal term. Convergence of the proposed method is also studied. In general this is a well-written paper and the results are interesting.

**Strengths:**

A new method to solve nonconvex functional constrained problem.

Convergence of the proposed method is analyzed,

**Weaknesses:**

More numerical experiments will make this paper stronger.

**Questions:**

Assumption A2 only requires the feasible set X to be compact. However, in the top layer, a projection operator is used. If the the set X is compact but the shape is very irregular, there can be multiple distinct points in X that is closest to x. In this case, how to define the projection operator? Will different choices of projection operator affect the convergence of the proposed method?

For the numerical experiments, if the feasible set X is irregular, say a star shape in R^2, how to apply the proposed method and what is the performance?

If the PL condition is not satisfied, numerically does the proposed method work?

Will the complexity of g or f affect the error and convergence rate theoretically and empirically? I do not see it from Theorem 1.

**Limitations:**

Limitations are discussed.

---

> ### Author Rebuttal · Authors · 2023-08-09
>
> We thank reviewer 7siX for your helpful comments and questions.
>
> *Weaknesses: More numerical experiments will make this paper stronger.*
>
> We have incorporated new numerical results, including new loss functions and data set partitions, as well as two additional baselines. Please kindly refer to the general response and the attached file for further details.
>
> *Q1: Assumption A2 only requires the feasible set $\mathcal{X}$ to be compact. However, in the top layer, a projection operator is used. If the the set $\mathcal{X}$ is compact but the shape is very irregular, there can be multiple distinct points in $\mathcal{X}$ that is closest to $x$. In this case, how to define the projection operator? Will different choices of projection operator affect the convergence of the proposed method?*
>
> **R1.**
> In A2, we assume that the feasible set $\mathcal{X}$ is a type of projection-friendly set, implying the existence of a closed-form projection operator. This operator ensures that a single projection step keeps the iterate within the feasible set. Your concern is indeed valid, and we will additionally incorporate the convexity of the set to make the assumption more reasonable and clear. Under this assumption, the projection operator will not affect the convergence of the proposed method.
>
>
> *Q2: For the numerical experiments, if the feasible set $\mathcal{X}$ is irregular, say a star shape in $\mathbb{R}^2$, how to apply the proposed method and what is the performance?*
>
> **R2.**
>  As mentioned in response to the first question, our current method is unable to handle cases where the feasible set is irregular.
>
>
> *Q3: If the PL condition is not satisfied, numerically does the proposed method work?*
>
> **R3.**
>  Based on our current experimental results, even when the PL condition does not hold, our method can still work without theoretical guarantees.
>
>
> *Q4: Will the complexity of $g$ or $f$ affect the error and convergence rate theoretically and empirically? I do not see it from Theorem 1.*
>
> **R4.**
> The smoothness constants of $g$ and $f$ characterize the properties of these functions. They essentially quantify the differences in gradients at two different points. If these constants are large, it implies that the gradient of the corresponding objective might change rapidly, and vice versa. Both of these constants are located in the numerator of the theoretical convergence rate (please see equation (174) in the appendix, where $\sigma_4\sim\mathcal{O}(L=L_f+L_g)$). This indicates that if the gradients change significantly at different points, the method requires a greater number of iterations to achieve convergence.
>
> In our numerical results (refer to section E), we employ various datasets, resulting in different empirical loss functions. Figure 3 and Figure 4 illustrate distinct convergence behaviors among all the compared methods, indicating that the shapes of $g$ and $f$ also numerically impact the convergence.

---

### Author Rebuttal · Authors · 2023-08-10

We sincerely thank all the reviewers for their time spent reviewing our work and greatly appreciate the comments provided by each of you. In this general response, we would like to present the F1 score and test accuracy achieved by SLM and V-PBGD, as requested by reviewer aS1n, along with the newly added numerical experimental results that include two additional baseline algorithms in response to the concerns raised by reviewer 7siX.

First, we have further plotted the F1 score and test accuracy of SLM and V-PBGD in the attached file. It can be observed that the achievable test accuracy is consistent with what has been discussed in the main text. The results imply that SLM is more robust against overfitting than V-PBGD. However, the F1 score achieved by SLM is relatively lower than that of V-PBGD, with a difference of 0.2%.


Secondly, we extended our implementation by utilizing the codebase of BOME, which is publicly available on GitHub. We proceeded to conduct a performance comparison of SLM with two additional baseline algorithms: bi-level value-function-based sequential minimization (BVFSM) [R1] and iterative differentiation (ITD) based bilevel optimizer [R2]. This comparison was carried out on the data hyper-cleaning task.

In this particular scenario, a different setup was taken compared to our prior work. We introduced a clipping function at the lower level, which differs from our previous utilization of a sigmoid function. Additionally, the dataset was divided into four subsets: the training set, validation sets 1 and 2, and the test set. This variation sets this use case apart from our previous one.

The results of these experiments are presented in Figure 2 in the attached file. It is important to note that for this presentation, we plotted the test accuracy against time, enabling a more fair comparison. The outcomes demonstrate that SLM achieves higher test accuracy in a comparsion with these existing benchmarks. Furthermore, the convergence speed of SLM closely aligns with that of V-PBGD and BVFSM. Once again, these results emphasize the superiority of our proposed formulation in enhancing the generalization performance of machine learning models.

@Reviewer aS1n and VNHZ: Our response to the concerns regarding the error bounds can be found in our response to Q1 from reviewer aS1n. Please kindly note that these errors can be easily addressed, as the results have been displayed in the appendix.

We hope that we have addressed all your major concerns. Please let us know if there are any questions.

Thank you again for your time.

Authors of #8513


[R1] Liu et al., Value-function-based sequential minimization for bi-level optimization, 2021.


[R2] Ji et al., Bilevel optimization: Convergence analysis and
enhanced design, ICML, 2021.

---

### Author Response · Authors · 2023-08-16
**Reviewer-Author Discussion Phase**

Dear Reviewers,

We extend our heartfelt gratitude for the significant time and effort you have devoted to reviewing our work and offering insightful comments. Apart from the satisfaction expressed by reviewer aS1n with our rebuttal, we kindly request others' feedback on whether the rebuttal has effectively addressed your concerns. Should any questions remain that need further discussion, we eagerly anticipate your insights.

Once again, thank you for your invaluable feedback.

Best regards,

Authors of paper #8513

---

### Decision · Program_Chairs · 2023-09-21

**Decision:**

Accept (poster)

**Comment:**

The paper proposes a new algorithm, the Smoothed First-Order Lagrange Method (SLM), to address structured nonconvex functional constrained optimization (FCO) problems and nonconvex bilevel optimization (BO) problems. While the method showcases promising results and theoretical proofs for convergence, it was criticized for lacking clarity in definitions and making unclarified assumptions

A number of issues related to the assumptions and numerical results have been adequately addressed in the rebuttal period, with reviewers increasing the score.

From my perspective, the structured constrained optimization and bilevel optimization problems tackled by this paper are not well studied, making the paper's focus both interesting and important. The paper is generally well-written and easy to follow, the authors provide theoretical proofs for the algorithm's convergence. Specifically, the proposed SLM algorithm offers near-optimal convergence rates under specific conditions. Finally, numerical experiments show that SLM outperforms existing methods in certain situations.